# Temporal Difference Flows

**Jesse Farebrother** [1 2 †]   **Matteo Pirotta** [3]   **Andrea Tirinzoni** [3]   **Rémi Munos** [3]
**Alessandro Lazaric** [3]   **Ahmed Touati** [3]

## Abstract

Predictive models of the future are fundamental for an agent's ability to reason and plan. A common strategy learns a world model and unrolls it step-by-step at inference, where small errors can rapidly compound. Geometric Horizon Models (GHMs) offer a compelling alternative by directly making predictions of future states, avoiding cumulative inference errors. While GHMs can be conveniently learned by a generative analog to temporal difference (TD) learning, existing methods are negatively affected by bootstrapping predictions at train time and struggle to generate high-quality predictions at long horizons. This paper introduces Temporal Difference Flows (TD-Flow), which leverages the structure of a novel Bellman equation on probability paths alongside flow-matching techniques to learn accurate GHMs at over $5\times$ the horizon length of prior methods. Theoretically, we establish a new convergence result and primarily attribute TD-Flow's efficacy to reduced gradient variance during training. We further show that similar arguments can be extended to diffusion-based methods. Empirically, we validate TD-Flow across a diverse set of domains on both generative metrics and downstream tasks, including policy evaluation. Moreover, integrating TD-Flow with recent behavior foundation models for planning over policies demonstrates substantial performance gains, underscoring its promise for long-horizon decision-making.

## 1. Introduction

Predictive modeling lies at the heart of intelligent decision-making, enabling agents to reason and plan in complex environments. In Reinforcement Learning (RL), this pre-dictive capability has traditionally been achieved through world models that capture the transition structure of the environment. These models have enabled significant advances across numerous domains — from robotics manipulation employing model-predictive control (Sikchi et al., 2021; Hafner et al., 2023; Hansen et al., 2022; 2024), to sample-efficient exploration strategies (Schmidhuber, 1991; Stadie et al., 2016; Pathak et al., 2017), and sophisticated planning algorithms (Silver et al., 2016; 2017; Schrittwieser et al., 2020). However, while world models have demonstrated impressive results, they face fundamental limitations when deployed for long-horizon reasoning. The standard approach of unrolling predictions step-by-step leads to compounding errors, as small inaccuracies in each prediction accumulate and propagate forward in time (Talvitie, 2014; Jafferjee et al., 2020; Lambert et al., 2022). This "curse of horizon" presents a significant challenge for applications requiring reliable long-range predictions.

An alternative approach is to learn a generative model of future states directly, avoiding compounding errors during inference. These models, usually referred to as Geometric Horizon Models (GHM; Thakoor et al., 2022) or $\gamma$-models (Janner et al., 2020), are learned by leveraging the temporal difference structure of the successor measure (Blier et al., 2021). However, their reliance on bootstrapped predictions during training can lead to instability and growing inaccuracy over long horizons. As a result, current methods struggle to make accurate predictions beyond 20-50 steps, also limiting their utility for long-term decision-making. In this paper, we show that while state-of-the-art generative methods like flow matching (Lipman et al., 2023) and denoising diffusion (Ho et al., 2020) cannot be directly applied to learn long-horizon GHMs, their iterative nature can be leveraged to better exploit the temporal difference structure of the problem. This insight yields a new class of methods that provably converges to the successor measure while reducing the variance of their sample-based gradient estimates, enabling stable long-horizon predictions. Empirically, our approach produces significantly more accurate GHMs at all horizons, consistently outperforming state-of-the-art algorithms across domains and metrics, including prediction accuracy, value function estimation, and generalized policy improvement.

[†]Work done at Meta [1]McGill University [2]Mila - Québec AI Institute [3]FAIR at Meta. Correspondence to: Jesse Farebrother <jfarebro@cs.mcgill.ca>, Ahmed Touati <atouati@meta.com>.

*Proceedings of the $42^{nd}$ International Conference on Machine Learning*, Vancouver, Canada. PMLR 267, 2025. Copyright 2025 by the author(s).

## 2. Background

In the following, we use capital letters to denote random variables, sans-serif fonts for sets, and $\mathscr{P}(\mathsf{A})$ to denote the space of probability measures over a measurable set $\mathsf{A}$.

**Markov Decision Process** We consider a reward-free discounted Markov decision process $\mathcal{M} = (\mathsf{S}, \mathsf{A}, P, \gamma)$, which characterizes the dynamics of a sequential decision-making problem. At each step, the agent selects an action $a \in \mathsf{A}$ in state $s \in \mathsf{S}$ according to its policy $\pi : \mathsf{S} \to \mathsf{A}$. This action influences the transition to the next state $s' \in \mathsf{S}$, governed by the transition kernel $P : \mathsf{S} \times \mathsf{A} \to \mathscr{P}(\mathsf{S})$, which defines a probability measure over successor states. The discount factor $\gamma \in [0, 1)$ can be interpreted as implying a process that either continues with probability $\gamma$ or terminates with probability $1 - \gamma$. This interpretation naturally defines a geometric distribution of future states the agent will occupy, where states reached after $k$ steps are discounted by $\gamma^k$.

**Successor Measure** The normalized *successor measure* (Dayan, 1993; Blier et al., 2021) of a policy $\pi$ describes the discounted distribution of future states visited by $\pi$ starting from an initial state-action pair $(s, a)$. For the measurable subset $\mathsf{X} \subseteq \mathsf{S}$ the successor measure $m^\pi(\mathsf{X} \mid s, a)$ represents the probability that future states fall within $\mathsf{X}$, geometrically discounted by $\gamma$ according to the time of visitation. Formally, it is defined as:

$$m^\pi(\mathsf{X} \mid s, a) = \tag{1}$$
$$(1 - \gamma) \sum_{k=0}^{\infty} \gamma^k \Pr(S_{k+1} \in \mathsf{X} \mid S_0 = s, A_0 = a, \pi),$$

where $\Pr(\cdot \mid S_0, A_0, \pi)$ denotes the probability of state-action sequences $(S_k, A_k)_{k \geq 0}$ generated from $(S_0, A_0)$ following $S_k \sim P(\cdot \mid S_{k-1}, A_{k-1})$ and $A_k = \pi(S_k)$. The successor measure encapsulates the long-term dynamics of $\pi$, enabling value estimation for any reward function $r : \mathsf{S} \to \mathbb{R}$. Specifically, the value of taking action $a \in \mathsf{A}$ in state $s \in \mathsf{S}$ is the expected reward under states visited by $\pi$ amplified by the effective horizon $(1 - \gamma)^{-1}$:

$$Q^\pi(s, a) = (1 - \gamma)^{-1} \mathbb{E}_{X \sim m^\pi(\cdot \mid s, a)}\big[r(X)\big]. \tag{2}$$

Moreover, $m^\pi$ is the fixed point of the Bellman operator $\mathcal{T}^\pi : \mathscr{P}(\mathsf{S})^{\mathsf{S} \times \mathsf{A}} \to \mathscr{P}(\mathsf{S})^{\mathsf{S} \times \mathsf{A}}$ (Thakoor et al., 2022):

$$m^\pi(\cdot \mid s, a) = (\mathcal{T}^\pi m^\pi)(\cdot \mid s, a) \tag{3}$$
$$:= (1 - \gamma) P(\cdot \mid s, a) + \gamma (P^\pi m^\pi)(\cdot \mid s, a).$$

The operator $P^\pi$ applied to $m$ mixes the one-step kernel with the successor measure, accounting for transitioning from $(s, a)$ to a new state-action pair $(s', \pi(s'))$ and querying the successor measure $m(\cdot \mid s,' \pi(s'))$ thereafter:

$$(P^\pi m)(\mathrm{d}x \mid s, a) = \int_{s'} P(\mathrm{d}s' \mid s, a) \, m(\mathrm{d}x \mid s', \pi(s')).$$

**Geometric Horizon Model** A *Geometric Horizon Model* (GHM; Thakoor et al., 2022) or $\gamma$-model (Janner et al., 2020) is a generative model of the *normalized* successor measure. To learn the parametric model $\widetilde{m}(\cdots ; \theta) \approx m^\pi$ we can minimize a Monte-Carlo cross-entropy objective over source states from the empirical distribution $\rho$ as,

$$\mathbb{E}_{s \sim \rho, \, X \sim m^\pi(\cdot \mid S, \pi(A))}\big[ -\log \widetilde{m}(X \mid S, A; \theta))\big].$$

In order to sample from $m^\pi$ we deploy policy $\pi$ for $t \sim \mathrm{Geom}(1 - \gamma)$ steps resulting in state $X = S_t$. Similar to other Monte Carlo methods in RL, this approach is problematic when learning from off-policy data, often resulting in high-variance estimators that rely on importance sampling.

Alternatively, we can leverage the Bellman equation (3) to construct an off-policy iterative method for estimating $m^\pi$. Given initial weights $\theta^{(0)}$, each iteration updates $\theta$ by minimizing the following temporal-difference cross-entropy objective over transitions that need not come from policy $\pi$,

$$\mathbb{E}_{(S,A) \sim \rho, \, X \sim (\mathcal{T}^\pi \widetilde{m}^{(n)})(\cdot \mid S, A)}\big[-\log \widetilde{m}(X \mid S, A; \theta)\big]. \tag{4}$$

In the equation above and throughout the paper, we adopt the shorthand $\widetilde{m}^{(n)} = \widetilde{m}(\cdots ; \theta^{(n)})$. To generate samples $X \sim (\mathcal{T}^\pi \widetilde{m}^{(n)})(\cdot \mid S, A)$ we first draw a successor state $S' \sim P(\cdot \mid S, A)$; then with probability $1 - \gamma$, we return $S'$; otherwise, with probability $\gamma$, we return a *bootstrapped sample* drawn from $\widetilde{m}^{(n)}(\cdot \mid S', \pi(S'))$.

Several probabilistic models have been applied to this problem, including generative adversarial networks (e.g., Janner et al., 2020; Wiltzer et al., 2024b), normalizing flows (e.g., Janner et al., 2020), and variational auto-encoders (e.g., Thakoor et al., 2022; Tomar et al., 2024). We now turn our attention to a class of generative models based on the flow-matching framework specifically designed to leverage the underlying structure of the Bellman equation (3), enabling more effective generative models of the successor measure.

## 3. Temporal Difference Flows

Flow Matching (FM; Lipman et al., 2023; 2024; Liu et al., 2023; Albergo & Vanden-Eijnden, 2023) constructs a time-dependent probability path $m_t : \mathsf{S} \times \mathsf{A} \to \mathscr{P}(\mathsf{S})$ for $t \in [0, 1]$ that evolves smoothly from the source distribution $m_0 = p_0 \in \mathscr{P}(\mathsf{S})$ to the target distribution $m_1 \approx m^\pi$. This evolution is governed by a vector field $v_t : \mathsf{S} \times \mathsf{S} \times \mathsf{A} \to \mathsf{S}$, which dictates the instantaneous movement of samples along $m_t$. The relationship between $v_t$ and the resulting probability path $m_t$ is established through a time-dependent flow $\psi_t : \mathsf{S} \times \mathsf{S} \times \mathsf{A} \to \mathsf{S}$, defined by the following ODE:

$$\frac{\mathrm{d}}{\mathrm{d}t} \psi_t(x \mid s, a) = v_t\big(\psi_t(x \mid s, a) \mid s, a\big), \; \psi_0(x \mid s, a) = x$$

$$\iff \psi_t(x \mid s, a) = x + \int_0^t v_\tau\big(\psi_\tau(x \mid s, a) \mid s, a\big) \mathrm{d}\tau.$$

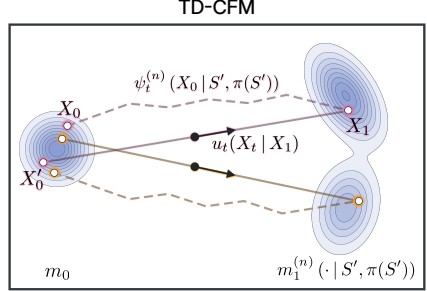
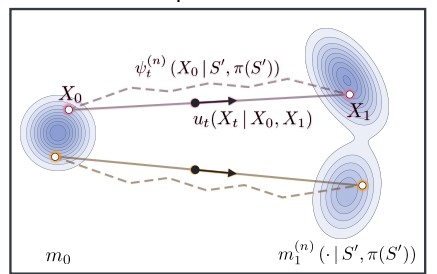
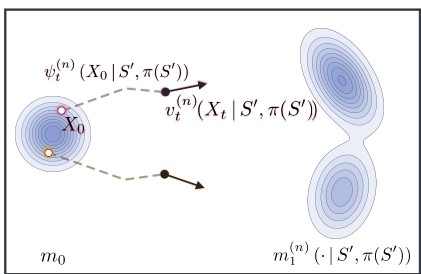

*Figure 1.* Visual depiction of TD-Flow variants. Samples are mapped from $m_0$ to the target distribution $m_1^{(n)}$ through the neural ODE $\psi_t^{(n)}$. Dashed lines depict the neural ODE trajectory; solid lines show the conditional probability path $u_t$. **(Left)** TD-CFM maps $X_0$ to $X_1$ before creating a separate conditional path between $X_0'$ and $X_1$, resulting in crossing paths. **(Middle)** TD-CFM(C) directly couples $X_0$ used to generate $X_1$ when constructing the conditional probability path. **(Right)** TD$^2$-CFM solves the neural ODE up to time $t$ to directly obtain the target velocity $\tilde{v}_t$.

We say that $v_t$ generates $m_t$ if its flow $\psi_t$ satisfies $X_t := \psi_t(X_0 \mid S, A) \sim m_t(\cdot \mid S, A)$ for $X_0 \sim m_0$. In words, the flow $\psi_t$ pushes samples forward through time, ensuring they are distributed according to $m_t$ at time $t$. To learn this transformation, we can minimize the squared $L^2$ distance between a parameterized vector field $\tilde{v}_t(\cdots ; \theta)$ and the true vector field $v_t$ over $t \sim \mathcal{U}([0,1])$, yielding the Monte-Carlo Flow Matching (MC-FM) loss $\ell_{\text{MC-FM}}(\theta)$:

$$\mathbb{E}_{\rho,t,X_t}\left[\left\|\tilde{v}_t(X_t \mid S, A; \theta) - v_t(X_t \mid S, A)\right\|^2\right],$$
$$\text{where } X_t \sim m_t(\cdot \mid S, A). \qquad \text{(MC-FM; 5)}$$

Despite its conceptual simplicity, direct optimization of the flow matching objective above proves challenging due to the inaccessibility of the true probability path $m_t$ and its associated vector field $v_t$.

Alternatively, Lipman et al. (2023) shows that we can sidestep this problem entirely by introducing additional conditioning information. Instead of directly modeling the probability path $m_t$ we can introduce a random variable $Z$ and define a *conditional path* on $Z$ as $p_{t|Z} : \mathsf{S} \times \mathsf{Z} \to \mathscr{P}(\mathsf{S})$ (Lipman et al., 2024; Tong et al., 2024). The conditional velocity field $u_{t|Z} : \mathsf{S} \times \mathsf{Z} \to \mathsf{S}$ that generates $p_{t|Z}$ can now be computed in closed form for many simple choices of $Z$ and $p_{t|Z}$. One such choice is taking $Z = X_1$ and performing a linear Gaussian interpolation from $X_0 \to X_1$ resulting in $p_{t|1}(\cdot \mid X_1) = \mathcal{N}(\cdot \mid tX_1, (1-t)^2 I)$ with the corresponding vector field given by $u_{t|1}(x \mid X_1) = (X_1 - x)/(1-t)$. Armed with the ability to sample from $p_{t|1}$ and to compute $u_{t|1}$, we can directly learn $\tilde{v}_t$ by optimizing the Monte-Carlo *Conditional* Flow Matching (MC-CFM) objective $\ell_{\text{MC-CFM}}(\theta)$:

$$\mathbb{E}_{\rho,t,Z,X_t}\left[\left\|\tilde{v}_t(X_t \mid S, A; \theta) - u_{t|Z}(X_t \mid Z)\right\|^2\right],$$
$$\text{where } Z = X_1 \sim m^\pi(\cdot \mid S, A), X_t \sim p_{t|Z}(\cdot \mid Z). \qquad \text{(MC-CFM; 6)}$$

Remarkably, both (MC-FM; 5) and (MC-CFM; 6) share the same gradient and converge to the same solution.

**Proposition 1** (Lipman et al. 2024). *Given a conditional probability path $p_{t|Z}$ and vector field $u_{t|Z}$ with their associated marginal counterparts $p_t(x)$ and $v_t(x)$, we have*

$$\nabla_\theta \ell_{\text{MC-FM}}(\theta) = \nabla_\theta \ell_{\text{MC-CFM}}(\theta).$$

**TD-CFM** While (MC-CFM; 6) requires direct access to samples from the target distribution $m^\pi$, we can instead learn from an offline dataset $\rho$ containing only one-step transitions $(S, A, S')$ through an iterative process similar to (4). Starting with initial parameters $\theta^{(0)}$, at each iteration, we minimize the TD-Conditional Flow Matching (TD-CFM) loss $\ell_{\text{TD-CFM}}$ — an extension of (MC-CFM; 6) that differs only in its sampling procedure:

$$X_0 \sim p_0$$
$$Z = X_1 \sim (1-\gamma)\,\delta_{S'} + \gamma\,\delta_{\widetilde{\psi}_1^{(n)}(X_0 \mid S', \pi(S'))} \cdot \qquad \text{(TD-CFM; 7)}$$

In this procedure, with probability $1 - \gamma$, we return the successor state $S'$. Otherwise, with probability $\gamma$ we sample from the neural ordinary differential equation (Chen et al., 2018) $\widetilde{\psi}_t^{(n)}$ with corresponding vector field $\tilde{v}_t^{(n)}(X_t \mid S', \pi(S'))$ from $X_0 \sim p_0$ to produce a sample $X_1 \sim \widetilde{m}^{(n)}(\cdot \mid S', \pi(S'))$.

**Coupled TD-CFM** Although (TD-CFM; 7) offers a principled way of learning the flow from noise to data, an increasingly popular strategy to improve flow matching methods is to correlate noise and data whenever a "natural" coupling is available (e.g., Liu et al., 2023; Shi et al., 2023; Pooladian et al., 2023; Tong et al., 2024; De Bortoli et al., 2024). Motivated by this idea, we observe that the process used to generate $X_1$ described above already provides a direct coupling between $X_0$ and $X_1$. We can leverage this coupling by conditioning the probability path $p_{t|Z}$ on both endpoints, i.e., $Z = (X_0, X_1)$, rather than just conditioning on $Z = X_1$ as in TD-CFM. As illustrated in Figure 1, this coupling helps align $X_t$ with the path generated by

$\widetilde{\psi}_t^{(n)}$, potentially simplifying the regression problem. This procedure gives rise to the Coupled TD-Conditional Flow Matching (TD-CFM(C)) loss $\ell_{\text{TD-CFM(C)}}$ which now extends $\ell_{\text{TD-CFM}}$, again, differing only in its sampling procedure:

$$X_0 \sim p_0$$
$$X_1 \sim (1-\gamma)\,\delta_{S'} + \gamma\,\delta_{\widetilde{\psi}_1^{(n)}(X_0|S',\pi(S'))}$$
$$Z = (X_0, X_1)\,. \qquad\text{(TD-CFM(C); 8)}$$

A convenient approach to specifying the conditional path $p_{t|Z}$ is to define $X_t = \phi_t(X_0, X_1) = \alpha_t X_1 + \beta_t X_0$ as the affine interpolant between $X_0$ and $X_1$, with the interpolation coefficients satisfying the boundary conditions $\alpha_0 = \beta_1 = 0$, $\alpha_1 = \beta_0 = 1$, and monotonicity constraints $\dot{\alpha}_t > 0, -\dot{\beta}_t > 0$, where the over-dot denotes the time derivative. From this definition, the conditional vector field arises as the time derivative of this interpolant defined as $u_{t|0,1}(X_t \mid X_0, X_1) = \dot{\phi}_t(X_0, X_1) = \dot{\alpha}_t X_1 + \dot{\beta}_t X_0$ (Albergo et al., 2023). A simple choice of the interpolation coefficients that yields a linear (straight-line) conditional path is given by $\beta_t = 1 - \alpha_t = 1 - t$.

**TD²-CFM** While (TD-CFM(C); 8) improves upon (TD-CFM; 7) by accounting for the coupling between bootstrapped samples and their generating noise, both methods rely upon fitting an ad-hoc conditional vector field $u_{t|Z}$ that generates the surrogate conditional path $p_{t|Z}$. To formulate a more structured approach, we exploit the linearity of the Bellman equation, as detailed in the following result.

**Lemma 1.** *Let $\vec{p}_t$ be a probability path for $P$ generated by vector field $\vec{v}_t$ and $\widehat{p}_t^{(n)}$ be a probability path for $P^\pi m_1^{(n)}$ generated by $\widehat{v}_t^{(n)}$ such that $\vec{p}_0 = \widehat{p}_0^{(n)} = m_0$. For any $t \in [0, 1]$ and $(s, a)$ let $v_t^{(n+1)}(\cdot \mid s, a)$ be the solution of[1]*

$$\underset{v:\mathbb{R}^d \to \mathbb{R}^d}{\arg\min}\,(1-\gamma)\mathbb{E}_{\vec{X}_t \sim \vec{p}_t(\cdot|s,a)}\left[\left\|v(\vec{X}_t) - \vec{v}_t(\vec{X}_t \mid s, a)\right\|^2\right]$$
$$+ \gamma\mathbb{E}_{\widehat{X}_t \sim \widehat{p}_t^{(n)}(\cdot|s,a)}\left[\left\|v(\widehat{X}_t) - \widehat{v}_t^{(n)}(\widehat{X}_t \mid s, a)\right\|^2\right].$$

*Then $v_t^{(n+1)}$ induces a probability path $m_t^{(n+1)}$ such that $m_0^{(n+1)} = m_0$ and $m_1^{(n+1)} = \mathcal{T}^\pi m_1^{(n)}$.*

This result shows that it is possible to use two independent probability paths for the two terms in the sampling process induced by the Bellman operator. For the first term, we can use a standard CFM approach for $Z = X_1$ with conditional path $\vec{p}_{t|1}$ and vector field $\vec{u}_{t|1}$, which induces the marginal,

$$\vec{v}_t(x|s,a) = \int \vec{u}_{t|1}(x \mid x_1)\frac{\vec{p}_{t|1}(x \mid x_1)P(\mathrm{d}x_1|s,a)}{\vec{p}_t(x|s,a)},$$

---

[1]Notice here that the minimization is over the space of all functions and not the parameterized vector fields $\tilde{v}_t(\cdots;\theta)$.

where $\vec{p}_t(x|s,a) = \int \vec{p}_{t|1}(x|s')P(\mathrm{d}s'|s,a)$. For the second term, we can leverage the GHM $m_t^{(n)}$ learned at the previous iteration to construct the marginal,

$$\widehat{v}_t^{(n)}(x|s,a) = \int v_t^{(n)}(x|s',a')\frac{m_t^{(n)}(x|s',a')P(\mathrm{d}s'|s,a)}{\widehat{p}_t^{(n)}(x|s,a)},$$

where $\widehat{p}_t^{(n)}(x \mid s, a) = \int m_t^{(n)}(x \mid s', a')P(\mathrm{d}s' \mid s, a)$, and $a' = \pi(s')$. This shows that $m_t^{(n)}$ plays the role of a conditional probability path for the bootstrapped term and $v_t^{(n)}$ is its associated conditional vector field. We can then use the equivalence between FM and CFM in Proposition 1 to replace the marginal probability paths and vector fields in Lemma 1 with their conditional counterparts to obtain the loss:

$$\vec{\ell}(\theta) = \mathbb{E}_{\rho,t,Z,\vec{X}_t}\left[\left\|\tilde{v}_t(\vec{X}_t \mid S, A; \theta) - \vec{u}_{t|Z}(\vec{X}_t \mid Z)\right\|^2\right],$$
$$\text{where } Z = X_1 \sim P(\cdot \mid S, A),\ \vec{X}_t \sim \vec{p}_{t|Z}(\cdot \mid Z)\,,$$

$$\widehat{\ell}(\theta) = \mathbb{E}_{\rho,t,\widehat{X}_t}\left[\left\|\tilde{v}_t(\widehat{X}_t \mid S, A; \theta) - \tilde{v}_t^{(n)}(\widehat{X}_t \mid S', \pi(S'))\right\|^2\right],$$
$$\text{where } X_0 \sim p_0,\ S' \sim P(\cdot \mid S, A),$$
$$\widehat{X}_t = \widetilde{\psi}_t^{(n)}(X_0 \mid S', \pi(S'))\,,$$

$$\ell_{\text{TD}^2\text{-CFM}}(\theta) = (1-\gamma)\vec{\ell}(\theta) + \gamma\widehat{\ell}(\theta)\,. \qquad\text{(TD²-CFM; 9)}$$

Since we now bootstrap the previous estimate not only in the sampling process but also in the objective function, we refer to this method as TD²-Conditional Flow Matching (TD²-CFM). The right panel of Figure 1 depicts the process of obtaining the bootstrapped vector field $\tilde{v}_t^{(n)}$ for TD²-CFM. We provide further implementation details and pseudo-code for all TD-Flow methods in Appendix C.3.1. Next, we extend our TD² result to the class of denoising diffusion models.

### 3.1. Extension to Diffusion Models

Denoising Diffusion models (Sohl-Dickstein et al., 2015; Ho et al., 2020) build a diffusion process starting from a data sample $X_0 \sim q_0 = m^\pi(\cdot \mid S, A)$[2] and corrupting it via a stochastic differential equation (SDE),

$$\mathrm{d}X_t = f(t)\,X_t\,\mathrm{d}t + g(t)\,\mathrm{d}W_t\,, \qquad (10)$$

where $t \in [0, T]$ for some time horizon $T$, $f, g : [0, T] \to \mathbb{R}$ is drift and diffusion term, and $W_t \in \mathbb{R}^d$ is a standard Brownian motion. The forward process of the linear SDE (10) has an analytic Gaussian kernel $q_{t|0}(\cdot \mid X_0) = \mathcal{N}(\cdot \mid \alpha_t X_0, \sigma_t^2 I)$, where $\alpha_t$ and $\sigma_t$ can be computed in closed form. To sample from the target data distribution $q_0$, we can solve the reverse SDE (Song & Ermon, 2019) from time $T$ to 0:

$$\mathrm{d}X_t = \Big(f(t)\,X_t - g(t)\,\nabla_{X_t}\log q_t(X_t \mid S, A)\Big)\mathrm{d}t + g(t)\,\mathrm{d}\overline{W}_t$$
$$(11)$$

---

[2]Different to flow matching, time is inverted in diffusion models and ranges from 0 to $T$.

where $\overline{W}_t$ is the reverse-time Brownian motion and $q_t$ is the marginal distribution of both the forward (16) and reverse (17) process. To simulate (11), we can train a parametrized score function $\tilde{s}_t(x \mid s, a; \theta)$ to approximate $\nabla_{x_t} \log q_t(x_t \mid s, a)$ using the denoising diffusion / score matching objective (Vincent, 2011) $\ell_{\text{DD}}(\theta)$:

$$\mathbb{E}_{\rho, t, X_0, X_t}\left[\left\|\tilde{s}_t(X_t \mid S, A; \theta) - \nabla_{X_t} \log q_{t|0}(X_t \mid X_0)\right\|^2\right],$$

$$\text{where } X_0 \sim m^\pi(\cdot \mid S, A), \ X_t \sim q_{t|0}(\cdot \mid X_0). \quad \text{(DD; 12)}$$

**Temporal Difference Diffusion** Following the blueprint in §3, we define an iterative process starting from $\tilde{s}^{(0)} = \tilde{s}(\cdots; \theta^{(0)})$ and minimize at each iteration the Temporal-Difference Denoising Diffusion (TD-DD) loss $\ell_{\text{TD-DD}}(\theta)$:

$$\mathbb{E}_{\rho, t, X_0, X_t}\left[\left\|\tilde{s}(X_t \mid S, A; \theta) - \nabla_x \log q_{t|0}(X_t \mid X_0)\right\|^2\right],$$

$$\text{where } X_0 \sim \left(\mathcal{T}^\pi \widetilde{m}_{0|T}^{(n)}\right)(\cdot \mid S, A), \ X_t \sim q_{t|0}(\cdot \mid X_0).$$
$$\text{(TD-DD; 13)}$$

Once again, to sample $X_0 \sim \left(\mathcal{T}^\pi \widetilde{m}_{0|T}^{(n)}\right)(\cdot \mid S, A)$, we proceed as follows: with probability $1 - \gamma$, we draw a successor state $S' \sim P(\cdot \mid S, A)$; conversely, with probability $\gamma$, we sample from the bootstrapped model by solving the reverse SDE with score function $\tilde{s}^{(n)}$, initiated from $X_T$. Following an approach analogous to Lemma 1, we demonstrate in Appendix B that we can employ two distinct diffusion processes for the two terms involved in the Bellman operator, which consequently leads to the TD²-DD objective:

$$\vec{\ell}(\theta) = \mathbb{E}_{\rho, t, \vec{X}_t}\left[\left\|\tilde{s}_t(\vec{X}_t \mid S, A; \theta) - \nabla_{\vec{X}_t} q_{t|0}(\vec{X}_t \mid S')\right\|^2\right],$$
$$\text{where } \vec{X}_t \sim q_{t|0}(\cdot \mid S'),$$

$$\widehat{\ell}(\theta) = \mathbb{E}_{\rho, t, \widehat{X}_t}\left[\left\|\tilde{s}_t(\widehat{X}_t \mid S, A; \theta) - \tilde{s}_t^{(n)}(\widehat{X}_t \mid S', \pi(S'))\right\|^2\right],$$
$$\text{where } X_T \sim q_T, \ \widehat{X}_t \sim q_{t|T}^{(n)}(\cdot \mid S', \pi(S')),$$

$$\ell_{\text{TD}^2\text{-DD}}(\theta) = (1 - \gamma)\vec{\ell}(\theta) + \gamma\widehat{\ell}(\theta). \quad \text{(TD}^2\text{-DD; 14)}$$

## 4. Theoretical Analysis

We now study the learning dynamics of an idealized version of the TD-Flow methods, assuming that the flow-matching loss is minimized exactly at each iteration. Under this assumption, at each iteration we compute a probability path $m_t^{(n)}$ such that $m_1^{(n)} = \mathcal{T}^\pi m_1^{(n-1)}$, which implies that $m_1^{(n)} \to m^\pi$ by the contraction property of $\mathcal{T}^\pi$. The following result shows that the overall probability paths $m_t^{(n)}$ follow a similar process. Proofs are deferred to Appendix E.

**Theorem 1.** *For any $n \geq 1$, the probability paths generated by* TD-CFM, TD-CFM(C), *or* TD²-CFM *satisfy*

$$m_t^{(n+1)}(x \mid s, a) = \left(\mathcal{B}_t^\pi m_t^{(n)}\right)(x \mid s, a), \ \forall t \in [0, 1]$$

where $\mathcal{B}_t^\pi m := (1 - \gamma)P_t + \gamma P^\pi m$ and $P_t(x|s, a) := \int p_{t|1}(x \mid x_1)P(x_1|s, a)\mathrm{d}x_1$. *For any $t \in [0, 1]$, the operator $\mathcal{B}_t^\pi$ is a $\gamma$-contraction in 1-Wasserstein distance, that is, for any couple of probability paths $p_t, q_t$,*

$$\sup_{s, a} W_1\left((\mathcal{B}_t^\pi p_t)(\cdot \mid s, a), (\mathcal{B}_t^\pi q_t)(\cdot \mid s, a)\right)$$
$$\leq \gamma \sup_{s, a} W_1\left(p_t(\cdot \mid s, a), q_t(\cdot \mid s, a)\right).$$

Theorem 1 shows that all TD-flow methods fundamentally implement the same update where the probability path at $t \in [0, 1]$ is obtained by applying a Bellman-like operator $\mathcal{B}_t$ to the previous iteration. This operator is a $\gamma$-contraction as $\mathcal{T}^\pi$, directly implying the following result.

**Corollary 1.** *Let $\{m_t^{(n)}\}_{n \geq 0}$ be the sequence of probability paths produced by* TD-CFM, TD-CFM(C), *or* TD²-CFM *starting from an arbitrary vector field $v_t^{(0)}$. Then,*

$$\lim_{n \to \infty} m_t^{(n)} = \overline{m}_t = \mathcal{B}_t \overline{m}_t,$$

*where $\overline{m}_t$ is the unique fixed point of $\mathcal{B}_t$, and $\overline{m}_t = m_t^{\text{MC}}$, where $m_t^{\text{MC}}(\cdot \mid s, a) = \int p_{t|1}(\cdot \mid x_1) m^\pi(x_1 \mid s, a)$ is the probability path of the Monte-Carlo approach (*MC-CFM; 6*).*

This corollary shows that the fixed point of $\mathcal{B}_t$ coincides with the probability path generated in Monte-Carlo Conditional Flow Matching (MC-CFM; 6), which assumes direct access to samples of $m^\pi$. An important subtlety in Theorem 1 is that all algorithms apply the same operator for $n \geq 1$, but the result holds for $n = 0$ only for TD²-CFM. This means that even starting from the same $\theta^{(0)}$, the three algorithms may generate different sequences $\{m_t^{(n)}\}_{n \geq 0}$, while still converging to $\overline{m}_t$. In Theorems 5 and 6, we show we can reconcile TD-CFM(C) and TD-CFM with TD²-CFM under a mild assumption on the form of the initial vector field.

While Theorem 1 analyzes an idealized version of the algorithms, in practice gradients are estimated from samples and the following analysis reveals important differences in their variance. We introduce the (unbiased) sample-based gradients for each of the algorithms,

$$\mathbb{E}\left[g_{\text{TD-CFM}}(Y_{\text{TD-CFM}})\right] = \nabla_\theta \ell_{\text{TD-CFM}}(\theta),$$
$$\mathbb{E}\left[g_{\text{TD-CFM(C)}}(Y_{\text{TD-CFM(C)}})\right] = \nabla_\theta \ell_{\text{TD-CFM(C)}}(\theta)$$
$$\mathbb{E}\left[g_{\text{TD}^2\text{-CFM}}(Y_{\text{TD}^2\text{-CFM}})\right] = \nabla_\theta \ell_{\text{TD}^2\text{-CFM}}(\theta),$$

where $Y$ summarizes the random variables involved in the loss definitions in (TD-CFM; 7), (TD-CFM(C); 8), and (TD²-CFM; 9) (see Appendix E.6 for a formal definition of the gradients). We want to compare the total variance of the gradient estimates $\sigma^2 = \text{Tr}\left(\text{Cov}_Y\left[g(Y)\right]\right)$, where Tr denotes the trace.

**Theorem 2.** *For any $n \geq 1$ and $t \in [0,1]$, assume that $m_t^{(n)}(x \mid s,a) = \int p_{t|1}(x \mid x_1)m_1^{(n)}(x_1 \mid s,a)\mathrm{d}x_1$, then*

$$\sigma_{\text{TD-CFM}}^2 = \sigma_{\text{TD}^2\text{-CFM}}^2 +$$
$$\gamma^2\, \mathbb{E}\left[\text{Tr}\big(\text{Cov}_{X_1|s,a,X_t}\big[\nabla_\theta v_t(X_t|s,a;\theta)^\top u_{t|1}(X_t|X_1)\big]\big)\right].$$

**Theorem 3.** *For any $n \geq 1$ and $t \in [0,1]$, assume that $m_t^{(n)}(x \mid s,a) = \int p_{t|0,1}(x \mid x_0,x_1)m_{0,1}^{(n)}(x_0,x_1 \mid s,a)\mathrm{d}x_0\mathrm{d}x_1$ [3], then we obtain*

$$\sigma_{\text{TD-CFM(C)}}^2 = \sigma_{\text{TD}^2\text{-CFM}}^2 +$$
$$\gamma^2\mathbb{E}\left[\text{Tr}\big(\text{Cov}_{Z|S,A,X_t}\big[\nabla_\theta v_t(X_t|S,A;\theta)^\top u_{t|Z}(X_t|Z)\big]\big)\right],$$

*where $Z = (X_0, X_1)$. Furthermore, if we use straight conditional paths, i.e., $X_t = tX_1 + (1-t)X_0$, and the linear interpolant $X_t$ does not intersect for any $s, a, s'$, then $\sigma_{\text{TD-CFM(C)}}^2 = \sigma_{\text{TD}^2\text{-CFM}}^2$.*

In both results, the probability path $m_t^{(n)}$ from the previous iteration must be identical for the algorithms being compared. The analysis reveals that TD-CFM and TD-CFM(C) suffer from a larger variance compared to $\text{TD}^2$-CFM, which uses the vector field $v^{(n)}$ both to sample $X_t$ and as a target for the regression problem. This variance gap is "discounted" by $\gamma^2$, which suggests that the performance of these algorithms would be similar for problems with small horizons but would increase as $\gamma \to 1$. The extra variance in both cases stems from samples generated by the algorithm (i.e., they do not depend on the transitions available in the dataset). In this sense, we can refer to it as *computational variance*, and in principle, it could be reduced by increasing the number of samples $X_0$, $X_1$, and $X_t$ used in gradient computation. While the variance of TD-CFM and TD-CFM(C) cannot be directly compared, we expect that constructing $X_t$ from $X_0$ and $X_1$ (instead of $X_1$ only) will tend to reduce its variance. Specifically, when $X_t$ is obtained by linear interpolation between $X_0$ and $X_1$, and it does not generate crossing paths, the variance of TD-CFM(C) reduces to the one of $\text{TD}^2$-CFM.

## 5. Experiments

We now present a series of experiments to assess the efficacy of our TD-based flow and diffusion approaches with baselines employing Generative Adversarial Networks (Goodfellow et al., 2014) and $\beta$-Variational Auto-Encoders (Higgins et al., 2017). Following the methodology from Touati et al. (2023); Pirotta et al. (2024), we benchmark 22 tasks spanning 4 domains (Maze, Walker, Cheetah, Quadruped) from the DeepMind Control Suite (Tunyasuvunakool et al., 2020).

---
[3] $m_{0,1}^{(n)}(x_0,x_1|s,a) = m_0(x_0)\delta_{\psi_1^{(n)}(x_0|s,a)}(x_1)$ is the joint distribution of $(X_0, X_1)$, *i.e* the endpoints of the ODE.

For a single policy, we evaluate how well each method models its i) successor measure and ii) value function. While lower errors in estimating the successor measure are expected to lead to better value estimation, this is not always the case since modeling errors may disproportionally affect states with negligible rewards. Additionally, motivated by our theoretical results, we explore how the probability path's design affects our proposed methods' relative performance.

Finally, we examine the scalability of our approach by learning a generative model of the successor measure across a class of parameterized policies derived from the Forward-Backward (FB) representation (Touati & Ollivier, 2021; Touati et al., 2023), a non-generative model of the successor measure. We conclude by demonstrating how $\text{TD}^2$ enables more effective planning for task-relevant policies when performing Generalized Policy Improvement (GPI; Barreto et al., 2017), far surpassing the capabilities of FB alone.

### 5.1. Empirical Evaluation of Geometric Horizon Models

Before benchmarking, we must first obtain a policy to evaluate. We follow the approach taken in Thakoor et al. (2022) and pre-train a set of deterministic policies – one for each task – using TD3 (Fujimoto et al., 2018). The final policy obtained from this pre-training phase is now fixed for the remainder of our experiments. GHM training proceeds in an off-policy manner where we learn the successor measure of a TD3 policy using transition data from the ExoRL dataset (Yarats et al., 2022); specifically, we use a dataset of 10M transitions collected by a random network distillation policy (Burda et al., 2019). All GHM methods are trained for 3M gradient steps using the AdamW optimizer (Loshchilov & Hutter, 2019) with a batch size of 1024 and weight decay of 0.001. We maintain a target network using an exponential moving average of the training parameters with a step size of 0.001. Special care was taken to match the capacity of the neural networks between methods with a UNet-style architecture employed for all flow and diffusion methods, while the GAN and VAE baselines use an MLP with residual connections for all their respective networks. Full details for the training methodology, network architecture, and hyperparameters can be found in Appendix C.

We implement all conditional flow matching methods (TD-CFM, TD-CFM(C), $\text{TD}^2$-CFM) with the Optimal Transport Gaussian conditional path from Lipman et al. (2023). When constructing our bootstrap targets, we sample from the neural ODE using the Midpoint solver with a constant step size of $t/10$ for a maximum of 10 steps. For $\text{TD}^2$-CFM, we sample $t \sim \mathcal{U}([0,1])$; otherwise, we integrate to $t = 1$ and construct $X_t$ using the conditional path. For Denoising Diffusion methods (TD-DD, $\text{TD}^2$-DD), we train a DDPM (Ho et al., 2020) by discretizing $\beta \in (0.1, 20)$ using $T = 1,000$ steps. We construct diffusion bootstrapped targets using

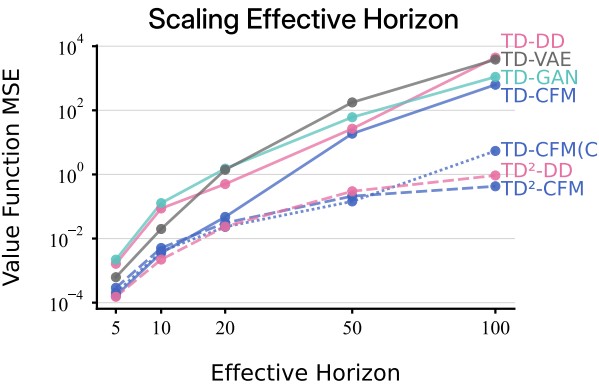

Figure 2. Value-Function prediction error as a function of the effective horizon $(1-\gamma)^{-1}$ for $\gamma \in \{0.8, 0.9, 0.95, 0.98, 0.99\}$ on the POINTMASS loop task. TD$^2$ methods show impressive robustness to increasingly long-horizon predictions.

|  | Method | EMD ↓ | Norm NLL ↓ | MSE(V) ↓ |
|---|---|---|---|---|
| CHEETAH | TD-DD | 20.22 (0.26) | 2.824 (0.195) | 454.49 (131.97) |
|  | TD$^2$-DD | 14.14 (1.08) | 0.806 (0.016) | 189.15 (23.63) |
|  | TD-CFM | 12.26 (0.02) | 0.886 (0.024) | 228.77 (2.20) |
|  | TD-CFM(C) | 10.51 (0.06) | 0.447 (0.020) | 140.78 (18.72) |
|  | TD$^2$-CFM | 10.57 (0.07) | 0.422 (0.014) | 135.22 (19.79) |
|  | GAN | 23.97 (0.46) | — | 2463.22 (628.05) |
|  | VAE | 83.77 (0.41) | — | 1284.27 (37.62) |
| POINTMASS | TD-DD | 0.149 (0.001) | 2.974 (0.100) | 1245.20 (29.27) |
|  | TD$^2$-DD | 0.027 (0.001) | 0.761 (0.082) | 11.13 (3.09) |
|  | TD-CFM | 0.062 (0.003) | 0.554 (0.033) | 355.56 (82.83) |
|  | TD-CFM(C) | 0.022 (0.002) | −0.696 (0.094) | 11.89 (3.16) |
|  | TD$^2$-CFM | 0.021 (0.000) | −0.843 (0.027) | 8.74 (2.09) |
|  | GAN | 0.203 (0.037) | — | 1257.26 (112.86) |
|  | VAE | 0.410 (0.036) | — | 1821.89 (69.78) |
| QUADRUPED | TD-DD | 28.33 (0.33) | 1.908 (0.041) | 1490.75 (444.49) |
|  | TD$^2$-DD | 22.64 (2.47) | 0.861 (0.028) | 159.03 (14.64) |
|  | TD-CFM | 15.73 (0.06) | 1.056 (0.002) | 525.06 (28.90) |
|  | TD-CFM(C) | 14.38 (0.03) | 0.488 (0.003) | 155.25 (5.58) |
|  | TD$^2$-CFM | 14.51 (0.05) | 0.379 (0.011) | 141.77 (3.10) |
|  | GAN | 36772.12 (13898.25) | — | 2634.69 (798.38) |
|  | VAE | 60.27 (0.28) | — | 1156.33 (36.52) |
| WALKER | TD-DD | 20.58 (0.24) | 2.649 (0.137) | 382.40 (458.63) |
|  | TD$^2$-DD | 12.09 (0.12) | 0.537 (0.060) | 39.04 (6.08) |
|  | TD-CFM | 13.53 (0.11) | 0.713 (0.028) | 225.27 (42.43) |
|  | TD-CFM(C) | 11.91 (0.02) | 0.219 (0.016) | 30.71 (3.44) |
|  | TD$^2$-CFM | 11.92 (0.10) | 0.104 (0.001) | 28.35 (6.10) |
|  | GAN | 24.51 (0.89) | — | 3690.65 (1117.94) |
|  | VAE | 111.73 (2.53) | — | 2457.61 (16.25) |

Table 1. Evaluation results comparing our TD-based methods along with GAN and VAE baselines for a single-policy. Results are computed over 19 tasks from 4 domains and further averaged across 3 seeds. For each metric we highlight the best performing methods.

| Method | EMD ↓ | Norm NLL ↓ | MSE(V) ↓ |
|---|---|---|---|
| TD-CFM(C) | 14.08 (12.42) | 1.79 (1.98) | 310.45 (258.94) |
| TD$^2$-CFM | 0.09 (0.09) | −0.01 (0.04) | −3.36 (7.76) |

Table 2. Performance difference for TD-CFM(C) and TD$^2$-CFM when employing a curved instead of straight conditional path. Lower is better with negative values indicating a net improvement for using a curved path.

20 steps of the DDIM (Song et al., 2021a) sampler. For TD-DD, we solve to $t = 0$ and regress towards the noise that re-corrupted our sample. Alternatively, TD$^2$-DD directly regresses towards the noise prediction from the target network at a randomly selected noise level. The first baseline we consider is a GHM instantiated as a Generative Adversarial Network (Goodfellow et al., 2014) similar to the one found in Janner et al. (2020). We follow the best practices from Huang et al. (2024) with the primary modification being a relativistic discriminator (Jolicoeur-Martineau, 2019) equipped with a zero-centered gradient penalty on both real and fake samples. For our second baseline, we implement a $\beta$-VAE (Higgins et al., 2017) following the practices outlined in Thakoor et al. (2022).

To evaluate the quality of our models, we first generate samples from the ground truth successor measure $m^\pi$ according to the following procedure. We first randomly sample 64 source states $S_0$ from the initial state distribution and execute policy $\pi$ for $1,000$ steps. Along each trajectory, we resample 2048 states with replacement according to the stopping time $t \sim \text{Geometric}(1-\gamma)$. For the same 64 source states, we generate a matching set of 2048 samples from each GHM. Now in possession of these two sets of samples, we evaluate the: 1) log-likelihood of the true samples for models with tractable densities (i.e., diffusion and flow methods); 2) Earth Mover's Distance (EMD; Rubner et al., 2000), which quantifies the minimal transport cost between the two empirical distributions; and 3) mean-squared error of a Monte-Carlo estimate of the true value function $Q^\pi$ and the value function derived from GHM samples using (2). Full details can be found in Appendix C.1.

Having established our training framework, baselines, and evaluation protocol, we proceed to investigate a key prediction from our theoretical analysis. Our variance analysis

suggests that our TD-Flow framework should enable more stable training across extended temporal horizons. To validate this hypothesis, we train each GHM for 3 seeds on the loop task in the Maze domain while varying the effective horizon $(1-\gamma)^{-1}$ across five values: $\{5, 10, 20, 50, 100\}$. Figure 2 illustrates the relationship between value function MSE and the effective horizon. The results demonstrate that TD$^2$-based methods maintain consistent performance even as the effective horizon increases, while alternative approaches show significant performance degradation. Notably, at an effective horizon of 100, TD$^2$-based methods maintain their accuracy and achieve performance improvements of nearly four orders of magnitude compared to their naive implementations. These results empirically support for our initial hypothesis, with the stability of TD$^2$ methods aligning with our predictions.

In the following, we shift our attention to a more in-depth analysis of the largest horizon of 100 ($\gamma = 0.99$). For each

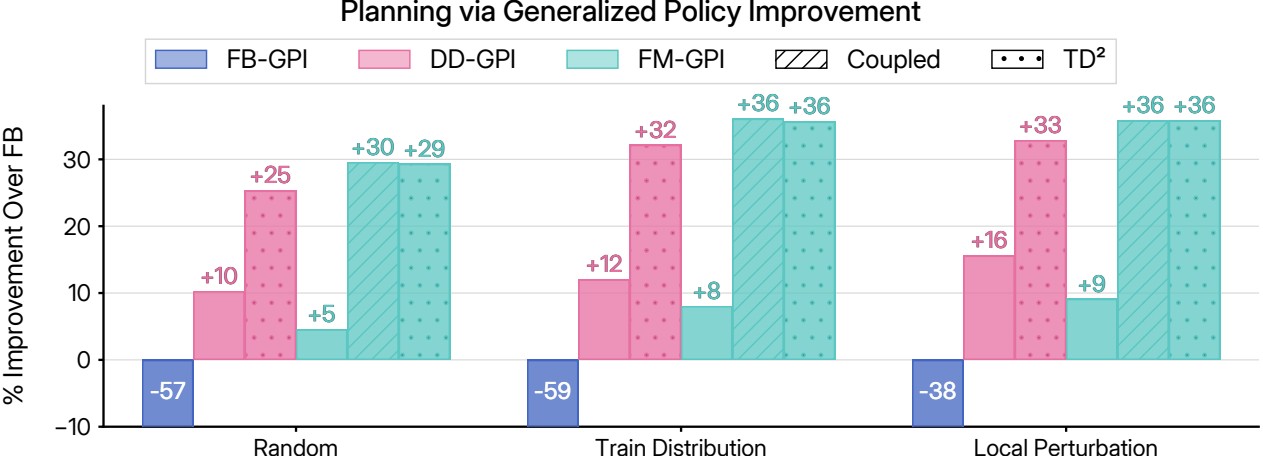

*Figure 3.* Performance improvement over the zero-shot Forward Backward (FB; Touati & Ollivier, 2021) policies when planning with Generalized Policy Improvement (GPI; Barreto et al., 2017). FB-GPI performs GPI over the FB value-function $Q^{\pi_w}$. DD-GPI and FM-GPI perform GPI with the value function implied by the GHM $m^{\pi_w}$ for our diffusion-based and flow-based methods, respectively. Results are averaged over 22 tasks across 4 domains.

algorithm, we train a GHM for 3 independent seeds for all domains and tasks. Table 1 reports aggregate performance across our full suite of metrics. For each domain and metric, we highlight results in a $1\%$ range with respect to the best-performing method. The results demonstrate a clear pattern of superior performance for TD$^2$-based algorithms: TD$^2$-CFM achieves significant improvements over TD-CFM with a $10\times$ reduction in value-function MSE, $1.5\times$ reduction in EMD, and $3\times$ reduction in log-likelihood, averaged across all four domains. In line with our theoretical predictions, the coupled variant of TD-CFM performs comparably to TD$^2$-CFM, given straight conditional paths. While a comparison between flow matching and diffusion is not at the core of this paper, in our experiments, flow matching generally outperforms diffusion across all metrics. We posit this is primarily due to noise in the diffusion process adversely impacting an already noisy prediction problem for large horizons.

Given the comparable performance between TD-CFM(C) and TD$^2$-CFM with straight conditional paths, we next examine how these methods behave with alternative path geometries. Our theoretical analysis suggests an important distinction: TD$^2$-CFM should maintain its effectiveness with non-straight paths, while the performance of TD-CFM(C) should degrade. To test this prediction, we maintain the methodology above while replacing conditional path in (TD$^2$-CFM; 9) with the following curved path $p_{t|1}(\cdot \mid X_1) = \mathcal{N}(\cdot \mid \alpha_t X_1, \beta_t^2)$ with coefficients $\alpha_t = \sin\left(\frac{\pi}{2}t\right)$ and $\beta_t = \cos\left(\frac{\pi}{2}t\right)$. The corresponding conditional vector field is now given by $u_{t|1}(X_t|X_1) = \left(\dot{\alpha}_t - \frac{\alpha_t}{\beta_t}\right)X_1 + \frac{\dot{\beta}_t}{\beta_t}X_t$. Additionally, for TD-CFM(C) we condition the curved path above on $X_0$ and $X_1$ resulting in the conditional vector field

$u_{t|0,1}(X_t \mid X_0, X_1) = \frac{\pi}{2}\left(\beta_t X_1 - \alpha_t X_0\right)$. Table 2 illustrates the performance difference relative to the straight path results (Table 1) averaged across all domains and tasks. The results strongly support our theoretical prediction: TD$^2$-CFM not only maintained but surprisingly improved performance compared to the linear path. In contrast, TD-CFM(C) showed significant performance degradation, confirming our hypothesis about its limitations with non-straight paths.

## 5.2. Planning via Generalized Policy Improvement

We now turn our attention towards training policy-conditioned GHMs which can be utilized for test-time planning. To accomplish this, we first pre-train a Forward Backward (FB; Touati & Ollivier, 2021; Touati et al., 2023) representation using the same dataset of 10M transitions as described in §5.1. This pre-training yields a class of $w$-conditioned policies $\pi_w$, where each $w \in \mathsf{W} = \mathsf{S}^{d-1}(\sqrt{d})$ represents an embedding of a reward function situated on a $d$-dimensional hypersphere with radius $\sqrt{d}$. We then train the GHM $m^{\pi_w}$ conditioned on the policy by incorporating the embedding $w$ directly into the model's input. All GHM methods are trained for 8M gradient steps, maintaining the same parameters used in §5.1, with the exception of a higher weight decay coefficient of $0.01$. For additional insights into the accuracy of the policy-conditioned GHMs, we direct the reader to Appendix D. Overall, we observed similar trends to those seen in our single-policy experiments.

Given that both FB and $w$-conditioned GHM models enable estimation of a policy's value function $Q^{\pi_w}$, we can utilize this information to perform Generalized Policy Improvement (GPI; Barreto et al., 2017) during evaluation. Specifi-

cally, at each time step $t$, we choose an action $a_t = \pi_{w_t}(s_t)$, where $w_t$ is derived as follows:

$$w_t \in \underset{w \sim D(\mathsf{W})}{\arg\max} \underbrace{(1-\gamma)^{-1} \mathbb{E}_{X \sim m^{\pi_w}(\cdot|s_t, \pi_w(s_t)))}\big[r(X)\big]}_{Q^{\pi_w}(s_t, \pi_w(s_t))}.$$

(15)

Here $D(\mathsf{W})$ is a sampling distribution over $\mathsf{W}$. We consider three such distributions: *i) Random*: uniform distribution over $\mathsf{W}$; *ii) Local Perturbation*: we perturb the embedding $w_r$ of the task reward $r$ by the uniform distribution; *iii) Train Distribution*: we sample $w$ from the training distribution used by FB. To approximate (15), we sample 255 embeddings from $D(\mathsf{W})$ and explicitly include the task embedding $w_r$, resulting in a maximization over 256 policies. To estimate $Q^{\pi_w}$, we average the reward over 128 states sampled from $m^{\pi_w}$. Performance is measured by averaging returns over 100 episodes, each lasting 1000 steps.

Figure 3 illustrates the average percentage of improvement for each algorithm and $w$-sampling strategy relative to the performance of the FB policy $\pi_{w_r}$ for the task reward $r$. We refer to Appendix D for a more detailed view of these results. All TD-based GHM approaches lead to a significant improvement over the base FB policy, with TD-CFM(C) and TD$^2$-CFM providing $\approx 30\%+$ improvement with all sampling approaches. TD$^2$-DD also leads to significant performance gains but is still dominated by the flow matching methods. Notably, FB-based GPI not only fails to improve performance but actually deteriorates it on average with significant degradation observed in three out of four domains (detailed results available in Appendix D). When comparing different distributions $D(\mathsf{W})$, we observe that while FB-GPI's performance fluctuates considerably, GHM methods maintain their robustness across distributions, showing only minor variation. These results underscore the ability of our improved GHMs to make long-term predictions enabling powerful planning capabilities.

## 6. Discussion

In this paper, we introduced temporal difference flows, a novel generative modeling approach that significantly advances long-horizon predictive models of state. By leveraging the successor measure's temporal difference structure both in its sampling procedure and learning objective, TD$^2$-CFM and TD$^2$-DD effectively address challenges associated with modeling long-range state dynamics. The methods developed in this paper provide a robust theoretical and empirical foundation that demonstrates the advantages of our framework across a range of tasks, metrics, and domains. We envision numerous exciting applications emerging from this work, particularly around imitation learning (Wu et al., 2025; Jain et al., 2025), planning (Sutton, 1991; Thakoor et al., 2022; Zhu et al., 2024), and off-policy evaluation (Precup et al., 2000; 2001; Nachum et al., 2019; Fujimoto et al.,

2021). Furthermore, recent work on consistency models (Song et al., 2023; Yang et al., 2024) and self-distillation (Frans et al., 2025) suggests promising avenues for tackling the computational burden of sampling — a limitation common to the family of iterative generative models that our approach builds upon.

## Impact Statement

This paper presents work whose goal is to advance the field of Machine Learning. There are many potential societal consequences of our work, none of which we feel must be specifically highlighted here.

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

# Appendices

## A. Related Work

The Successor Representation (Dayan, 1993) was originally proposed for tabular MDPs and was later generalized to continuous state spaces with the Successor Measure (Blier et al., 2021). Successor Features (Barreto et al., 2017; 2020) extends these ideas by instead modeling the evolution of multi-dimensional features assuming rewards decompose linearly over these features. Prior works have leveraged these methods for zero-shot policy evaluation (Dayan, 1993; Barreto et al., 2017; Wiltzer et al., 2024b), zero-shot policy optimization (Borsa et al., 2019; Touati & Ollivier, 2021; Touati et al., 2023; Park et al., 2024; Zhu et al., 2024; Cetin et al., 2024; Tirinzoni et al., 2025), imitation learning (Pirotta et al., 2024; Jain et al., 2025), exploration (Machado et al., 2020; Jain et al., 2023), representation learning (Le Lan et al., 2022; 2023a;b; Farebrother et al., 2023; Ghosh et al., 2023), and building temporal abstractions (Machado et al., 2018; 2023).

Janner et al. (2020) originally proposed a method to learn a generative model of the successor measure with modeling techniques spanning from Generative Adversarial Networks (Goodfellow et al., 2014) to Normalizing Flows (Dinh et al., 2015; Rezende & Mohamed, 2015) like RealNVP (Dinh et al., 2017). Followup work (e.g., Thakoor et al., 2022; Tomar et al., 2024) explored other generative modeling techniques including various types of auto-encoders (e.g., Higgins et al., 2017; van den Oord et al., 2017). Also of note is recent work learning generative models of multi-dimensional cumulants including features (Wiltzer et al., 2024a; Zhu et al., 2024) and multi-variate reward functions (Zhang et al., 2021). Prior work by Wiltzer et al. (2024b) sought to deal with the instability of long-horizon predictions in GHMs by employing an $n$-step mixture distribution where they sample $t \sim \text{Geometric}(1-\gamma)$ and bootstrap if $t > n$; otherwise returning the state at time $t$ along the trajectory. Without resorting to importance sampling this approach is limited to the on-policy setting. Finally, most closely related to our work is that of Schramm & Boularias (2024) who provide a preliminary and limited derivation of what we term $\text{TD}^2\text{-DD}$. In contrast, our work not only rigorously formalizes and significantly extends these ideas but also integrates them into the more general flow-matching framework (Lipman et al., 2023; 2024), additionally incorporating extensions to score-matching (Song et al., 2021b;b) and diffusion (Sohl-Dickstein et al., 2015; Ho et al., 2020). Moreover, we conduct an extensive empirical analysis, demonstrating the efficacy of our approach — an aspect notably absent from Schramm & Boularias (2024).

## B. Extension to Score Matching and Diffusion Models

This section extends our framework to score matching and denoising diffusion models. We leverage the unification of these methods under stochastic differential equations (Song et al., 2021b) introducing an analogous class of Temporal Difference Diffusion methods.

### B.1. Background

Both score-based generative modeling (Song & Ermon, 2019) and diffusion probabilistic modeling (Sohl-Dickstein et al., 2015; Ho et al., 2020) can be unified under the framework of stochastic differential equations (SDE) introduced in Song et al. (2021b). Unlike in flow-matching, time is inverted in diffusion models and ranges from time 0 to $T$. Given the data distribution $q_0$ and prior simple distribution $q_T$ (the "noise" distribution), we construct a diffusion process $\{X_t\}_{t \in [0,T]}$ such that $X_0 \sim q_0$ and $X_T \sim q_T$. This diffusion can be modeled as the solution to an Ito SDE:

$$\mathrm{d}X_t = f(t)\, X_t\, \mathrm{d}t + g(t)\, \mathrm{d}W_t \ \mid\ X_0 \sim q_0 \,, \tag{16}$$

where $W_t$ is a standard Brownian motion and $f : [0, T] \to \mathbb{R}^d$ is scalar function called the drift coefficient, and $g : [0, T] \to \mathbb{R}$ is scalar function known as diffusion coefficient.

Generating samples from $X_0 \sim q_0$ consists in sampling $X_T \sim q_T$ and reversing the forward-SDE process in (16). A known result from Anderson (1982) states that the reverse of a diffusion process is also a diffusion process, running backward in time and given by the reverse-time SDE:

$$\mathrm{d}X_t = \Big( f(t)\, X_t - g(t)^2\, \nabla_{X_t} \log q_t(X_t) \Big) \mathrm{d}t + g(t)\, \mathrm{d}\overline{W}_t \ \mid\ X_T \sim q_T \,, \tag{17}$$

where $\overline{W}_t$ is a Brownian motion when time flows backwards from $T$ to 0, $\mathrm{d}t$ is an infinitesimal negative timestep and $q_t$ is

the marginal distribution of $X_t$. Therefore, once we learn the score of the marginal distribution $\nabla_x \log q_t(x)$, we can sample from $q_0$ by simulating the reverse diffusion process (17).

To estimate $\nabla_x \log q_t(x)$, we can train a time-dependent score-based model $\tilde{s}(\cdots; \theta) : [0, T] \times \mathbb{R}^d \to \mathbb{R}^d$ via the denoising diffusion / score matching objective (Vincent, 2011; Song & Ermon, 2019):

$$\ell_{\text{DD}}(\theta) = \mathbb{E}_{t \sim \mathcal{U}([0,1]), X_0 \sim q_0} \mathbb{E}_{X_t \sim q_{t|0}(\cdot | X_0)} \left[ \left\| \tilde{s}_t(X_t; \theta) - \nabla_{X_t} \log q_{t|0}(X_t \mid X_0) \right\|^2 \right]. \tag{18}$$

For $\ell_{\text{DD}}$ to be tractable, we need to know the conditional probability $q_{t|0}$. Usually, specific choices of the drift and diffusion coefficients $f_t$ and $g_t$ are used such that $q_{t|0}$ is always a Gaussian distribution $\mathcal{N}(\cdot \mid \alpha_t x_0, \sigma_t^2)$, where the mean $\alpha_t$ and variance $\sigma_t^2$ can be computed in closed-form. The global minimizer of $\ell_{\text{DD}}(\theta)$ denoted by $s_t^\star(x)$ is equal to the score function $\nabla_x \log q_t(x)$, thanks to the following proposition:

**Proposition 2** (Vincent 2011). *Let $q_t(x) = \int q_0(x_0) q_{t|0}(x|x_0) \, dx_0$, then we have:*

$$\nabla_\theta \ell_{\text{DD}}(\theta) = \nabla_\theta \mathbb{E}_{t, X_t \sim q_t} \left[ \left\| \tilde{s}_t(X_t; \theta) - \nabla_{X_t} \log q_t(X_t) \right\|^2 \right]. \tag{19}$$

## B.2. Temporal Difference Diffusion

To learn a predictive model of $m^\pi$ using diffusion from an offline dataset, we follow a similar approach to what we presented in §3 and we define an iterative process starting from initial weights $\theta^{(0)}$ and at each iteration minimizing the Temporal-Difference Denoising Diffusion (TD-DD) loss:

$$\ell_{\text{TD-DD}}(\theta) = \mathbb{E}_{\rho, t, X_0, X_t} \left[ \left\| \tilde{s}_t(X_t \mid S, A; \theta) - \nabla_x \log q_{t|0}(X_t \mid X_1) \right\|^2 \right],$$
$$\text{where}, \ X_0 \sim \left( \mathcal{T}^\pi \widetilde{m}_{0|T}^{(n)} \right) (\cdot \mid S, A), \ X_t \sim q_{t|0}(\cdot \mid X_0). \tag{TD-DD; 20}$$

In order to sample $X_0 \sim \left( \mathcal{T}^\pi \widetilde{m}_{0|T}^{(n)} \right) (\cdot \mid s, a)$, with probability $1 - \gamma$, we return the successor state $S' \sim P(\cdot \mid S, A)$. Otherwise, with probability $\gamma$ we solve the following reverse-time SDE from $X_T$ using the score $\tilde{s}_t^{(n)}$,

$$dX_t = \left( f(t) X_t - g(t)^2 \tilde{s}_t^{(n)}(X_t \mid S, A) \right) dt + g(t) d\overline{W}_t. \tag{21}$$

Minimizing $\ell_{\text{TD-DD}}(\theta)$ leads to score function $\tilde{s}_t^{(n+1)}(s \mid s, a)$ generating a marginal probability $q_t^{(n+1)}$ that approximates $\mathcal{T}^\pi q_0^{(n)}$ at $t = 0$.

Following the $\text{TD}^2$-CFM blueprint, we can further exploit the structure of the target bootstrapped distribution to design an improved diffusion process that converts Gaussian noise to $\mathcal{T}^\pi q_0^{(n)}$. First, we show below that the mixture of a diffusion process is also a diffusion process with modified drift and diffusion functions.

**Lemma 2.** *Consider two diffusion processes with drift functions $\vec{f}$ and $\widehat{f}$, sharing the same diffusion coefficient $g$:*

$$dX_t = \vec{f}_t(X_t) \, dt + g(t) \, dW$$
$$dX_t = \widehat{f}_t(X_t) \, dt + g(t) \, dW.$$

*Let $\vec{q}_t$ and $\widehat{q}_t$ be their marginal distribution, then the diffusion process corresponding to the mixture marginal distribution $q_t = (1 - \gamma)\vec{q}_t + \gamma\widehat{q}_t$ is:*

$$dX_t = \frac{(1 - \gamma)\vec{q}_t \vec{f}_t + \gamma\widehat{q}_t \widehat{f}_t}{(1 - \gamma)\vec{q}_t + \gamma\widehat{q}_t} (X_t) \, dt + g(t) \, dW.$$

*Proof.* The marginal probabilities $\vec{p}$ and $\widehat{p}$ are characterized by the Fokker-Planck equations:

$$\frac{\partial \vec{p}_t}{\partial t} = -\text{div}(\vec{p}_t \vec{f}_t) + \frac{g_t^2}{2} \Delta \vec{p}_t$$
$$\frac{\partial \widehat{p}_t}{\partial t} = -\text{div}(\widehat{p}_t \widehat{f}_t) + \frac{g_t^2}{2} \Delta \widehat{p}_t$$

where $\mathrm{div}$ is the divergence operator and $\Delta = \mathrm{div}\nabla$ is the Laplace operator. Therefore,

$$\frac{\partial p_t}{\partial t} = (1-\gamma)\frac{\partial \vec{p}_t}{\partial t} + \gamma\frac{\partial \widehat{p}_t}{\partial t}$$

$$= -\mathrm{div}(\vec{p}_t\vec{f}_t) + \frac{g_t^2}{2}\Delta\vec{p}_t - \mathrm{div}(\widehat{p}_t\widehat{f}_t) + \frac{g_t^2}{2}\Delta\widehat{p}_t$$

$$= -\mathrm{div}\left((1-\gamma)\vec{p}_t\vec{f}_t + \gamma\widehat{p}_t\widehat{f}_t\right) + \frac{g_t^2}{2}\Delta\left((1-\gamma)\vec{p}_t + \gamma\widehat{p}_t\right)$$

$$= \mathrm{div}\left(p_t\frac{(1-\gamma)\vec{p}_t\vec{f}_t + \gamma\widehat{p}_t\widehat{f}_t)}{(1-\gamma)\vec{p}_t + \gamma\widehat{p}_t}\right) + \frac{g_t^2}{2}\Delta p_t.$$

The drift $\frac{(1-\gamma)\vec{p}_t\vec{f}_t + \gamma\widehat{p}_t\widehat{f}_t}{(1-\gamma)\vec{p}_t + \gamma\widehat{p}_t}$ and the diffusion coefficient $g_t$ satisfy the Fokker-Planck equation with the probability path $p_t$, and therefore their associated diffusion process generate $p_t$. $\qquad\square$

Lemma 2 can be easily extended to the case of a continuous mixture of diffusion processes.

This result shows that it is possible to use two independent diffusion processes for the two terms in the sampling process induced by the Bellman operator. For the first, we can use the standard noising diffusion process:

$$\vec{q}_t(x \mid s, a) = \int q_{t|0}(x \mid s')P(\mathrm{d}s' \mid s, a),$$

where we sample $X_t \sim q_{t|0}(\cdot \mid s')$ by simulating a simple forward diffusion process (16). For the second term, we can leverage the GHM $m_t^{(n)}$ at the previous iteration to construct the process,

$$\widehat{q}_t^{(n)}(x \mid s, a) = \int m_t^{(n)}(x \mid s', \pi(s'))\,P(\mathrm{d}s' \mid s, a),$$

where $m_t^{(n)}(x \mid s', a')$ is the marginal probability of the reverse SDE induced by the score $s^{(n)}$,

$$\mathrm{d}X_t = \left(f(t)\,X_t - g(t)^2\,s_t^{(n)}(X_t \mid s, a)\right)\mathrm{d}t + g(t)\,\mathrm{d}\overline{W}_t.$$

Additionally, $\widehat{q}_t^{(n)}(x \mid s, a)$, as continuous mixture of diffusion's marginals $m_t^{(n)}(x \mid s', \pi(s'))$ weighted by $P(s' \mid s, a)$, can be generated by the diffusion process,

$$\mathrm{d}X_t = \left(f(t)\,X_t - g(t)^2\,\widehat{s}_t(X_t \mid s, a)\right)\mathrm{d}t + g(t)\,\mathrm{d}\overline{W}_t, \text{ where}$$

$$\widehat{s}_t(x_t \mid s, a) = \frac{\int P(\mathrm{d}s' \mid s, a)\,q_t^{(n)}(x \mid s', \pi(s'))\,s_t^{(n)}(x_t \mid s', \pi(s'))}{\int P(\mathrm{d}s' \mid s, a)\,q_t^{(n)}(x \mid s', \pi(s'))}.$$

Given these two diffusion processes, the target probability $q_t^{(n+1)} = (1-\gamma)\vec{q}_t + \gamma\widehat{q}_t^{(n)}$ can be generated by the following reverse SDE,

$$\mathrm{d}X_t = \left(f(t)X_t - g(t)^2\,s_t^{(n+1)}(X_t \mid s, a)\right)\mathrm{d}t + g(t)\,\mathrm{d}\overline{W}_t,$$

where $s_t^{(n+1)}(x \mid s, a) = \frac{(1-\gamma)\vec{q}_t\nabla_x\log\vec{q}_t + \gamma\widehat{q}_t^{(n)}\widehat{s}_t^{(n)}}{(1-\gamma)\vec{q}_t + \gamma\widehat{q}_t^{(n)}}(x \mid s, a)$. Therefore, we can learn $\tilde{s}_t(\cdots; \theta)$ to approximate $s_t^{(n+1)}$ by minimizing the loss,

$$\ell(\theta) = (1-\gamma)\mathbb{E}_{\rho, t, X_t \sim \vec{q}_t(\cdot \mid S, A)}\left[\left\|\tilde{s}(X_t \mid S, A; \theta) - \nabla_{X_t}\log\vec{q}_t(X_t \mid S, A)\right\|^2\right] \tag{22}$$

$$+ \gamma\mathbb{E}_{\rho, t, X_t \sim \widehat{q}_t^{(n)}(\cdot \mid S, A)}\left[\left\|\tilde{s}(X_t \mid S, A; \theta) - \widehat{s}_t^{(n)}(X_t \mid S, A)\right\|^2\right].$$

We can simplify the first term via Proposition 2 (since $\vec{q}_t(x|s,a) = \int q_{t|0}(x|s')P(\mathrm{d}s'|s,a)$), hence we have

$$\nabla_\theta \, \mathbb{E}_{\rho,t,X_t \sim \vec{q}_t(\cdot|s,a)} \left[ \left\| \tilde{s}(X_t \mid s,a;\theta) - \nabla_{X_t} \log \vec{q}_t(X_t \mid S,A) \right\|^2 \right] =$$

$$\nabla_\theta \, \mathbb{E}_{\rho,t,X_t \sim q_{t|0}(\cdot|S')} \left[ \left\| \tilde{s}(X_t \mid S,A;\theta) - \nabla_{X_t} \log q_{t|0}(X_t \mid S') \right\|^2 \right].$$

Moreover, using a similar argument for equivalence between the gradient of marginal and conditional flow-matching objectives, we can show that

$$\nabla_\theta \, \mathbb{E}_{\rho,t,X_t \sim \hat{q}_t^{(n)}(\cdot|S,A)} \left[ \left\| \tilde{s}(X_t \mid S,A;\theta) - \hat{s}_t^{(n)}(X_t \mid S,A) \right\|^2 \right] =$$

$$\nabla_\theta \, \mathbb{E}_{\rho,t,X_T \sim q_T, X_t \sim q_{t|T}^n(\cdot|s,a)} \left[ \left\| \tilde{s}(X_t \mid S,A;\theta) - s_t^{(n)}(X_t \mid S,A) \right\|^2 \right].$$

This leads us to the final $\mathrm{TD}^2$-DD loss function,

$$\ell_{\mathrm{TD}^2\text{-DD}}(\theta) = (1-\gamma)\mathbb{E}_{\rho,t,X_t \sim q_{t|0}(\cdot|S')} \left[ \left\| \tilde{s}_t(X_t \mid S,A;\theta) - \nabla_x \log p_{t|0}(X_t \mid S') \right\|^2 \right] \tag{23}$$

$$+ \gamma \mathbb{E}_{\rho,t,X_t \sim q_{t|T}^{(n)}(\cdot|S',\pi(S'))} \left[ \left\| \tilde{s}(X_t \mid S,A;\theta) - \tilde{s}_t^{(n)}(X_t \mid S',\pi(S')) \right\|^2 \right].$$

# C. Experimental Details

## C.1. Evaluation

Evaluating a GHM can be challenging, TD-based losses employing bootstrapping do not provide a good signal as to the quality of the learned model. Instead, we opt to measure 1) the likelihood of a trajectory coming from the true discounted occupancy of a given policy, 2) the Earth Mover's Distance (EMD; Rubner et al., 2000) between samples from the true occupancy and our GHM which provides an estimate of the distance between these two probability distributions, and 3) the value-function approximation error. In all cases, to obtain samples from the true discounted occupancy, we collect trajectories $\{(s_0, s_1, \ldots, s_T)\}_{i=1}^N$ from policy $\pi$ and subsequently re-sample states according to $t \sim \text{Geometic}(1 - \gamma)$ for a particular discount factor $\gamma \in [0, 1)$. Armed with samples from $m^\pi$ we compute the aforementioned metrics following the procedures stated below along with the parameter values outlined in Table 3.

*Table 3.* Evaluation hyper-parameters for both single and multi-policy experiments.

| Evaluation | Hyperparameter | Value |
|---|---|---|
| EMD | Number of states $s_0$ | 64 |
| | Number of $m$-samples per state | 2048 |
| | Number of episodes per state | 1 |
| | Episode length | 1000 |
| MSE(V) | Number of state $s_0$ | 64 |
| | Number of GHM-samples per state | 2048 |
| | Number of episodes per state | 1 |
| | Episode length | 1000 |
| GPI | Number of $z$ samples | 256 |
| | Number of GHM samples | 128 |
| | Number of FB inference samples | $250,000$ |

**Normalized Negative Log-Likelihood.** To compute the log-likelihood of our flow matching and diffusion methods, we take advantage of the following change in variables formula (Dinh et al., 2015; Rezende & Mohamed, 2015; Chen et al., 2018),

$$\log\left(\widetilde{m}(x_1 \mid s, a; \theta)\right) = \log \varphi(x_0) + \int_0^1 \frac{\partial \log\left(\widetilde{m}(x_t \mid s, a; \theta)\right)}{\partial x_t} \, dt \, ,$$

where $\varphi$ is the probability density function of a standard Gaussian distribution, which acts as the prior on $x_0$. The change in log density over time can be written as the following differential equation called the instantaneous change of variables formula (Chen et al., 2018, Theorem 1),

$$\frac{\partial \log\left(\widetilde{m}(x_t \mid s, a; \theta)\right)}{\partial x_t} = -\text{Tr}\left(\frac{\partial \tilde{v}_t(x_t \mid s, a; \theta)}{\partial x_t}\right) \, .$$

We can now compute the log-likelihood for a sample $X \sim m^\pi(\cdot \mid s, a)$ by integrating the total change in log-density backward in time from $x_1 = X$ to obtain $x_0$ which has tractable likelihood. In practice, we solve the following coupled initial value problem using numerical integration (Grathwohl et al., 2019),

$$\begin{bmatrix} x_0 \\ \log \widetilde{m}(x_1 \mid s, a; \theta) - \log \varphi(x_0) \end{bmatrix} = \int_1^0 \begin{bmatrix} -\tilde{v}_t(x_t \mid s, a; \theta) \\ \text{Tr}\left(\frac{\partial \tilde{v}_t(x_t \mid s, a; \theta)}{\partial x_t}\right) \end{bmatrix} dt \, ,$$

$$\text{where} \begin{bmatrix} x_1 \\ \log \widetilde{m}(x \mid s, a; \theta) - \log \widetilde{m}(x_1 \mid s, a; \theta) \end{bmatrix} = \begin{bmatrix} X \\ 0 \end{bmatrix} \, .$$

(24)

For all experiments we report the negative log-likelihood *normalized by the dimension of the observation space*.

**Earth Mover's Distance** We compute the Earth Mover's Distance (EMD; Rubner et al., 2000), also known as the Wasserstein-1 distance, between $m = 2048$ samples from the ground truth distribution $X \sim m^\pi(\cdot \mid S_k, A_k)$ and our learned GHM $\widetilde{X} \sim \widetilde{m}(\cdot \mid S_k, A_k; \theta)$ for a set of randomly sampled state-action pairs $\{(S_k, A_k)\}_{k=1}^n$. Intuitively, the EMD quantifies the minimum cost required to transform one distribution into another, where the cost is defined in terms of the Euclidean distance between states $X^{(i)}, X^{(j)}$. Formally, we have,

$$\text{EMD}(\{X^{(1)}, \ldots, X^{(m)}\}, \{\widetilde{X}^{(1)}, \ldots, \widetilde{X}^{(m)}\}) = \min_{\xi \in \Xi} \sum_{i,j} \xi_{ij} \sum_{k=1}^d \left(X_k^{(i)} - \widetilde{X}_k^{(j)}\right)^2 \, ,$$

where $\xi$ is a transport plan such that $\xi_{ij}$ specifies the proportion of mass moved from $X_i$ to $\widetilde{X}_j$. We report the average EMD across $n = 64$ source states using the Python Optimal Transport (Flamary et al., 2021) library.

**Value Function Mean Square Error (MSE(V)).** We compute the mean square error between a Monte-Carlo estimation $\widetilde{V}_{\mathrm{MC}}^{\pi}$ of the value function $V^{\pi}(s)$ and the estimation $\widetilde{V}_{\mathrm{GHM}}$ obtained using the learned model. We obtain $\widetilde{V}_{\mathrm{MC}}^{\pi}$ by collecting a trajectory $\{(s_0, s_1, \ldots, s_T)\}$ from policy $\pi$ and computing the discounted sum of rewards. We generate a single trajectory since both the policy and the environment are deterministic. The GHM estimate is given by (2), i.e.,

$$\widetilde{V}_{\mathrm{GHM}}^{\pi}(s) = (1-\gamma)^{-1} \mathbb{E}_{\widetilde{X} \sim \widetilde{m}(\cdot | s, \pi(s))} \left[ r(\widetilde{X}) \right].$$

Then, $\mathrm{MSE}(\widetilde{V}_{\mathrm{MC}}^{\pi}, \widetilde{V}_{\mathrm{GHM}}^{\pi}) = \mathbb{E}_{S_0 \sim \nu} \left[ (\widetilde{V}_{\mathrm{GHM}}^{\pi}(S_0) - \widetilde{V}_{\mathrm{MC}}^{\pi}(S_0))^2 \right]$. We average our results over 64 initial states $S_0$ sampled from the initial state distribution $\nu$.

**Planning with GPI.** We evaluate planning performance by computing the average return over 100 episodes, each lasting $1,000$ steps, for every task. For the Forward-Backward representation (Touati & Ollivier, 2021), we directly follow the policy $\pi_{w_r}$ (thus $a_t = \pi_{w_r}(s_t)$) where $w_r = \mathbb{E}_{(S,R) \sim \rho}[B(s) \cdot R]$ is the zero-shot policy embedding inferred using $250,000$ transitions labeled with the task reward function $r$. Given that FB provides a direct way of estimating the value function of a policy (i.e., $Q_r^{\pi_w}(s, a) = F(s, a, w)^T z_r$), we can do planning in the policy embedding space by solving the following problem:

$$w_t^{\text{FB-GPI}} \in \arg\max_{w \sim D(\mathsf{W})} F(s_t, \pi_w(s_t), w)^T w_r.$$

This optimization problem requires no generation except sampling from $D(\mathsf{W})$. We approximate the max using 255 samples from $D(\mathsf{W})$ and additionally incorporating $w_r$ to ultimately maximize over 256 policies. On the other hand, for GHM-GPI, we solve the following optimization problem,

$$w_t^{\text{GHM-GPI}} \in \arg\max_{w \sim D(\mathsf{W})} \underbrace{(1-\gamma)^{-1} \mathbb{E}_{X \sim m^{\pi_w}(\cdot | s_t, \pi_w(s_t)))} \left[ r(X) \right]}_{Q^{\pi_w}(s_t, \pi_w(s_t))},$$

which requires generating samples from $m^{\pi_w}$. In our experiments we generate 128 samples from $m^{\pi_w}$.

### C.2. Environments

Experiments in this paper were conducted with a subset of domains from the DeepMind Control Suite (Tunyasuvunakool et al., 2020) highlighted in Figure 4.

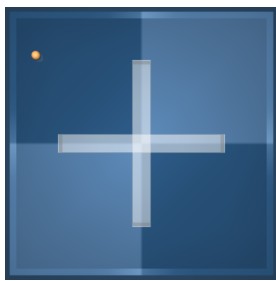 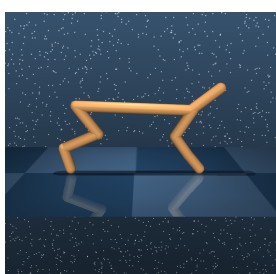 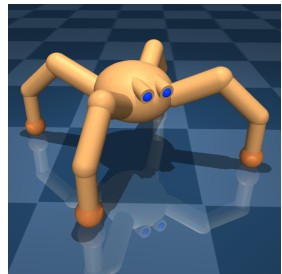 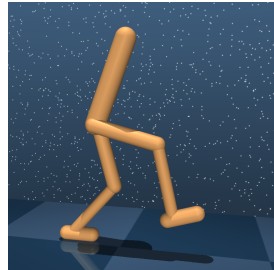

*Figure 4.* A visual depiction of each domain used in our experiments from the DeepMind Control Suite (Tunyasuvunakool et al., 2020). From left to right: MAZE, CHEETAH, QUADRUPED, WALKER.

## C.3. Geometric Horizon Models

This section describes each class of generative model used for our empirical experiments.

### C.3.1. FLOW MATCHING

**Algorithm 1** Template for TD-Flow algorithms

1: **Inputs**: offline dataset $\mathcal{D}$, policy $\pi$, batch size $n$, Polyak coefficient $\zeta$, weight decay $\lambda$, randomly initialized weights $\theta$, discount factor $\gamma$, learning rate $\eta$, one-step conditional path $\vec{p}_{t|1}$ and conditional vector-field $\vec{u}_{t|1}$, bootstrap path $\widehat{p}_t$ and vector-field $\widehat{v}_t$.
2: **for** $n = 1, \ldots$ **do**
3:     Sample mini-batch $\{(S_k, A_k, S'_k)\}_{k=1}^K$ from $\mathcal{D}$
4:     **for** $k = 1, \ldots, K$ **do**
5:         Sample $t_k \sim \mathcal{U}([0,1])$
6:         Sample $\vec{X}_k \sim \vec{p}_{t_k|1}(\cdot \mid S'_k)$
7:         $\vec{\ell}_k(\theta) = \left\| v_{t_k}(\vec{X}_k \mid S_k, A_k; \theta) - \vec{u}_{t_k|1}(\vec{X}_k \mid S'_k) \right\|^2$
8:         Sample $\widehat{X}_k \sim \widehat{p}_{t_k}(\cdot \mid S'_k, \pi(S'_k); \bar{\theta})$
9:         $\widehat{\ell}_k(\theta) = \left\| v_{t_k}(\widehat{X}_k \mid S_k, A_k; \theta) - \widehat{v}_{t_k}(\widehat{X}_k \mid S'_k, \pi(S'_k); \bar{\theta}) \right\|^2$
10:     **end for**
11:     `#Compute loss`
12:     $\ell(\theta) = \frac{1}{K} \sum_{k=1}^K (1-\gamma)\vec{\ell}_k(\theta) + \gamma\widehat{\ell}_k(\theta)$
13:     `#Perform gradient step`
14:     $\theta \leftarrow \theta - \eta\nabla_\theta\left(\ell(\theta) + \lambda\|\theta\|^2\right)$
15:     `#Update parameters of target vector field`
16:     $\bar{\theta} \leftarrow \zeta\bar{\theta} + (1-\zeta)\theta$
17: **end for**

*Table 4.* Summary of how different TD-flow algorithms generate the target probability path and vector field. The neural ode $\psi_t$ is defined by the vector field $\widehat{v}_t$ computed at iteration $n$.

| | $\widehat{\mathbf{p}}_{\mathbf{t}}$ | $\widehat{\mathbf{v}}_{\mathbf{t}}$ |
|---|---|---|
| TD-CFM | $X_0 \sim m_0$ 
 $X_1 = \psi_1(X_0 \mid S', A'; \bar\theta)$ 
 $X_t \sim p_{t|1}(\cdot \mid X_1)$ | $u_{t|1}(X_t \mid X_1)$ |
| TD-CFM(C) | $X_0 \sim m_0$ 
 $X_1 = \psi_1(X_0 \mid S', A'; \bar\theta)$ 
 $X_t \sim p_{t|0,1}(\cdot \mid X_0, X_1)$ | $u_{t|0,1}(X_t \mid X_0, X_1)$ |
| TD$^2$-CFM | $X_0 \sim m_0$ 
 $X_t = \psi_t(X_0 \mid S', A'; \bar\theta)$ | $v_t(X_t \mid S', A'; \bar\theta)$ |

To discuss the TD-Flow methods introduced herein, we first unify the loss function through defining a general template for the loss as,

$$
\begin{aligned}
\ell(\theta) = {} & (1-\gamma)\mathbb{E}_{\rho,t,X_t\sim\vec{p}_{t|1}(\cdot|S')}\left[\left\| v_t(X_t \mid S, A; \theta) - \vec{u}_{t|1}(X_t \mid S') \right\|^2\right] \\
& + \gamma\mathbb{E}_{\rho,t,X_t\sim\widehat{p}_t^{(n)}(\cdot|Z)}\left[\left\| v_t(X_t \mid S, A; \theta) - \widehat{v}_t^{(n)}(X_t \mid Z) \right\|^2\right].
\end{aligned}
$$

We can now recover each algorithm by a specific choice of the target probability path $\widehat{p}_t^{(n)}$ and vector field $\widehat{v}_t^{(n)}$ as illustrated in Figure 4. Based on this unified structure, we present pseudo-code for the TD flow methods in Figure 1. In practice, instead of proceeding through full iterations, we use standard mini-batch gradient updates with a target network $\bar\theta$ updated as a moving average of $\theta$.

When employing the conditional probability path $\vec{p}_{t|1}$ and vector field $\vec{u}_{t|1}$ we use the standard Gaussian linear interpolation defined as $\vec{p}_{t|1}(\cdot \mid X_1) = \mathcal{N}(\cdot \mid tX_1, (1-t)^2 I)$, hence $X_t = tX_1 + (1-t)X_0 \sim p_{t|1}$, resulting in $\vec{u}_{t|1}(X_t \mid X_1) = (X_1 - X_t)/(1-t)$ (Lipman et al., 2023). The source distribution for all experiments is $m_0(\cdot) = \mathcal{N}(\cdot \mid 0, I)$. To sample from the Neural ODE we use the Midpoint method with a constant step size of $\mathrm{d}t = t/10$ for a total of 10 steps. We found both coupled and TD$^2$ methods do not require many solver steps and hypothesize this is due to the reduction in transport cost as analyzed in Appendix E.7.

For all flow and diffusion-based methods, we employ a U-Net-style architecture (Ronneberger et al., 2015) that has hierarchical skip connections throughout an MLP. We embed the timestep $t$ by first increasing its dimensionality with a sinusoidal embedding before transforming it through a two-layer MLP with mish activations (Misra, 2019). We further process additional conditioning information, such as the state-action pair and Forward-Backward embedding $z$ through an additional two-layer MLP, whose result then gets concatenated with our time embedding. Finally, the network integrates all prior conditioning information through FiLM modulation (Perez et al., 2018) that replaces the learned affine transformation for layer normalization (Ba et al., 2016).

### C.3.2. DENOISING DIFFUSION

We train a Denoising Diffusion Probabilistic Model (DDPM; Ho et al., 2020) using the same architecture as our flow matching model above, with the output now being interpreted as a prediction of the noise seed $\epsilon_0$ that began the diffusion process. We discretize the diffusion process using $1{,}000$ steps with $\beta_{\min} = 0.1$ and $\beta_{\max} = 20$. We employ the DDIM sampler(Song et al., 2021a) with 50 sampling steps for both training and evaluation.

For evaluating our DDPM model, we compute exact log-likelihoods using the instantaneous change of variables formula (Chen et al., 2018) along with the probability flow ODE from Song et al. (2021b). That is, we solve the initial value problem in (24) using the vector field,

$$v_t(x_t \mid s, a; \theta) = -\frac{1}{2}\left(\beta_{\min} + t\left(\beta_{\max} - \beta_{\min}\right)\right)\left(x_t - \frac{1}{\sqrt{1 - \bar{\alpha}_t}}\epsilon_t(x_t \mid s, a; \theta)\right).$$

We now outline the losses for each of the TD-DPM experiments in the paper:

**TD-DD** To train our vanilla Diffusion GHM we employ the standard DDPM-style objective, that is, we optimize the following loss:

$$\mathbb{E}_{\substack{\rho,\, t,\, \epsilon \sim \mathcal{N}(\cdot \mid 0, I) \\ X_0 \sim \left(\mathcal{T}^\pi \widetilde{m}^{(n)}\right)(\cdot \mid S, A)}}\left[\left\|\epsilon - \epsilon_t(\sqrt{\bar{\alpha}_t}X_0 + \sqrt{1 - \bar{\alpha}_t}\epsilon \mid S, A; \theta)\right\|^2\right], \tag{25}$$

where $\bar{\theta}$ are the target parameters and $\bar{\alpha}$ are the standard diffusion coefficients as seen in Ho et al. (2020).

**TD$^2$-DD** As outlined in §3.1 we can split our DDPM loss into two terms, one that will use standard DDPM training on one-step transitions and the second term that will regress to our target networks noise prediction. This materializes as,

$$\vec{\ell}(\theta) = \mathbb{E}_{\rho, t, \epsilon, X_0}\left[\left\|\epsilon_t(\sqrt{\bar{\alpha}_t}X_0 + \sqrt{1 - \bar{\alpha}_t}\epsilon \mid S, A; \theta) - \epsilon\right\|^2\right]$$
$$\text{where } X_0 \sim P(\cdot \mid S, A),$$

$$\widehat{\ell}(\theta) = \mathbb{E}_{\rho, t, \epsilon, \widehat{X}_t}\left[\left\|\epsilon_t(\widehat{X}_t \mid S, A; \theta) - \epsilon_t^{(n)}(\widehat{X}_t \mid S', \pi(S'))\right\|^2\right]$$
$$\text{where } \widehat{X}_t \sim q_{t|T}^{(n)}(\cdot \mid S', \pi(S'))$$

$$\ell_{\text{TD}^2\text{-DD}}(\theta) = (1 - \gamma)\vec{\ell}(\theta) + \gamma\widehat{\ell}(\theta) \tag{26}$$

### C.3.3. GENERATIVE ADVERSARIAL NETWORK

We implement a modern Generative Adversarial Network (GAN; Goodfellow et al., 2014) baseline based on the recommendations in Huang et al. (2024). Specifically, we train a relativistic GAN (Jolicoeur-Martineau, 2019) resulting in the following loss,

$$\ell_{\text{GAN}}(\theta_{\text{G}}, \theta_{\text{D}}) = \mathbb{E}_{\rho, X_0, X_1}\left[f\left(D(G(X_0 \mid S, A; \theta_{\text{G}}); \theta_{\text{D}}) - D(X_1 \mid S, A; \theta_{\text{D}})\right)\right],$$
$$\text{where } X_0 \sim \mathcal{N}(\cdot \mid 0, I), X_1 \sim \left(\mathcal{T}^\pi \widetilde{m}^{(n)}\right)(\cdot \mid S, A),$$

We take $f(x) = -\log\left(1 + \exp\left(-x\right)\right)$ to be the log-sigmoid function (Jolicoeur-Martineau, 2019) and further add the following zero-centered gradient penalties on the discriminator,

$$R_1(\theta_{\text{D}}) = \mathbb{E}_{\rho, X \sim \left(\mathcal{T}^\pi \widetilde{m}^{(n)}\right)(\cdot \mid S, A)}\left[\left\|\nabla_X D(X \mid S, A)\right\|^2\right],$$
$$R_2(\theta_{\text{G}}, \theta_{\text{D}}) = \mathbb{E}_{\rho, X \sim (\mathcal{T}^\pi \widetilde{m})(\cdot \mid S, A; \theta_{\text{G}})}\left[\left\|\nabla_X D(X \mid S, A)\right\|^2\right].$$

The penalty $R_1$ penalizes the gradient norm of the discriminator $D$ on "real data" sampled from our current iterate $\widetilde{m}^{(n)}$, whereas $R_2$ penalizes the gradient norm on "fake data" generated directly from the current generator. We experimented with different coefficients and schedules for these gradient penalties and settled on a linear decay schedule from $0.05 \rightarrow 0.005$

throughout training. Furthermore, as is common practice, we impose a schedule on the second moment EMA coefficient $\beta_2$ of Adam (Kingma & Ba, 2015) to increase from $0.9 \rightarrow 0.99$ throughout training.

The generator and discriminator architecture in our GAN is implemented as a Residual MLP with leaky ReLU activations with the same FiLM-style conditioning (Perez et al., 2018) as our flow and diffusion models. The input to our generator is random noise sampled from an isotropic Gaussian with dimensionality equal to the number of state dimensions in the environment.

### C.3.4. VARIATIONAL AUTO-ENCODER

We implement a $\beta$-Variational Auto-Encoder (Kingma & Welling, 2014; Higgins et al., 2017) following the best practices outlined in Thakoor et al. (2022). That is, we train our VAE to minimize the following loss,

$$\ell_{\text{VAE}}(\theta_{\text{E}}, \theta_{\text{D}}) = \mathbb{E}_{\rho, X_1}\left[ \mathbb{E}_{X_0 \sim q_{\theta_{\text{E}}}(\cdot | S, A, X_1)}\left[ \log p_{\theta_{\text{D}}}(X_1 \mid S, A, X_0)\right] - \beta D_{\text{KL}}(q_{\theta_{\text{E}}} \| p_0)\right],$$
$$\text{where } X_1 \sim \left(\mathcal{T}^\pi \widetilde{m}^{(n)}\right)(\cdot \mid S, A).$$

We employ a similar architecture to our GAN-GHM and use a residual MLP for the encoder and decoder. We use an isotropic Gaussian latent space with the number of latents equal to the number of state dimensions in the environment. We also swept over $\beta \in \{0.1, 0.2, 0.3, 0.4, 0.5, 0.6, 0.7, 0.8, 0.9, 1.0\}$ on the MAZE domain and chose $\beta = 0.5$ for the rest of our experiments. Overall, we found the $\beta$-VAE-based GHM to be very unstable and likely requires very careful fine-tuning of $\beta$ to get adequate performance at long-horizons.

## C.4. Hyperparameters

We report the hyper-parameters for training the GHM models used in the single and multi-policy experiments. Table 5 shows the parameters for Flow Matching and Denoising Diffusion. We also report the hyper-parameters for pre-training the Forward-Backward representation (Touati & Ollivier, 2021) utilized in the multi-policy GHM experiments in Table 8.

*Table 5.* Flow Matching and Denoising Diffusion hyper-parameters used for the single and multi-policy experiments across tasks and domains. We highlight any differences depending on the training context.

|  | Hyperparameter | Single Policy | Multi-Policy |
|---|---|---|---|
| Flow Matching (Lipman et al., 2023) | ODE Solver | Midpoint | Midpoint |
|  | ODE $\mathrm{d}t$ (train) | 0.1 | 0.1 |
|  | ODE $\mathrm{d}t$ (eval) | 0.1 | 0.05 (0.1 for GPI) |
| Diffusion (DDPM) (Ho et al., 2020) | $\beta_{\min}$ | 0.1 | 0.1 |
|  | $\beta_{\max}$ | 20 | 20 |
|  | Discretization Steps | $1,000$ | $1,000$ |
|  | SDE Solver | DDIM (Song et al., 2021a) | DDIM (Song et al., 2021a) |
|  | SDE Solver Steps (train) | 20 | 20 |
|  | SDE Solver Steps (eval) | 20 | 20 |
| Network (U-Net) (Ronneberger et al., 2015) | $t$-Positional Embedding Dim. | 256 | 256 |
|  | $t$-Positional Embedding MLP | $(256, 256)$ | $(256, 256)$ |
|  | Hidden Activation | mish (Misra, 2019) | mish (Misra, 2019) |
|  | Blocks per Stage | 1 | 1 |
|  | Block Dimensions | $(512, 512, 512)$ | $(1024, 1024, 1024)$ |
| Conditional Encoder | Encoder Input | $s, a$ | $s, a, z$ |
|  | Encoder MLP | $(512, 512, 512)$ | $(1024, 1024, 1024)$ |
|  | Encoder Activation | mish (Misra, 2019) | mish (Misra, 2019) |
| Optimizer (AdamW) (Loshchilov & Hutter, 2019) | AdamW $\beta_1$ | 0.9 | 0.9 |
|  | AdamW $\beta_2$ | 0.999 | 0.999 |
|  | AdamW $\epsilon$ | $10^{-4}$ | $10^{-4}$ |
|  | Learning Rate | $10^{-4}$ | $10^{-4}$ |
|  | Weight Decay | $10^{-3}$ | $10^{-2}$ |
| Common | Gradient Steps | 3M | 8M |
|  | Batch Size | 1024 | 1024 |
|  | Target Network EMA | $10^{-3}$ | $10^{-4}$ |

*Table 6.* $\beta$-VAE (Higgins et al., 2017) hyper-parameters for single policy experiments across tasks and domains.

| | Hyperparameter | Value |
|---|---|---|
| $\beta$-VAE (Higgins et al., 2017) | $\beta$ | 10 |
| | Latent Prior | $\mathcal{N}(0, I)$ |
| | Latent Dimension | \|S\| |
| Network | Encoder | Residual MLP |
| | Decoder | Residual MLP |
| | Hidden Activation | mish (Misra, 2019) |
| | Blocks per Stage | 1 |
| | Block Dimensions | $(512, 512, 512)$ |
| Conditional Encoder | Encoder Input | $s, a$ |
| | Encoder MLP | $(512, 512, 512)$ |
| | Encoder Activation | mish (Misra, 2019) |
| Optimizer (AdamW) (Loshchilov & Hutter, 2019) | AdamW $\beta_1$ | 0.9 |
| | AdamW $\beta_2$ | 0.999 |
| | AdamW $\epsilon$ | $10^{-4}$ |
| | Learning Rate | $10^{-4}$ |
| | Weight Decay | $10^{-3}$ |
| Common | Gradient Steps | 3M |
| | Batch Size | 1024 |
| | Target Network EMA | $10^{-3}$ |

*Table 7.* GAN hyper-parameters for single policy experiments across tasks and domains.

| | Hyperparameter | Value |
|---|---|---|
| RGAN (Jolicoeur-Martineau, 2019) | Grad. Penalty Coef | Linear($0.05 \rightarrow 0.005$) |
| | Latent Prior | $\mathcal{N}(0, I)$ |
| | Latent Dimension | \|S\| |
| Network | Generator | Residual MLP |
| | Discriminator | Residual MLP |
| | Hidden Activation | Leaky ReLU |
| | Blocks per Stage | 1 |
| | Block Dimensions | $(512, 512, 512)$ |
| Conditional Encoder | Encoder Input | $s, a$ |
| | Encoder MLP | $(512, 512, 512)$ |
| | Encoder Activation | Leaky ReLU |
| Optimizer (AdamW) (Loshchilov & Hutter, 2019) | AdamW $\beta_1$ | 0.9 |
| | AdamW $\beta_2$ | Linear($0.9 \rightarrow 0.99$) |
| | AdamW $\epsilon$ | $10^{-4}$ |
| | Learning Rate | $10^{-4}$ |
| | Weight Decay | $10^{-3}$ |
| Common | Gradient Steps | 3M |
| | Batch Size | 1024 |
| | Target Network EMA | $10^{-3}$ |

*Table 8.* Forward Backward Representation hyper-parameters. We largely reuse the hyper-parameters from Pirotta et al. (2024) and highlight any deviations.

| | Hyperparameter | Walker | Cheetah | Quadruped | Maze |
|---|---|---|---|---|---|
| Forward Backward (Touati & Ollivier, 2021) | Embedding Dimension $d$ | 100 | 50 | 50 | 100 |
| | Embedding Prior | $S^d$ | $S^d$ | $S^d$ | $S^d$ |
| | Embedding Prior Goal Prob. | 0 | 0 | 0 | $1/2$ |
| | $B$ Normalization | $\ell_2$ | $\ell_2$ | $\ell_2$ | $\ell_2$ |
| | Orthonormal Loss Coeff. | 1 | 1 | 1 | 1 |
| Policy (TD3) (Fujimoto et al., 2018) | Target Policy Noise | $\mathcal{N}(0, 0.2)$ | $\mathcal{N}(0, 0.2)$ | $\mathcal{N}(0, 0.2)$ | $\mathcal{N}(0, 0.2)$ |
| | Target Policy Clipping | 0.3 | 0.3 | 0.3 | 0.3 |
| | Policy Update Frequency | 1 | 1 | 1 | 1 |
| Optimizer (Adam) (Kingma & Ba, 2015) | Learning Rate (F, B) | $(10^{-4}, 10^{-4})$ | $(10^{-4}, 10^{-4})$ | $(10^{-4}, 10^{-4})$ | $(10^{-4}, 10^{-6})$ |
| | Learning Rate ($\pi$) | $10^{-4}$ | $10^{-4}$ | $10^{-4}$ | $10^{-6}$ |
| | Adam $\beta_1$ | 0.9 | 0.9 | 0.9 | 0.9 |
| | Adam $\beta_2$ | 0.999 | 0.999 | 0.999 | 0.999 |
| | Adam $\epsilon$ | $10^{-8}$ | $10^{-8}$ | $10^{-8}$ | $10^{-8}$ |
| Common | Batch Size | 2048 | 1024 | 2048 | 1024 |
| | Gradient Steps | 3M | 3M | 3M | 5M |
| | Discount Factor $\gamma$ | 0.98 | 0.98 | 0.98 | 0.99 |
| | Target Network EMA | 0.99 | 0.99 | 0.99 | 0.99 |
| | Reward Inference Samples | $250,000$ | $250,000$ | $250,000$ | $250,000$ |

# D. Additional Experimental Results

In this section, we report additional results about the experiments.

**Single policy.** We report metrics averaged over tasks using a curved conditional path in Table 12. We also report the performance per task in Table 13. Table 11 shows the performance of the single-policy experiments (§5.1 in the main paper) expanded for each task. While the performance of TD-based methods is reasonably stable across tasks, VAE and GAN have a large variance across tasks. For example, the EMD of GAN diverges in 2 tasks out of 4 in QUADRUPED.

**Multiple policies and planning.** We report aggregate performance across our full suite of evaluation metrics for the multi-policy experiments in Table 14. We also report per-task metrics in Table 16. We can notice that $\text{TD}^2$-DD achieves quite a high EMD compared to TD-DD while achieving a better MSE(V). By further inspecting the generated samples (see Figure 5), we found that TD-DD tends to generate highly concentrated samples, while $\text{TD}^2$-DD is more diffuse. However, the samples generated by TD-DD appear to be better at a visual inspection. This may explain the discrepancy between the two metrics. Finally, we report aggregate planning performance in Table 15 and per-task results in Table 17.

**Comparison with planning with one-step world model** we include in Table 9 results for a Model Predictive Path Integral (MPPI) controller with a learned dynamics model. We train a similar capacity dynamics model to that of $\text{TD}^2$-CFM before evaluating MPPI with a finite horizon of 32 for locomotion tasks and 128 for maze, where at each step we sample 256 action candidates and perform 10 optimization rounds with 64 elites (top-k actions) per round. The results show that GPI with $\text{TD}^2$-CFM significantly outperforms MPPI in 3/4 domains with comparable results in Walker. MPPI notably displayed instability related to compounding errors in environments with difficult to model dynamics.

**Impact of number of ODE integration steps** we report in Table 10 an empirical analysis showing how prediction quality degrades as we reduce the number of integration steps on the Loop task in Pointmass Maze. The results show that $\text{TD}^2$-CFM remains robust even at coarse discretizations of the ODE with as little as 5 integration steps, while we observe with a predictable degradation when the number of steps is too small,

*Table 9.* Comparison with planning with one-step world models

| Domain | FB | TD²-CFM-GPI | MPPI |
|---|---|---|---|
| Cheetah | 479.35 (14.56) | 693.63 (5.50) | 541.22 (5.28) |
| Pointmass | 472.45 (14.40) | 800.99 (8.56) | 286.43 (54.95) |
| Quadruped | 627.28 (1.98) | 695.73 (2.07) | 156.80 (122.89) |
| Walker | 526.66 (5.94) | 627.63 (7.97) | 658.15 (21.46) |

*Table 10.* Ablation of the number of ODE integration steps

| ODE Steps | NLL ↓ | EMD ↓ | MSE(VF) ↓ |
|---|---|---|---|
| 2 | -0.48 (0.21) | 0.076 (0.003) | 379.52 (81.75) |
| 5 | -2.75 (0.15) | 0.036 (0.000) | 23.82 (2.05) |
| 10 | -2.85 (0.17) | 0.025 (0.001) | 7.71 (2.75) |
| 20 | -2.99 (0.04) | 0.0218 (0.001) | 4.40 (0.82) |

*Table 11.* **Per task** results for the **single policy** experiments.

**Walker**

| Task | Method | EMD↓ | NLL↓ | MSE(V)↓ |
|---|---|---|---|---|
| RUN | TD-DD | 20.06 (0.27) | 2.713 (0.189) | 120.21 (52.91) |
| | $TD^2$-DD | 11.05 (0.01) | 0.543 (0.164) | 24.02 (25.54) |
| | TD-CFM | 12.46 (0.35) | 0.608 (0.026) | 148.56 (29.24) |
| | TD-CFM(C) | 10.90 (0.05) | 0.112 (0.018) | 9.53 (1.37) |
| | $TD^2$-CFM | 10.59 (0.13) | −0.026 (0.005) | 11.90 (7.59) |
| | GAN | 23.99 (1.15) | — | 827.79 (130.38) |
| | VAE | 114.95 (2.51) | — | 646.96 (21.57) |
| SPIN | TD-DD | 21.55 (0.17) | 2.754 (0.062) | 1812.90 (2016.68) |
| | $TD^2$-DD | 13.57 (0.02) | 0.561 (0.014) | 21.81 (9.55) |
| | TD-CFM | 15.46 (0.26) | 0.838 (0.021) | 379.13 (180.63) |
| | TD-CFM(C) | 12.94 (0.08) | 0.321 (0.009) | 22.36 (4.99) |
| | $TD^2$-CFM | 13.27 (0.15) | 0.200 (0.007) | 7.14 (1.72) |
| | GAN | 26.85 (1.98) | — | 2948.74 (4541.66) |
| | VAE | 103.73 (6.86) | — | 431.70 (87.04) |
| STAND | TD-DD | 19.82 (0.07) | 2.579 (0.180) | 56.11 (18.32) |
| | $TD^2$-DD | 12.15 (0.20) | 0.487 (0.040) | 21.65 (4.76) |
| | TD-CFM | 13.30 (0.17) | 0.669 (0.041) | 32.95 (8.33) |
| | TD-CFM(C) | 12.25 (0.11) | 0.218 (0.002) | 12.76 (3.17) |
| | $TD^2$-CFM | 12.27 (0.14) | 0.126 (0.019) | 14.96 (10.23) |
| | GAN | 22.98 (1.31) | — | 5041.85 (654.87) |
| | VAE | 114.46 (0.28) | — | 3863.70 (38.24) |
| WALK | TD-DD | 21.29 (0.46) | 2.635 (0.072) | 121.50 (34.67) |
| | $TD^2$-DD | 11.59 (0.34) | 0.558 (0.080) | 88.67 (28.94) |
| | TD-CFM | 12.91 (0.16) | 0.738 (0.084) | 340.43 (63.65) |
| | TD-CFM(C) | 11.55 (0.02) | 0.225 (0.036) | 78.20 (14.20) |
| | $TD^2$-CFM | 11.54 (0.27) | 0.118 (0.022) | 79.41 (16.24) |
| | GAN | 24.21 (1.43) | — | 5944.23 (302.73) |
| | VAE | 113.79 (1.65) | — | 4888.07 (78.85) |

**Quadruped**

| Task | Method | EMD↓ | NLL↓ | MSE(V)↓ |
|---|---|---|---|---|
| JUMP | TD-DD | 27.89 (0.67) | 1.890 (0.025) | 1778.78 (611.15) |
| | $TD^2$-DD | 25.62 (3.75) | 0.906 (0.013) | 12.88 (2.07) |
| | TD-CFM | 15.68 (0.15) | 1.068 (0.006) | 523.10 (42.47) |
| | TD-CFM(C) | 14.12 (0.00) | 0.518 (0.002) | 10.10 (1.32) |
| | $TD^2$-CFM | 14.27 (0.06) | 0.426 (0.005) | 12.89 (2.86) |
| | GAN | 18.23 (0.34) | — | 3546.34 (984.61) |
| | VAE | 60.54 (0.29) | — | 1939.62 (22.15) |
| RUN | TD-DD | 28.01 (1.02) | 1.975 (0.061) | 438.92 (310.44) |
| | $TD^2$-DD | 22.79 (3.08) | 0.856 (0.033) | 32.38 (4.36) |
| | TD-CFM | 15.74 (0.05) | 1.051 (0.026) | 170.86 (19.61) |
| | TD-CFM(C) | 14.62 (0.11) | 0.457 (0.006) | 26.01 (4.44) |
| | $TD^2$-CFM | 14.75 (0.05) | 0.338 (0.004) | 18.36 (2.62) |
| | GAN | 19.21 (0.13) | — | 195.11 (144.29) |
| | VAE | 60.56 (0.21) | — | 428.69 (10.48) |
| STAND | TD-DD | 28.57 (0.50) | 1.832 (0.034) | 2083.77 (1767.03) |
| | $TD^2$-DD | 20.81 (1.81) | 0.867 (0.040) | 20.09 (19.08) |
| | TD-CFM | 15.03 (0.18) | 1.003 (0.026) | 505.51 (88.47) |
| | TD-CFM(C) | 13.91 (0.02) | 0.483 (0.005) | 12.86 (4.65) |
| | $TD^2$-CFM | 14.07 (0.12) | 0.393 (0.021) | 7.77 (0.91) |
| | GAN | 91273.39 (81559.61) | — | 3631.15 (2289.14) |
| | VAE | 59.42 (0.49) | — | 859.51 (101.82) |
| WALK | TD-DD | 28.83 (0.41) | 1.934 (0.075) | 1661.52 (402.07) |
| | $TD^2$-DD | 21.36 (1.70) | 0.815 (0.040) | 570.75 (35.38) |
| | TD-CFM | 16.48 (0.09) | 1.103 (0.006) | 900.78 (85.36) |
| | TD-CFM(C) | 14.89 (0.01) | 0.494 (0.006) | 572.02 (24.55) |
| | $TD^2$-CFM | 14.96 (0.13) | 0.361 (0.022) | 528.06 (11.32) |
| | GAN | 55777.67 (28193.15) | — | 3166.15 (54.62) |
| | VAE | 60.57 (0.54) | — | 1397.52 (100.28) |

**Pointmass Maze**

| Task | Method | EMD↓ | NLL↓ | MSE(V)↓ |
|---|---|---|---|---|
| LOOP | TD-DD | 0.189 (0.003) | 3.462 (0.232) | 4717.87 (83.53) |
| | $TD^2$-DD | 0.031 (0.003) | 0.577 (0.027) | 4.27 (1.36) |
| | TD-CFM | 0.071 (0.007) | 0.748 (0.070) | 677.48 (154.81) |
| | TD-CFM(C) | 0.025 (0.002) | −0.703 (0.032) | 10.91 (2.35) |
| | $TD^2$-CFM | 0.020 (0.001) | −0.674 (0.072) | 1.75 (0.13) |
| | GAN | 0.225 (0.014) | — | 2276.26 (361.04) |
| | VAE | 0.456 (0.045) | — | 4011.19 (85.44) |
| REACH BOTTOM LEFT | TD-DD | 0.139 (0.002) | 2.808 (0.058) | 320.80 (27.06) |
| | $TD^2$-DD | 0.025 (0.001) | 0.980 (0.174) | 5.76 (3.15) |
| | TD-CFM | 0.059 (0.004) | 0.520 (0.031) | 224.13 (33.19) |
| | TD-CFM(C) | 0.024 (0.002) | −0.729 (0.167) | 16.58 (12.10) |
| | $TD^2$-CFM | 0.020 (0.002) | −0.984 (0.053) | 10.44 (7.08) |
| | GAN | 0.269 (0.150) | — | 1199.80 (212.47) |
| | VAE | 0.313 (0.029) | — | 981.22 (195.70) |
| REACH BOTTOM RIGHT | TD-DD | 0.174 (0.004) | 3.270 (0.257) | 230.79 (18.24) |
| | $TD^2$-DD | 0.025 (0.001) | 0.640 (0.283) | 4.82 (2.61) |
| | TD-CFM | 0.066 (0.002) | 0.549 (0.040) | 166.07 (35.75) |
| | TD-CFM(C) | 0.023 (0.001) | −0.759 (0.034) | 10.95 (2.63) |
| | $TD^2$-CFM | 0.020 (0.002) | −0.855 (0.022) | 4.84 (3.08) |
| | GAN | 0.170 (0.018) | — | 416.75 (54.72) |
| | VAE | 0.505 (0.051) | — | 489.06 (6.44) |
| REACH TOP LEFT | TD-DD | 0.102 (0.001) | 2.407 (0.059) | 593.98 (72.33) |
| | $TD^2$-DD | 0.033 (0.003) | 0.863 (0.255) | 34.43 (10.96) |
| | TD-CFM | 0.055 (0.006) | 0.454 (0.167) | 472.54 (308.65) |
| | TD-CFM(C) | 0.021 (0.003) | −0.517 (0.445) | 14.85 (3.28) |
| | $TD^2$-CFM | 0.025 (0.002) | −0.797 (0.057) | 23.48 (5.46) |
| | GAN | 0.132 (0.022) | — | 1350.49 (716.52) |
| | VAE | 0.321 (0.029) | — | 2404.42 (498.13) |
| REACH TOP RIGHT | TD-DD | 0.141 (0.002) | 2.924 (0.243) | 362.56 (8.06) |
| | $TD^2$-DD | 0.023 (0.003) | 0.743 (0.259) | 6.38 (1.55) |
| | TD-CFM | 0.059 (0.002) | 0.501 (0.018) | 237.57 (47.18) |
| | TD-CFM(C) | 0.020 (0.001) | −0.771 (0.090) | 6.18 (3.37) |
| | $TD^2$-CFM | 0.018 (0.001) | −0.903 (0.074) | 3.21 (2.22) |
| | GAN | 0.218 (0.044) | — | 1043.01 (337.10) |
| | VAE | 0.453 (0.106) | — | 1223.57 (80.69) |

**Cheetah**

| Task | Method | EMD↓ | NLL↓ | MSE(V)↓ |
|---|---|---|---|---|
| FLIP | TD-DD | 20.31 (0.31) | 2.669 (0.086) | 601.62 (314.84) |
| | $TD^2$-DD | 14.44 (1.79) | 0.758 (0.028) | 172.03 (35.51) |
| | TD-CFM | 11.90 (0.03) | 0.868 (0.008) | 211.92 (26.25) |
| | TD-CFM(C) | 10.55 (0.03) | 0.485 (0.024) | 124.08 (17.89) |
| | $TD^2$-CFM | 10.67 (0.04) | 0.447 (0.021) | 67.76 (21.99) |
| | GAN | 23.55 (2.52) | — | 3608.55 (1948.65) |
| | VAE | 83.00 (1.02) | — | 3339.01 (44.80) |
| FLIP BACKWARD | TD-DD | 16.67 (0.02) | 2.647 (0.186) | 1043.27 (369.92) |
| | $TD^2$-DD | 12.99 (2.64) | 0.894 (0.025) | 463.04 (89.08) |
| | TD-CFM | 10.91 (0.12) | 0.927 (0.047) | 398.66 (59.04) |
| | TD-CFM(C) | 9.90 (0.07) | 0.542 (0.023) | 410.49 (77.16) |
| | $TD^2$-CFM | 10.11 (0.14) | 0.542 (0.006) | 370.69 (112.59) |
| | GAN | 20.80 (1.56) | — | 3761.79 (785.37) |
| | VAE | 84.65 (0.31) | — | 918.32 (25.62) |
| RUN | TD-DD | 20.26 (0.06) | 2.907 (0.336) | 46.48 (13.06) |
| | $TD^2$-DD | 16.91 (4.04) | 0.813 (0.028) | 86.53 (55.44) |
| | TD-CFM | 12.21 (0.05) | 0.872 (0.032) | 54.98 (11.01) |
| | TD-CFM(C) | 10.44 (0.08) | 0.434 (0.018) | 24.52 (5.89) |
| | $TD^2$-CFM | 10.53 (0.08) | 0.412 (0.020) | 27.69 (5.44) |
| | GAN | 25.48 (2.01) | — | 183.47 (72.39) |
| | VAE | 83.91 (0.57) | — | 109.45 (9.86) |
| RUN BACKWARD | TD-DD | 21.47 (0.32) | 3.074 (0.376) | 20.28 (5.95) |
| | $TD^2$-DD | 13.04 (1.22) | 0.818 (0.016) | 14.87 (2.34) |
| | TD-CFM | 13.38 (0.20) | 0.989 (0.056) | 37.90 (2.98) |
| | TD-CFM(C) | 11.02 (0.05) | 0.452 (0.023) | 8.71 (1.05) |
| | $TD^2$-CFM | 11.06 (0.08) | 0.414 (0.016) | 8.33 (1.89) |
| | GAN | 24.77 (0.43) | — | 270.21 (4.08) |
| | VAE | 82.91 (0.36) | — | 734.77 (22.94) |
| WALK | TD-DD | 21.57 (0.84) | 2.790 (0.151) | 546.05 (86.30) |
| | $TD^2$-DD | 12.85 (1.67) | 0.780 (0.047) | 238.01 (11.17) |
| | TD-CFM | 12.27 (0.12) | 0.802 (0.034) | 377.45 (101.61) |
| | TD-CFM(C) | 10.24 (0.17) | 0.354 (0.021) | 176.99 (28.54) |
| | $TD^2$-CFM | 10.18 (0.08) | 0.336 (0.021) | 229.89 (21.93) |
| | GAN | 24.39 (1.11) | — | 3520.88 (1050.76) |
| | VAE | 84.39 (0.41) | — | 2138.32 (233.01) |
| WALK BACKWARD | TD-DD | 21.05 (0.30) | 2.854 (0.094) | 469.23 (133.50) |
| | $TD^2$-DD | 14.64 (2.48) | 0.771 (0.019) | 160.42 (42.25) |
| | TD-CFM | 12.89 (0.14) | 0.857 (0.033) | 291.71 (66.89) |
| | TD-CFM(C) | 10.88 (0.01) | 0.412 (0.023) | 99.90 (4.20) |
| | $TD^2$-CFM | 10.86 (0.12) | 0.381 (0.014) | 106.97 (10.45) |
| | GAN | 24.86 (0.34) | — | 3434.43 (189.45) |
| | VAE | 83.73 (0.63) | — | 465.72 (16.06) |

*Table 12.* Results averaged over tasks for the **single policy** experiments with a **curved** conditional path.

| Domain | Method | EMD ↓ | NLL ↓ | MSE(V) ↓ |
|---|---|---|---|---|
| CHEETAH | TD-CFM | 13.91 (0.73) | 1.354 (0.017) | 477.89 (40.53) |
| | TD-CFM(C) | 25.86 (18.91) | 1.295 (0.067) | 189.21 (17.69) |
| | TD²-CFM | 10.79 (0.03) | 0.412 (0.014) | 121.67 (5.68) |
| POINTMASS MAZE | TD-CFM | 0.091 (0.003) | 1.156 (0.081) | 758.16 (103.54) |
| | TD-CFM(C) | 0.089 (0.008) | 4.340 (0.456) | 679.29 (20.11) |
| | TD²-CFM | 0.021 (0.000) | −0.806 (0.017) | 9.22 (1.40) |
| QUADRUPED | TD-CFM | 15.63 (0.09) | 1.478 (0.088) | 273.68 (34.07) |
| | TD-CFM(C) | 34.00 (6.96) | 0.930 (0.036) | 522.28 (155.42) |
| | TD²-CFM | 14.56 (0.02) | 0.327 (0.014) | 142.18 (9.38) |
| WALKER | TD-CFM | 13.10 (0.11) | 1.147 (0.042) | 608.47 (124.62) |
| | TD-CFM(C) | 33.20 (4.66) | 1.039 (0.052) | 189.66 (102.83) |
| | TD²-CFM | 12.00 (0.05) | 0.099 (0.005) | 27.56 (0.53) |

*Table 13.* **Per task** results for the **single policy** experiments with a **curved** conditional path.

**Walker**

| Task | Method | EMD ↓ | NLL ↓ | MSE(V) ↓ |
|---|---|---|---|---|
| RUN | TD-CFM | 12.39 (0.17) | 1.218 (0.107) | 326.14 (56.12) |
| | TD-CFM(C) | 23.80 (4.91) | 0.923 (0.191) | 69.39 (8.84) |
| | TD²-CFM | 10.69 (0.06) | −0.040 (0.008) | 11.69 (4.01) |
| SPIN | TD-CFM | 14.08 (0.12) | 1.410 (0.189) | 896.83 (278.52) |
| | TD-CFM(C) | 47.39 (3.14) | 1.801 (0.186) | 401.22 (321.52) |
| | TD²-CFM | 13.37 (0.11) | 0.198 (0.008) | 7.65 (2.29) |
| STAND | TD-CFM | 13.24 (0.23) | 0.896 (0.053) | 274.60 (121.20) |
| | TD-CFM(C) | 36.32 (13.80) | 0.625 (0.053) | 159.74 (31.53) |
| | TD²-CFM | 12.50 (0.17) | 0.119 (0.008) | 9.42 (1.95) |
| WALK | TD-CFM | 12.69 (0.20) | 1.067 (0.015) | 936.30 (86.71) |
| | TD-CFM(C) | 25.29 (7.62) | 0.808 (0.049) | 128.28 (65.70) |
| | TD²-CFM | 11.42 (0.20) | 0.119 (0.026) | 81.47 (3.53) |

**Quadruped**

| Task | Method | EMD ↓ | NLL ↓ | MSE(V) ↓ |
|---|---|---|---|---|
| JUMP | TD-CFM | 15.31 (0.17) | 1.460 (0.188) | 115.99 (138.59) |
| | TD-CFM(C) | 39.28 (8.90) | 0.980 (0.062) | 686.51 (314.49) |
| | TD²-CFM | 14.36 (0.07) | 0.358 (0.010) | 10.84 (3.05) |
| RUN | TD-CFM | 15.61 (0.16) | 1.450 (0.060) | 104.52 (33.53) |
| | TD-CFM(C) | 40.27 (7.59) | 0.898 (0.040) | 240.50 (58.83) |
| | TD²-CFM | 14.73 (0.06) | 0.288 (0.015) | 21.13 (3.52) |
| STAND | TD-CFM | 15.24 (0.11) | 1.515 (0.215) | 173.07 (34.09) |
| | TD-CFM(C) | 22.77 (6.86) | 0.924 (0.053) | 275.03 (249.91) |
| | TD²-CFM | 14.17 (0.09) | 0.342 (0.019) | 7.05 (1.80) |
| WALK | TD-CFM | 16.37 (0.10) | 1.486 (0.022) | 701.13 (83.58) |
| | TD-CFM(C) | 33.68 (4.69) | 0.917 (0.036) | 887.11 (120.92) |
| | TD²-CFM | 14.99 (0.08) | 0.318 (0.016) | 529.71 (35.40) |

**Pointmass Maze**

| Task | Method | EMD ↓ | NLL ↓ | MSE(V) ↓ |
|---|---|---|---|---|
| LOOP | TD-CFM | 0.112 (0.015) | 1.465 (0.171) | 1888.54 (444.66) |
| | TD-CFM(C) | 0.132 (0.031) | 5.191 (1.328) | 1354.09 (102.55) |
| | TD²-CFM | 0.020 (0.000) | −0.708 (0.013) | 2.31 (0.59) |
| REACH BOTTOM LEFT | TD-CFM | 0.096 (0.012) | 1.091 (0.142) | 628.74 (118.04) |
| | TD-CFM(C) | 0.078 (0.001) | 3.942 (0.576) | 820.02 (52.88) |
| | TD²-CFM | 0.022 (0.001) | −0.883 (0.057) | 10.55 (9.13) |
| REACH BOTTOM RIGHT | TD-CFM | 0.097 (0.001) | 1.296 (0.220) | 290.21 (29.94) |
| | TD-CFM(C) | 0.109 (0.009) | 5.310 (0.552) | 409.28 (10.79) |
| | TD²-CFM | 0.019 (0.001) | −0.833 (0.049) | 2.64 (0.30) |
| REACH TOP LEFT | TD-CFM | 0.070 (0.003) | 0.894 (0.139) | 500.63 (142.18) |
| | TD-CFM(C) | 0.048 (0.002) | 2.821 (0.268) | 75.79 (20.06) |
| | TD²-CFM | 0.025 (0.002) | −0.738 (0.011) | 26.56 (9.99) |
| REACH TOP RIGHT | TD-CFM | 0.083 (0.004) | 1.035 (0.138) | 482.68 (128.45) |
| | TD-CFM(C) | 0.080 (0.001) | 4.436 (0.305) | 737.30 (23.75) |
| | TD²-CFM | 0.019 (0.001) | −0.866 (0.026) | 4.02 (1.75) |

**Cheetah**

| Task | Method | EMD ↓ | NLL ↓ | MSE(V) ↓ |
|---|---|---|---|---|
| FLIP | TD-CFM | 12.92 (1.25) | 1.324 (0.042) | 342.71 (129.09) |
| | TD-CFM(C) | 22.90 (15.00) | 1.364 (0.108) | 140.32 (42.14) |
| | TD²-CFM | 10.89 (0.08) | 0.433 (0.012) | 74.34 (6.50) |
| FLIP BACKWARD | TD-CFM | 14.52 (4.08) | 1.346 (0.190) | 576.31 (169.57) |
| | TD-CFM(C) | 25.46 (25.58) | 1.427 (0.027) | 388.45 (87.18) |
| | TD²-CFM | 10.48 (0.23) | 0.538 (0.034) | 283.84 (40.81) |
| RUN | TD-CFM | 14.00 (0.77) | 1.390 (0.043) | 114.51 (3.11) |
| | TD-CFM(C) | 17.42 (5.78) | 1.423 (0.091) | 37.23 (8.74) |
| | TD²-CFM | 10.85 (0.08) | 0.405 (0.010) | 32.58 (8.42) |
| RUN BACKWARD | TD-CFM | 14.50 (0.31) | 1.439 (0.102) | 109.32 (5.35) |
| | TD-CFM(C) | 38.06 (28.90) | 1.283 (0.110) | 101.24 (149.88) |
| | TD²-CFM | 11.06 (0.05) | 0.399 (0.007) | 12.32 (2.34) |
| WALK | TD-CFM | 13.66 (0.71) | 1.290 (0.041) | 1040.43 (147.86) |
| | TD-CFM(C) | 21.01 (16.43) | 1.096 (0.028) | 343.71 (66.91) |
| | TD²-CFM | 10.45 (0.04) | 0.323 (0.010) | 213.87 (23.09) |
| WALK BACKWARD | TD-CFM | 13.83 (0.89) | 1.336 (0.033) | 684.05 (21.31) |
| | TD-CFM(C) | 30.29 (22.58) | 1.178 (0.206) | 124.29 (17.61) |
| | TD²-CFM | 11.00 (0.11) | 0.372 (0.015) | 113.09 (22.45) |

*Table 14.* **Per domain** results for the quantitative **multi-policy** experiments.

| Domain | Method | EMD↓ | NLL↓ | MSE(V)↓ |
|---|---|---|---|---|
| CHEETAH | TD-DD | 17.79 (0.40) | 1.442 (0.042) | 534.82 (107.81) |
| | TD²-DD | 74.35 (7.49) | 0.771 (0.020) | 253.89 (21.42) |
| | TD-CFM | 12.54 (0.04) | 1.044 (0.044) | 826.54 (58.01) |
| | TD-CFM(C) | 11.19 (0.11) | 0.581 (0.011) | 249.02 (19.81) |
| | TD²-CFM | 11.06 (0.08) | 0.481 (0.008) | 230.34 (44.81) |
| POINTMASS | TD-DD | 0.152 (0.006) | 2.048 (0.093) | 662.96 (76.86) |
| | TD²-DD | 0.349 (0.037) | 0.666 (0.027) | 312.98 (66.46) |
| | TD-CFM | 0.087 (0.003) | 0.771 (0.025) | 580.94 (41.28) |
| | TD-CFM(C) | 0.063 (0.000) | 0.174 (0.021) | 220.11 (100.36) |
| | TD²-CFM | 0.060 (0.002) | 0.043 (0.022) | 169.74 (85.76) |
| QUADRUPED | TD-DD | 20.21 (1.76) | 1.403 (0.022) | 499.88 (292.17) |
| | TD²-DD | 135.79 (9.24) | 0.901 (0.051) | 415.29 (101.86) |
| | TD-CFM | 15.06 (0.08) | 0.950 (0.024) | 391.12 (141.00) |
| | TD-CFM(C) | 14.98 (0.15) | 0.528 (0.016) | 176.62 (13.73) |
| | TD²-CFM | 14.74 (0.12) | 0.340 (0.010) | 178.95 (30.43) |
| WALKER | TD-DD | 21.49 (0.64) | 1.441 (0.009) | 571.72 (196.76) |
| | TD²-DD | 104.44 (2.84) | 0.688 (0.009) | 180.45 (47.82) |
| | TD-CFM | 15.08 (0.28) | 0.920 (0.023) | 768.13 (66.48) |
| | TD-CFM(C) | 13.57 (0.09) | 0.414 (0.019) | 179.39 (24.52) |
| | TD²-CFM | 13.70 (0.33) | 0.307 (0.008) | 154.75 (8.70) |

*Table 15.* **Per domain** results for the **multi-policy** experiments evaluating planning performance with generalized policy improvement.

| Domain | Method | Planner | Z-Distribution $D(\mathbb{Z})$ | | |
|---|---|---|---|---|---|
| | | | Random | Local Perturbation | Train Distribution |
| CHEETAH | FB | — | | 479.35 (14.56) | |
| | FB | GPI | 275.32 (2.50) | 401.08 (5.92) | 269.59 (8.18) |
| | TD-DD | GPI | 574.05 (3.88) | 604.53 (11.87) | 620.72 (14.29) |
| | TD²-DD | GPI | 662.17 (0.94) | 680.22 (5.98) | 678.98 (3.67) |
| | TD-CFM | GPI | 403.54 (81.24) | 426.46 (81.69) | 372.40 (99.68) |
| | TD-CFM(C) | GPI | 681.52 (6.49) | 700.97 (6.57) | 697.81 (3.16) |
| | TD²-CFM | GPI | 682.21 (5.41) | 692.72 (7.96) | 693.63 (5.50) |
| POINTMASS | FB | — | | 472.45 (14.40) | |
| | FB | GPI | −0.64 (7.70) | 240.54 (23.69) | −17.74 (4.34) |
| | TD-DD | GPI | 569.05 (37.58) | 599.92 (37.26) | 537.69 (47.54) |
| | TD²-DD | GPI | 763.95 (38.02) | 805.72 (2.23) | 788.87 (17.13) |
| | TD-CFM | GPI | 625.44 (23.12) | 671.53 (52.75) | 695.70 (27.88) |
| | TD-CFM(C) | GPI | 800.87 (3.46) | 812.44 (1.58) | 808.03 (2.77) |
| | TD²-CFM | GPI | 790.34 (14.16) | 813.90 (1.62) | 800.99 (8.56) |
| QUADRUPED | FB | — | | 627.28 (1.98) | |
| | FB | GPI | 671.95 (0.58) | 674.09 (0.53) | 646.05 (2.28) |
| | TD-DD | GPI | 657.98 (1.87) | 662.29 (1.46) | 657.44 (4.71) |
| | TD²-DD | GPI | 667.24 (6.32) | 671.54 (1.40) | 665.52 (5.12) |
| | TD-CFM | GPI | 669.35 (5.82) | 672.46 (4.96) | 668.61 (5.74) |
| | TD-CFM(C) | GPI | 695.52 (4.51) | 697.65 (5.21) | 696.18 (3.29) |
| | TD²-CFM | GPI | 696.58 (4.10) | 696.57 (2.36) | 695.73 (2.07) |
| WALKER | FB | — | | 526.66 (5.94) | |
| | FB | GPI | 35.23 (0.98) | 37.51 (1.20) | 39.04 (1.48) |
| | TD-DD | GPI | 512.65 (19.19) | 553.35 (14.28) | 533.37 (27.24) |
| | TD²-DD | GPI | 509.39 (10.26) | 598.40 (6.44) | 609.28 (5.87) |
| | TD-CFM | GPI | 506.62 (15.84) | 524.34 (4.75) | 537.24 (17.20) |
| | TD-CFM(C) | GPI | 513.24 (17.77) | 608.80 (16.14) | 624.19 (19.45) |
| | TD²-CFM | GPI | 518.07 (20.74) | 617.08 (6.55) | 627.63 (7.97) |

*Table 16.* **Per task** results for the quantitative **multi-policy** experiments.

### Walker

| Task | Method | EMD ↓ | NLL ↓ | MSE(V) ↓ |
|---|---|---|---|---|
| FLIP | TD-DD | 24.22 (0.37) | 1.595 (0.021) | 494.85 (221.39) |
| | TD²-DD | 108.16 (1.64) | 0.893 (0.065) | 103.71 (34.77) |
| | TD-CFM | 16.01 (0.33) | 1.120 (0.037) | 431.62 (64.40) |
| | TD-CFM(C) | 14.77 (0.38) | 0.704 (0.083) | 74.42 (13.13) |
| | TD²-CFM | 14.81 (0.56) | 0.546 (0.012) | 73.86 (26.41) |
| RUN | TD-DD | 21.28 (0.97) | 1.389 (0.005) | 53.28 (20.52) |
| | TD²-DD | 102.69 (3.60) | 0.546 (0.070) | 6.35 (0.88) |
| | TD-CFM | 14.99 (0.65) | 0.845 (0.085) | 209.80 (54.21) |
| | TD-CFM(C) | 13.01 (0.35) | 0.260 (0.089) | 32.84 (8.26) |
| | TD²-CFM | 13.20 (0.36) | 0.180 (0.076) | 34.61 (21.58) |
| SPIN | TD-DD | 21.31 (0.65) | 1.482 (0.015) | 1093.50 (700.34) |
| | TD²-DD | 103.72 (1.69) | 0.903 (0.067) | 115.58 (28.18) |
| | TD-CFM | 15.16 (0.53) | 1.020 (0.036) | 482.78 (24.82) |
| | TD-CFM(C) | 14.22 (0.06) | 0.605 (0.033) | 170.20 (48.23) |
| | TD²-CFM | 14.34 (0.20) | 0.449 (0.056) | 197.13 (26.98) |
| STAND | TD-DD | 21.34 (0.66) | 1.459 (0.029) | 594.94 (219.72) |
| | TD²-DD | 103.86 (4.22) | 0.630 (0.030) | 250.96 (79.14) |
| | TD-CFM | 14.28 (0.32) | 0.829 (0.107) | 1371.68 (326.61) |
| | TD-CFM(C) | 13.43 (0.34) | 0.335 (0.067) | 265.09 (12.84) |
| | TD²-CFM | 13.52 (0.61) | 0.284 (0.062) | 166.16 (17.51) |
| WALK | TD-DD | 19.30 (0.80) | 1.282 (0.033) | 622.04 (186.99) |
| | TD²-DD | 103.79 (3.80) | 0.471 (0.055) | 425.65 (131.20) |
| | TD-CFM | 14.97 (0.14) | 0.787 (0.070) | 1344.77 (149.38) |
| | TD-CFM(C) | 12.39 (0.25) | 0.165 (0.042) | 354.40 (114.40) |
| | TD²-CFM | 12.63 (0.46) | 0.078 (0.072) | 301.97 (21.93) |

### Quadruped

| Task | Method | EMD ↓ | NLL ↓ | MSE(V) ↓ |
|---|---|---|---|---|
| JUMP | TD-DD | 20.23 (1.67) | 1.394 (0.024) | 279.84 (165.15) |
| | TD²-DD | 135.62 (9.10) | 0.921 (0.044) | 562.83 (170.42) |
| | TD-CFM | 15.25 (0.02) | 0.960 (0.006) | 365.14 (177.15) |
| | TD-CFM(C) | 15.24 (0.13) | 0.548 (0.008) | 129.02 (23.63) |
| | TD²-CFM | 15.00 (0.08) | 0.369 (0.004) | 139.10 (9.66) |
| RUN | TD-DD | 20.06 (1.67) | 1.405 (0.013) | 273.65 (192.14) |
| | TD²-DD | 135.28 (9.10) | 0.909 (0.049) | 171.76 (48.29) |
| | TD-CFM | 15.04 (0.02) | 0.961 (0.031) | 189.56 (63.62) |
| | TD-CFM(C) | 14.92 (0.17) | 0.538 (0.017) | 84.74 (6.77) |
| | TD²-CFM | 14.71 (0.12) | 0.351 (0.008) | 90.48 (10.33) |
| STAND | TD-DD | 20.01 (1.78) | 1.401 (0.033) | 1131.49 (863.23) |
| | TD²-DD | 135.81 (9.19) | 0.875 (0.054) | 669.65 (148.88) |
| | TD-CFM | 14.91 (0.10) | 0.931 (0.033) | 735.43 (274.17) |
| | TD-CFM(C) | 14.75 (0.30) | 0.508 (0.017) | 336.02 (16.08) |
| | TD²-CFM | 14.49 (0.24) | 0.309 (0.015) | 325.59 (79.83) |
| WALK | TD-DD | 20.55 (1.93) | 1.412 (0.035) | 314.53 (84.50) |
| | TD²-DD | 136.45 (9.57) | 0.901 (0.056) | 256.91 (58.22) |
| | TD-CFM | 15.06 (0.22) | 0.949 (0.030) | 274.37 (58.65) |
| | TD-CFM(C) | 15.02 (0.09) | 0.518 (0.024) | 156.72 (41.37) |
| | TD²-CFM | 14.76 (0.19) | 0.331 (0.019) | 160.62 (23.36) |

### Pointmass Maze

| Task | Method | EMD ↓ | NLL ↓ | MSE(V) ↓ |
|---|---|---|---|---|
| FAST SLOW | TD-DD | 0.164 (0.013) | 2.012 (0.089) | 1642.91 (26.55) |
| | TD²-DD | 0.350 (0.038) | 0.637 (0.046) | 236.52 (58.27) |
| | TD-CFM | 0.082 (0.004) | 0.772 (0.065) | 575.00 (75.51) |
| | TD-CFM(C) | 0.061 (0.002) | 0.083 (0.013) | 93.04 (5.55) |
| | TD²-CFM | 0.060 (0.003) | 0.010 (0.059) | 61.08 (20.86) |
| LOOP | TD-DD | 0.151 (0.007) | 2.094 (0.119) | 537.80 (22.89) |
| | TD²-DD | 0.337 (0.040) | 0.659 (0.028) | 213.93 (51.75) |
| | TD-CFM | 0.088 (0.003) | 0.782 (0.018) | 225.96 (59.92) |
| | TD-CFM(C) | 0.070 (0.007) | 0.266 (0.066) | 86.12 (23.47) |
| | TD²-CFM | 0.065 (0.003) | 0.101 (0.074) | 102.65 (27.28) |
| REACH BOTTOM LEFT | TD-DD | 0.131 (0.008) | 1.969 (0.143) | 207.45 (38.79) |
| | TD²-DD | 0.339 (0.050) | 0.510 (0.043) | 89.56 (41.56) |
| | TD-CFM | 0.078 (0.005) | 0.659 (0.044) | 376.64 (67.43) |
| | TD-CFM(C) | 0.048 (0.002) | −0.099 (0.054) | 73.84 (7.92) |
| | TD²-CFM | 0.042 (0.002) | −0.261 (0.024) | 14.20 (0.47) |
| REACH BOTTOM LEFT LONG | TD-DD | 0.144 (0.005) | 2.037 (0.062) | 1239.65 (627.94) |
| | TD²-DD | 0.355 (0.042) | 0.807 (0.010) | 1431.05 (342.62) |
| | TD-CFM | 0.105 (0.004) | 0.987 (0.060) | 2212.63 (504.47) |
| | TD-CFM(C) | 0.078 (0.002) | 0.457 (0.058) | 993.60 (639.56) |
| | TD²-CFM | 0.074 (0.005) | 0.310 (0.083) | 896.15 (598.65) |
| REACH BOTTOM RIGHT | TD-DD | 0.180 (0.004) | 2.106 (0.114) | 194.15 (75.47) |
| | TD²-DD | 0.369 (0.053) | 0.618 (0.035) | 112.24 (12.17) |
| | TD-CFM | 0.096 (0.004) | 0.724 (0.055) | 272.13 (33.13) |
| | TD-CFM(C) | 0.070 (0.005) | 0.063 (0.043) | 103.89 (10.69) |
| | TD²-CFM | 0.067 (0.007) | −0.104 (0.024) | 49.28 (13.90) |
| REACH TOP LEFT | TD-DD | 0.122 (0.005) | 2.083 (0.149) | 433.81 (37.98) |
| | TD²-DD | 0.343 (0.036) | 0.631 (0.046) | 158.15 (35.48) |
| | TD-CFM | 0.076 (0.003) | 0.679 (0.086) | 453.53 (88.84) |
| | TD-CFM(C) | 0.051 (0.003) | 0.092 (0.071) | 54.48 (4.15) |
| | TD²-CFM | 0.052 (0.003) | 0.022 (0.047) | 31.01 (7.91) |
| REACH TOP RIGHT | TD-DD | 0.149 (0.004) | 1.994 (0.093) | 221.28 (26.98) |
| | TD²-DD | 0.350 (0.022) | 0.563 (0.121) | 69.97 (34.75) |
| | TD-CFM | 0.074 (0.004) | 0.700 (0.078) | 250.17 (36.85) |
| | TD-CFM(C) | 0.051 (0.000) | 0.032 (0.010) | 39.79 (5.50) |
| | TD²-CFM | 0.047 (0.002) | −0.131 (0.020) | 17.01 (6.82) |
| SQUARE | TD-DD | 0.175 (0.008) | 2.088 (0.105) | 826.61 (162.87) |
| | TD²-DD | 0.347 (0.030) | 0.906 (0.033) | 192.41 (36.20) |
| | TD-CFM | 0.093 (0.002) | 0.869 (0.026) | 281.43 (52.24) |
| | TD-CFM(C) | 0.077 (0.005) | 0.566 (0.027) | 210.94 (97.86) |
| | TD²-CFM | 0.075 (0.006) | 0.392 (0.043) | 186.53 (58.91) |

### Cheetah

| Task | Method | EMD ↓ | NLL ↓ | MSE(V) ↓ |
|---|---|---|---|---|
| FLIP | TD-DD | 16.97 (0.45) | 1.358 (0.033) | 903.42 (267.90) |
| | TD²-DD | 73.44 (9.89) | 0.782 (0.065) | 308.54 (36.36) |
| | TD-CFM | 13.06 (0.46) | 0.964 (0.073) | 911.92 (135.18) |
| | TD-CFM(C) | 10.96 (0.58) | 0.564 (0.050) | 328.99 (27.34) |
| | TD²-CFM | 10.95 (0.32) | 0.443 (0.047) | 222.71 (32.96) |
| FLIP BACKWARD | TD-DD | 18.64 (0.48) | 1.442 (0.052) | 678.24 (40.56) |
| | TD²-DD | 75.09 (6.07) | 0.753 (0.007) | 215.67 (39.77) |
| | TD-CFM | 12.83 (0.38) | 0.966 (0.020) | 381.99 (112.95) |
| | TD-CFM(C) | 11.36 (0.21) | 0.582 (0.005) | 230.92 (14.13) |
| | TD²-CFM | 11.06 (0.18) | 0.476 (0.028) | 255.25 (57.02) |
| RUN | TD-DD | 17.61 (0.43) | 1.489 (0.054) | 87.64 (30.75) |
| | TD²-DD | 73.06 (6.95) | 0.742 (0.039) | 111.78 (49.51) |
| | TD-CFM | 12.22 (0.30) | 1.103 (0.066) | 194.36 (19.68) |
| | TD-CFM(C) | 10.75 (0.17) | 0.535 (0.028) | 34.90 (21.47) |
| | TD²-CFM | 10.74 (0.07) | 0.445 (0.021) | 24.71 (10.91) |
| RUN BACKWARD | TD-DD | 18.75 (0.31) | 1.475 (0.036) | 57.65 (8.84) |
| | TD²-DD | 76.43 (7.28) | 0.802 (0.013) | 90.76 (20.35) |
| | TD-CFM | 12.59 (0.11) | 1.083 (0.041) | 82.43 (5.56) |
| | TD-CFM(C) | 11.78 (0.17) | 0.632 (0.008) | 30.50 (4.34) |
| | TD²-CFM | 11.53 (0.15) | 0.534 (0.013) | 33.52 (3.75) |
| WALK | TD-DD | 16.77 (0.46) | 1.461 (0.037) | 805.51 (158.64) |
| | TD²-DD | 72.44 (7.86) | 0.757 (0.020) | 348.70 (58.96) |
| | TD-CFM | 11.91 (0.18) | 1.095 (0.042) | 1899.15 (131.04) |
| | TD-CFM(C) | 10.72 (0.14) | 0.551 (0.024) | 277.74 (117.41) |
| | TD²-CFM | 10.66 (0.18) | 0.464 (0.029) | 260.01 (153.43) |
| WALK BACKWARD | TD-DD | 18.00 (1.11) | 1.427 (0.073) | 676.44 (296.20) |
| | TD²-DD | 75.66 (7.41) | 0.787 (0.036) | 447.90 (49.14) |
| | TD-CFM | 12.62 (0.28) | 1.056 (0.067) | 1489.41 (90.89) |
| | TD-CFM(C) | 11.60 (0.17) | 0.621 (0.030) | 591.06 (12.67) |
| | TD²-CFM | 11.41 (0.18) | 0.523 (0.021) | 585.86 (103.16) |

*Table 17.* **Per task** results for **planning with GPI**.

### Walker

| Domain | Method | Planner | Z-Distribution $D(\mathbb{Z})$ Random | Local Perturbation | Train Distribution |
|---|---|---|---|---|---|
| | FB | — | | 326.94 (7.00) | |
| | FB | GPI | 14.13 (0.51) | 14.06 (0.43) | 14.57 (0.33) |
| RUN | TD-DD | GPI | 328.61 (2.66) | 303.45 (44.57) | 292.96 (52.91) |
| | $TD^2$-DD | GPI | 316.41 (4.59) | 338.14 (14.23) | 349.77 (12.78) |
| | TD-CFM | GPI | 301.88 (17.94) | 221.07 (3.80) | 199.50 (4.09) |
| | TD-CFM(C) | GPI | 325.92 (14.31) | 368.97 (16.10) | 367.38 (11.48) |
| | $TD^2$-CFM | GPI | 325.76 (12.88) | 362.79 (19.09) | 358.73 (24.84) |
| | FB | — | | 338.41 (2.98) | |
| | FB | GPI | 29.32 (2.15) | 36.99 (4.28) | 39.09 (5.90) |
| SPIN | TD-DD | GPI | 281.76 (74.27) | 304.28 (65.49) | 299.92 (72.32) |
| | $TD^2$-DD | GPI | 287.70 (50.25) | 298.07 (51.86) | 298.78 (34.52) |
| | TD-CFM | GPI | 323.56 (32.38) | 323.90 (34.69) | 328.47 (15.90) |
| | TD-CFM(C) | GPI | 266.80 (80.22) | 251.21 (64.39) | 284.37 (73.57) |
| | $TD^2$-CFM | GPI | 287.26 (84.37) | 313.07 (33.89) | 320.22 (96.14) |
| | FB | — | | 852.55 (19.44) | |
| | FB | GPI | 79.24 (2.32) | 80.60 (3.28) | 82.58 (3.06) |
| STAND | TD-DD | GPI | 852.49 (26.17) | 806.55 (7.62) | 872.85 (19.73) |
| | $TD^2$-DD | GPI | 839.00 (22.97) | 914.47 (6.14) | 936.41 (11.77) |
| | TD-CFM | GPI | 823.10 (15.06) | 758.91 (19.84) | 846.08 (7.85) |
| | TD-CFM(C) | GPI | 858.42 (5.92) | 931.74 (12.01) | 947.69 (6.45) |
| | $TD^2$-CFM | GPI | 863.16 (4.15) | 923.77 (9.88) | 963.10 (6.69) |
| | FB | — | | 588.74 (5.30) | |
| | FB | GPI | 18.23 (0.68) | 18.36 (1.15) | 19.94 (0.87) |
| WALK | TD-DD | GPI | 587.76 (6.46) | 799.12 (11.85) | 667.74 (81.98) |
| | $TD^2$-DD | GPI | 594.48 (3.74) | 842.94 (28.16) | 852.17 (12.09) |
| | TD-CFM | GPI | 577.95 (2.89) | 793.47 (21.27) | 774.91 (55.37) |
| | TD-CFM(C) | GPI | 601.81 (8.58) | 883.26 (4.06) | 897.33 (10.88) |
| | $TD^2$-CFM | GPI | 596.08 (3.41) | 868.69 (14.42) | 868.46 (44.44) |

### Quadruped

| Domain | Method | Planner | Z-Distribution $D(\mathbb{Z})$ Random | Local Perturbation | Train Distribution |
|---|---|---|---|---|---|
| | FB | — | | 683.96 (2.09) | |
| | FB | GPI | 742.71 (1.01) | 746.48 (1.63) | 718.52 (2.65) |
| JUMP | TD-DD | GPI | 673.33 (6.07) | 690.13 (6.14) | 677.58 (5.71) |
| | $TD^2$-DD | GPI | 744.92 (0.69) | 750.42 (2.30) | 745.29 (1.12) |
| | TD-CFM | GPI | 748.19 (10.47) | 753.72 (0.58) | 745.93 (12.60) |
| | TD-CFM(C) | GPI | 790.56 (14.06) | 795.84 (16.14) | 785.20 (13.69) |
| | $TD^2$-CFM | GPI | 796.39 (13.27) | 800.34 (9.63) | 791.43 (11.66) |
| | FB | — | | 452.38 (3.25) | |
| | FB | GPI | 486.71 (0.64) | 488.23 (0.48) | 469.03 (2.35) |
| RUN | TD-DD | GPI | 484.45 (1.07) | 482.81 (2.55) | 482.53 (2.38) |
| | $TD^2$-DD | GPI | 485.26 (1.63) | 486.35 (0.93) | 484.89 (2.43) |
| | TD-CFM | GPI | 488.93 (1.08) | 488.45 (0.62) | 488.98 (0.28) |
| | TD-CFM(C) | GPI | 491.66 (2.75) | 490.89 (2.05) | 491.81 (2.14) |
| | $TD^2$-CFM | GPI | 488.89 (1.35) | 488.65 (1.19) | 489.31 (1.03) |
| | FB | — | | 896.43 (5.80) | |
| | FB | GPI | 975.01 (1.40) | 977.94 (0.76) | 938.44 (7.04) |
| STAND | TD-DD | GPI | 976.59 (2.78) | 976.75 (0.86) | 975.25 (2.49) |
| | $TD^2$-DD | GPI | 981.26 (1.56) | 981.59 (1.45) | 979.46 (0.93) |
| | TD-CFM | GPI | 982.08 (1.27) | 981.06 (0.26) | 981.29 (1.34) |
| | TD-CFM(C) | GPI | 984.03 (1.20) | 984.50 (1.49) | 983.33 (1.20) |
| | $TD^2$-CFM | GPI | 984.36 (0.25) | 985.52 (0.89) | 984.36 (1.21) |
| | FB | — | | 476.34 (4.71) | |
| | FB | GPI | 483.37 (1.05) | 483.73 (3.02) | 458.20 (6.62) |
| WALK | TD-DD | GPI | 497.55 (10.40) | 499.45 (11.65) | 494.38 (19.75) |
| | $TD^2$-DD | GPI | 457.54 (23.37) | 467.78 (4.58) | 452.44 (17.14) |
| | TD-CFM | GPI | 458.20 (29.01) | 466.62 (19.30) | 458.24 (30.28) |
| | TD-CFM(C) | GPI | 515.84 (5.84) | 519.36 (14.37) | 524.37 (1.56) |
| | $TD^2$-CFM | GPI | 516.67 (3.49) | 511.77 (3.70) | 517.82 (2.58) |

### Pointmass Maze

| Domain | Method | Planner | Z-Distribution $D(\mathbb{Z})$ Random | Local Perturbation | Train Distribution |
|---|---|---|---|---|---|
| | FB | — | | 223.85 (23.81) | |
| | FB | GPI | 1.67 (0.30) | 74.52 (2.24) | 1.24 (0.28) |
| FAST SLOW | TD-DD | GPI | 169.55 (74.06) | 363.47 (23.78) | 148.59 (43.53) |
| | $TD^2$-DD | GPI | 781.84 (1.20) | 769.02 (5.03) | 768.67 (11.19) |
| | TD-CFM | GPI | 254.07 (85.86) | 546.75 (191.17) | 359.50 (144.41) |
| | TD-CFM(C) | GPI | 763.24 (15.57) | 776.51 (12.37) | 769.87 (13.18) |
| | $TD^2$-CFM | GPI | 773.51 (2.71) | 773.81 (4.71) | 772.22 (3.11) |
| | FB | — | | 317.59 (8.55) | |
| | FB | GPI | 81.99 (5.01) | 315.10 (1.95) | 61.41 (3.58) |
| LOOP | TD-DD | GPI | 462.86 (5.90) | 430.51 (72.09) | 593.64 (56.15) |
| | $TD^2$-DD | GPI | 876.91 (9.21) | 889.03 (2.40) | 878.78 (2.43) |
| | TD-CFM | GPI | 832.91 (27.77) | 797.10 (57.17) | 852.81 (16.74) |
| | TD-CFM(C) | GPI | 873.85 (21.16) | 885.90 (4.21) | 875.45 (3.43) |
| | $TD^2$-CFM | GPI | 885.07 (2.79) | 885.50 (4.21) | 878.26 (0.64) |
| | FB | — | | 830.60 (0.63) | |
| | FB | GPI | 0.18 (0.17) | 127.90 (20.14) | 0.11 (0.10) |
| REACH BOTTOM LEFT | TD-DD | GPI | 781.69 (8.09) | 797.98 (3.52) | 795.12 (3.88) |
| | $TD^2$-DD | GPI | 823.28 (2.76) | 820.15 (1.89) | 824.00 (1.40) |
| | TD-CFM | GPI | 808.61 (7.06) | 801.97 (2.97) | 813.36 (6.35) |
| | TD-CFM(C) | GPI | 824.02 (0.73) | 824.17 (1.77) | 824.18 (3.84) |
| | $TD^2$-CFM | GPI | 827.85 (1.45) | 820.98 (3.63) | 828.45 (3.10) |
| | FB | — | | 49.31 (0.09) | |
| | FB | GPI | −464.55 (19.21) | 0.58 (1.79) | −401.26 (28.43) |
| REACH BOTTOM LEFT LONG | TD-DD | GPI | 461.30 (7.43) | 468.73 (26.94) | 252.28 (241.97) |
| | $TD^2$-DD | GPI | 609.10 (11.64) | 597.03 (6.46) | 668.76 (4.02) |
| | TD-CFM | GPI | 180.27 (35.66) | 311.59 (152.06) | 439.47 (230.80) |
| | TD-CFM(C) | GPI | 631.52 (11.58) | 614.90 (8.82) | 688.44 (4.05) |
| | $TD^2$-CFM | GPI | 646.67 (9.38) | 639.90 (13.22) | 691.68 (2.99) |
| | FB | — | | 366.39 (27.01) | |
| | FB | GPI | 0.00 (0.00) | 0.00 (0.00) | 0.00 (0.00) |
| REACH BOTTOM RIGHT | TD-DD | GPI | 343.62 (112.70) | 470.97 (42.25) | 398.94 (81.80) |
| | $TD^2$-DD | GPI | 360.71 (312.31) | 674.54 (7.35) | 529.97 (137.98) |
| | TD-CFM | GPI | 394.78 (159.98) | 356.58 (69.31) | 548.65 (73.62) |
| | TD-CFM(C) | GPI | 642.67 (6.59) | 686.08 (4.46) | 679.75 (2.98) |
| | $TD^2$-CFM | GPI | 534.62 (57.49) | 687.66 (1.75) | 641.45 (2.00) |
| | FB | — | | 895.88 (1.26) | |
| | FB | GPI | 351.72 (17.68) | 837.14 (2.07) | 185.50 (13.00) |
| REACH TOP LEFT | TD-DD | GPI | 941.32 (16.86) | 812.40 (152.88) | 920.44 (28.63) |
| | $TD^2$-DD | GPI | 940.90 (5.41) | 967.44 (3.52) | 939.49 (10.02) |
| | TD-CFM | GPI | 964.27 (0.34) | 948.83 (11.63) | 955.82 (9.32) |
| | TD-CFM(C) | GPI | 940.00 (29.18) | 967.03 (3.38) | 931.02 (18.53) |
| | $TD^2$-CFM | GPI | 943.43 (19.57) | 967.06 (1.42) | 940.05 (18.13) |
| | FB | — | | 715.25 (4.47) | |
| | FB | GPI | 0.72 (0.96) | 358.22 (20.05) | 1.35 (0.85) |
| REACH TOP RIGHT | TD-DD | GPI | 766.59 (6.78) | 771.64 (9.55) | 733.83 (44.76) |
| | $TD^2$-DD | GPI | 822.44 (1.74) | 818.06 (6.60) | 823.09 (1.76) |
| | TD-CFM | GPI | 777.94 (46.86) | 765.68 (41.55) | 754.73 (45.71) |
| | TD-CFM(C) | GPI | 826.30 (1.36) | 824.87 (2.61) | 821.51 (5.64) |
| | $TD^2$-CFM | GPI | 809.75 (28.90) | 824.23 (1.88) | 788.98 (45.69) |
| | FB | — | | 337.33 (9.46) | |
| | FB | GPI | 4.89 (1.03) | 148.93 (0.90) | 2.97 (0.82) |
| SQUARE | TD-DD | GPI | 585.01 (39.45) | 587.71 (46.77) | 451.92 (5.35) |
| | $TD^2$-DD | GPI | 896.45 (7.94) | 910.52 (2.63) | 878.19 (11.18) |
| | TD-CFM | GPI | 790.65 (3.06) | 843.76 (8.91) | 841.25 (22.73) |
| | TD-CFM(C) | GPI | 905.41 (1.13) | 920.06 (2.51) | 874.00 (10.23) |
| | $TD^2$-CFM | GPI | 901.82 (1.29) | 910.41 (1.66) | 866.85 (7.95) |

### Cheetah

| Domain | Method | Planner | Z-Distribution $D(\mathbb{Z})$ Random | Local Perturbation | Train Distribution |
|---|---|---|---|---|---|
| | FB | — | | 221.55 (44.79) | |
| | FB | GPI | 355.27 (5.95) | 356.52 (9.99) | 355.94 (5.10) |
| FLIP | TD-DD | GPI | 451.93 (81.15) | 445.10 (100.81) | 424.78 (100.74) |
| | $TD^2$-DD | GPI | 702.98 (27.77) | 712.72 (16.66) | 683.62 (35.04) |
| | TD-CFM | GPI | 355.69 (110.25) | 420.53 (184.00) | 341.40 (124.16) |
| | TD-CFM(C) | GPI | 724.85 (8.19) | 710.02 (4.51) | 711.16 (13.29) |
| | $TD^2$-CFM | GPI | 722.08 (7.50) | 718.74 (14.51) | 713.66 (14.14) |
| | FB | — | | 463.12 (5.73) | |
| | FB | GPI | 238.33 (9.74) | 388.33 (25.98) | 249.60 (5.64) |
| FLIP BACKWARD | TD-DD | GPI | 620.00 (69.42) | 596.45 (38.20) | 595.59 (34.96) |
| | $TD^2$-DD | GPI | 706.99 (8.08) | 690.83 (3.20) | 706.75 (8.34) |
| | TD-CFM | GPI | 545.12 (184.05) | 540.36 (186.74) | 492.55 (173.13) |
| | TD-CFM(C) | GPI | 727.23 (25.25) | 716.22 (29.49) | 711.11 (20.97) |
| | $TD^2$-CFM | GPI | 709.19 (16.76) | 684.33 (37.92) | 694.16 (15.24) |
| | FB | — | | 310.39 (35.44) | |
| | FB | GPI | 200.65 (4.44) | 301.34 (11.26) | 191.10 (6.56) |
| RUN | TD-DD | GPI | 436.74 (3.52) | 438.90 (4.92) | 434.94 (3.02) |
| | $TD^2$-DD | GPI | 427.15 (16.50) | 429.98 (13.04) | 421.92 (14.83) |
| | TD-CFM | GPI | 206.96 (45.56) | 243.53 (60.37) | 238.96 (66.97) |
| | TD-CFM(C) | GPI | 465.08 (2.50) | 470.44 (5.05) | 462.89 (3.15) |
| | $TD^2$-CFM | GPI | 462.71 (0.73) | 467.25 (14.78) | 454.90 (10.61) |
| | FB | — | | 201.07 (10.72) | |
| | FB | GPI | 5.31 (2.02) | 102.20 (5.73) | 19.11 (2.52) |
| RUN BACKWARD | TD-DD | GPI | 165.02 (4.50) | 246.72 (12.09) | 325.40 (0.86) |
| | $TD^2$-DD | GPI | 224.90 (21.33) | 310.10 (22.82) | 322.33 (4.05) |
| | TD-CFM | GPI | 90.83 (28.26) | 92.46 (15.59) | 49.88 (29.15) |
| | TD-CFM(C) | GPI | 222.14 (36.05) | 342.15 (2.02) | 333.90 (3.00) |
| | $TD^2$-CFM | GPI | 252.70 (10.86) | 319.46 (35.05) | 332.21 (0.77) |
| | FB | — | | 792.89 (52.74) | |
| | FB | GPI | 830.00 (15.20) | 889.84 (5.00) | 733.11 (34.27) |
| WALK | TD-DD | GPI | 977.30 (3.13) | 978.74 (2.47) | 979.48 (3.47) |
| | $TD^2$-DD | GPI | 959.18 (30.39) | 955.97 (25.64) | 956.79 (29.06) |
| | TD-CFM | GPI | 767.47 (96.47) | 805.68 (104.96) | 853.73 (117.82) |
| | TD-CFM(C) | GPI | 985.04 (0.10) | 985.06 (0.29) | 984.90 (0.18) |
| | $TD^2$-CFM | GPI | 984.21 (0.03) | 984.46 (0.09) | 984.23 (0.07) |
| | FB | — | | 897.16 (32.19) | |
| | FB | GPI | 22.40 (10.18) | 373.19 (13.71) | 78.60 (18.72) |
| WALK BACKWARD | TD-DD | GPI | 793.32 (52.67) | 946.70 (12.57) | 982.37 (0.21) |
| | $TD^2$-DD | GPI | 951.82 (11.09) | 981.74 (0.27) | 982.45 (0.07) |
| | TD-CFM | GPI | 455.18 (190.16) | 456.19 (140.79) | 257.55 (173.06) |
| | TD-CFM(C) | GPI | 964.75 (4.54) | 981.93 (0.26) | 982.89 (0.25) |
| | $TD^2$-CFM | GPI | 962.41 (4.32) | 982.08 (0.16) | 982.64 (0.05) |

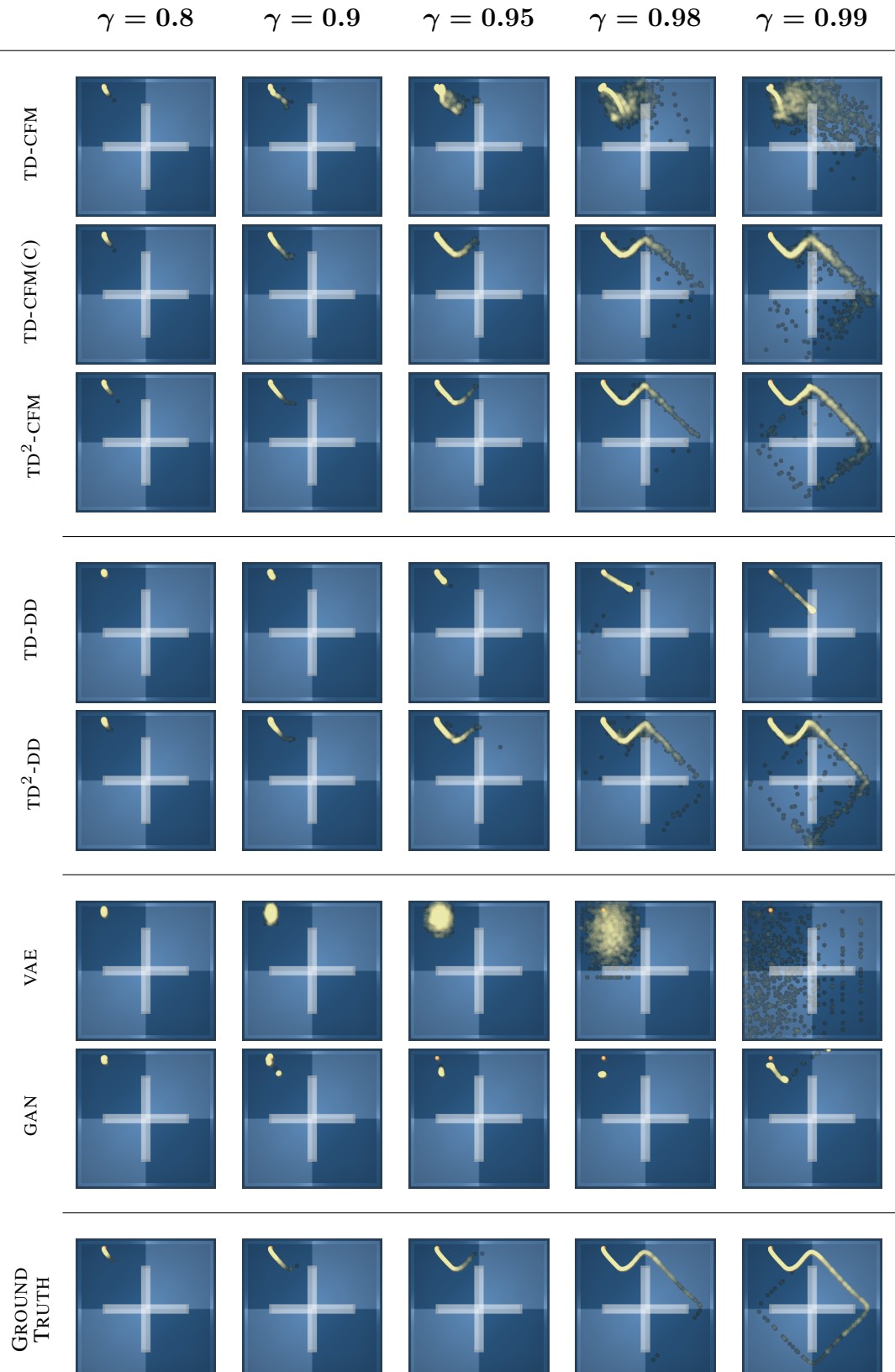

*Figure 5.* Qualitative samples generated with TD-CFM, TD-DD, VAE, and GAN methods for various discount factors $\gamma$ on the LOOP task in the POINTMASS MAZE domain. The last row depicts ground truth discounted occupancies.

# E. Theoretical Results

## E.1. Proofs of Main Results

**Lemma 1.** *Let $\vec{p}_t$ be a probability path for $P$ generated by vector field $\vec{v}_t$ and $\widehat{p}_t^{(n)}$ be a probability path for $P^\pi m_1^{(n)}$ generated by $\widehat{v}_t^{(n)}$ such that $\vec{p}_0 = \widehat{p}_0^{(n)} = m_0$. For any $t \in [0,1]$ and $(s,a)$ let $v_t^{(n+1)}(\cdot \mid s,a)$ be the solution of*[4]

$$\underset{v:\mathbb{R}^d \to \mathbb{R}^d}{\arg\min} (1-\gamma)\mathbb{E}_{\vec{X}_t \sim \vec{p}_t(\cdot|s,a)}\left[\left\|v(\vec{X}_t) - \vec{v}_t(\vec{X}_t \mid s,a)\right\|^2\right]$$

$$+ \gamma\mathbb{E}_{\widehat{X}_t \sim \widehat{p}_t^{(n)}(\cdot|s,a)}\left[\left\|v(\widehat{X}_t) - \widehat{v}_t^{(n)}(\widehat{X}_t \mid s,a)\right\|^2\right].$$

*Then $v_t^{(n+1)}$ induces a probability path $m_t^{(n+1)}$ such that $m_0^{(n+1)} = m_0$ and $m_1^{(n+1)} = \mathcal{T}^\pi m_1^{(n)}$.*

*Proof.* By Lemma 4, we have that

$$v_t^{(n+1)}(x \mid s,a) = \frac{(1-\gamma)\vec{p}_t(x|s,a)\vec{v}_t(x \mid s,a) + \gamma\widehat{p}_t^{(n)}(x|s,a)\widehat{v}_t^{(n)}(x \mid s,a)}{m_t^{(n+1)}(x|s,a)},$$

where $m_t^{(n+1)}(x|s,a) = (1-\gamma)\vec{p}_t(x|s,a) + \gamma\widehat{p}_t^{(n)}(x|s,a)$. Lemma 3 implies that $m_t^{(n+1)}$ is the probability path generated by $v_t^{(n+1)}$. It is easy to see that $m_0^{(n+1)} = m_0$ since $\vec{p}_0 = \widehat{p}_0^{(n)} = m_0$. Moreover, since $\vec{p}_1 = P$ and $\widehat{p}_1^{(n)} = P^\pi m_1^{(n)}$ by assumption, $m_1^{(n+1)} = (1-\gamma)P + \gamma P^\pi m_1^{(n)} = \mathcal{T}^\pi m_1^{(n)}$, which proves the result. $\qquad\square$

**Theorem 1.** *For any $n \geq 1$, the probability paths generated by TD-CFM, TD-CFM(C), or TD$^2$-CFM satisfy*

$$m_t^{(n+1)}(x \mid s,a) = \left(\mathcal{B}_t^\pi m_t^{(n)}\right)(x \mid s,a), \quad \forall t \in [0,1]$$

*where $\mathcal{B}_t^\pi m := (1-\gamma)P_t + \gamma P^\pi m$ and $P_t(x|s,a) := \int p_{t|1}(x \mid x_1)P(x_1|s,a)\mathrm{d}x_1$. For any $t \in [0,1]$, the operator $\mathcal{B}_t^\pi$ is a $\gamma$-contraction in 1-Wasserstein distance, that is, for any couple of probability paths $p_t, q_t$,*

$$\sup_{s,a} W_1\left((\mathcal{B}_t^\pi p_t)(\cdot \mid s,a), (\mathcal{B}_t^\pi q_t)(\cdot \mid s,a)\right)$$

$$\leq \gamma \sup_{s,a} W_1\left(p_t(\cdot \mid s,a), q_t(\cdot \mid s,a)\right).$$

*Proof.* To prove that the iterates of the three algorithms satisfy a Bellman-like update through the operator $\mathcal{B}_t^\pi$ we only need to apply Proposition 3 for TD$^2$-CFM, Theorem 5 for TD-CFM, and Theorem 6 for TD-CFM(C). That $\mathcal{B}_t$ is a $\gamma$-contraction in 1-Wasserstein distance can be seen by applying Theorem 4 with $k = 1$. $\qquad\square$

**Corollary 1.** *Let $\{m_t^{(n)}\}_{n\geq 0}$ be the sequence of probability paths produced by TD-CFM, TD-CFM(C), or TD$^2$-CFM starting from an arbitrary vector field $v_t^{(0)}$. Then,*

$$\lim_{n\to\infty} m_t^{(n)} = \overline{m}_t = \mathcal{B}_t \overline{m}_t,$$

*where $\overline{m}_t$ is the unique fixed point of $\mathcal{B}_t$, and $\overline{m}_t = m_t^{\mathrm{MC}}$, where $m_t^{\mathrm{MC}}(\cdot \mid s,a) = \int p_{t|1}(\cdot \mid x_1)m^\pi(x_1 \mid s,a)$ is the probability path of the Monte-Carlo approach (MC-CFM; 6).*

*Proof.* That $\mathcal{B}_t^\pi$ has a unique fixed point $\bar{m}_t$ to which every sequence $m_t^{(n)}$ converges to is a consequence of the Banach fixed point theorem applied on the space of all probability paths $m_t : \mathsf{S} \times \mathsf{A} \to \mathscr{P}(\mathbb{R}^d)$ equipped with the sup-1-Wasserstein metric. By inspecting the definition of $\mathcal{B}_t^\pi$, it is easy to see that $\bar{m}_t = (I - \gamma P^\pi)^{-1}P_t$. Since $P_t(x|s,a) = \int p_{t|1}(x|x_1)P(x_1|s,a)\mathrm{d}x_1$,

$$\bar{m}_t(x|s,a) = [(I - \gamma P^\pi)^{-1}P_t](x|s,a) = \int p_{t|1}(x|x_1)\underbrace{[(I - \gamma P^\pi)^{-1}P](x_1|s,a)}_{=m^\pi(x_1|s,a)}\mathrm{d}x_1 = m_t^{\mathrm{MC}}(x|s,a).$$

$\qquad\square$

**Theorem 2.** *For any $n \geq 1$ and $t \in [0, 1]$, assume that $m_t^{(n)}(x \mid s, a) = \int p_{t|1}(x \mid x_1) m_1^{(n)}(x_1 \mid s, a) \mathrm{d}x_1$, then*

$$\sigma^2_{\text{TD-CFM}} = \sigma^2_{\text{TD}^2\text{-CFM}} \; +$$
$$\gamma^2 \, \mathbb{E}\left[ \operatorname{Tr}\left( \operatorname{Cov}_{X_1|s,a,X_t}\left[ \nabla_\theta v_t(X_t|s, a; \theta)^\top u_{t|1}(X_t|X_1) \right] \right) \right].$$

*Proof.* See Theorem 7. $\qquad\square$

**Theorem 3.** *For any $n \geq 1$ and $t \in [0, 1]$, assume that $m_t^{(n)}(x \mid s, a) = \int p_{t|0,1}(x \mid x_0, x_1) m_{0,1}^{(n)}(x_0, x_1 \mid s, a) \mathrm{d}x_0 \mathrm{d}x_1$[5], then we obtain*

$$\sigma^2_{\text{TD-CFM(C)}} = \sigma^2_{\text{TD}^2\text{-CFM}} \; +$$
$$\gamma^2 \mathbb{E}\left[ \operatorname{Tr}\left( \operatorname{Cov}_{Z|S,A,X_t}\left[ \nabla_\theta v_t(X_t|S, A; \theta)^\top u_{t|Z}(X_t|Z) \right] \right) \right],$$

*where $Z = (X_0, X_1)$. Furthermore, if we use straight conditional paths, i.e., $X_t = tX_1 + (1 - t)X_0$, and the linear interpolant $X_t$ does not intersect for any $s, a, s'$, then $\sigma^2_{\text{TD-CFM(C)}} = \sigma^2_{\text{TD}^2\text{-CFM}}$.*

*Proof.* See Theorem 8. $\qquad\square$

### E.2. General Results

**Lemma 3.** *Let $v_t^1$ and $v_t^2$ be vector fields that generate the probability paths $p_t^1$ and $p_t^2$, respectively. Then, for any $\gamma \in [0, 1]$, the mixture probability path $p_t = (1 - \gamma)p_t^1 + \gamma p_t^2$ is generated by the vector field*

$$v_t := \frac{(1 - \gamma)p_t^1 v_t^1 + \gamma p_t^2 v_t^2}{(1 - \gamma)p_t^1 + \gamma p_t^2}. \tag{27}$$

*Proof.* Since $v_1^t$ (resp. $v_2^t$) generates $p_t^1$ (resp. $p_t^2$), we know from the continuity equation that:

$$\frac{\partial p_t^1}{\partial t} = \operatorname{div}(p_t^1 v_t^1), \quad \frac{\partial p_t^2}{\partial t} = \operatorname{div}(p_t^2 v_t^2),$$

where div denotes the divergence operator. Then, by linearity of div,

$$\begin{aligned}
\frac{\partial p_t}{\partial t} &= \frac{\partial \left( (1 - \gamma)p_t^1 + \gamma p_t^2 \right)}{\partial t} \\
&= (1 - \gamma)\operatorname{div}(p_t^1 v_t^1) + \gamma \operatorname{div}(p_t^2 v_t^2) \\
&= \operatorname{div}\left( (1 - \gamma)p_t^1 v_t^1 + \gamma p_t^2 v_t^2 \right) \\
&= \operatorname{div}\left( \frac{(1 - \gamma)p_t^1 v_t^1 + \gamma p_t^2 v_t^2}{(1 - \gamma)p_t^1 + \gamma p_t^2} \left( (1 - \gamma)p_t^1 + \gamma p_t^2 \right) \right) \\
&= \operatorname{div}\left( \frac{(1 - \gamma)p_t^1 v_t^1 + \gamma p_t^2 v_t^2}{(1 - \gamma)p_t^1 + \gamma p_t^2} p_t \right) \\
&= \operatorname{div}(v_t p_t).
\end{aligned}$$

Hence, $(v_t, p_t)$ satisfies the continuity equation, which implies that $v_t$ generates $p_t$. $\qquad\square$

**Lemma 4.** *Let $v_t^1$ and $v_t^2$ be vector fields that generate the probability paths $p_t^1$ and $p_t^2$, respectively. For $\gamma \in [0, 1]$, the vector field $v_t = \frac{(1-\gamma)p_t^1 v_t^1 + \gamma p_t^2 v_t^2}{(1-\gamma)p_t^1 + \gamma p_t^2}$ satisfies*

$$v_t = \underset{v:\mathbb{R}^d \to \mathbb{R}^d}{\arg\min} \left\{ (1 - \gamma) \, \mathbb{E}_{x_t \sim p_t^1}\left[ \|v_t(x_t) - v_t^1(x_t)\|^2 \right] + \gamma \, \mathbb{E}_{x_t \sim p_t^2}\left[ \|v_t(x_t) - v_t^2(x_t)\|^2 \right] \right\}.$$

*Proof.* Let $\ell_t(v) := (1 - \gamma) \, \mathbb{E}_{x_t \sim p_t^1}\left[ \|v_t(x_t) - v_t^1(x_t)\|^2 \right] + \gamma \, \mathbb{E}_{x_t \sim p_t^2}\left[ \|v_t(x_t) - v_t^2(x_t)\|^2 \right]$. The functional derivative of this quantity wrt $v$ evaluated at some point $x$ is

$$\nabla_v \ell_t(v)(x) = (1 - \gamma)p_1^t(x)(v_t(x) - v_t^1(x)) + \gamma p_2^t(x)(v_t(x) - v_t^2(x)).$$

Setting this to zero and solving for $v_t(x)$ yields the result. $\qquad\square$

### E.3. Analysis of TD$^2$-CFM

We study the learning dynamics of an idealized variant of TD$^2$-CFM which minimizes the flow-matching loss exactly. Starting from an arbitrary vector field $v_t^{(0)}$, at each iteration $n \geq 0$ we compute

$$v_t^{(n+1)}(\cdot|s,a) \in \underset{v:\mathbb{R}^d \to \mathbb{R}^d}{\arg\min}\, \ell_{\text{TD}^2\text{-CFM}}^{(n)}(t,s,a), \tag{28}$$

where

$$\ell_{\text{TD}^2\text{-CFM}}^{(n)}(t,s,a) := (1-\gamma)\vec{\ell}(t,s,a) + \gamma\widehat{\ell}(t,s,a)$$

$$\vec{\ell}(t,s,a) := \mathbb{E}_{S'\sim P(\cdot|s,a),X_t\sim p_{t|1}(\cdot|S')}\left[\left\|v(X_t|s,a) - u_t(X_t|S')\right\|^2\right]$$

$$\widehat{\ell}(t,s,a) := \mathbb{E}_{S'\sim P(\cdot|s,a),X_t\sim m_t^{(n)}(\cdot|s',\pi(s'))}\left[\left\|v(X_t|s,a) - v_t^{(n)}(X_t|S',\pi(S'))\right\|^2\right],$$

and $m_t^{(n)}(x|s,a)$ is the probability path generated by $v_t^{(n)}(x|s,a)$.

**Lemma 5.** *For any $n \geq 0$, the vector field minimizing (28) is*

$$v_t^{(n+1)}(x \mid s,a) =$$
$$\frac{(1-\gamma)\int u_{t|1}(x \mid x_1)p_{t|1}(x \mid x_1)P(x_1|s,a)\mathrm{d}x_1 + \gamma\mathbb{E}_{S'\sim P(\cdot|s,a)}[m_t^{(n)}(x|S',\pi(S'))v_t^{(n)}(x|S',\pi(S'))]}{m_t^{(n+1)}(x|s,a)}$$

*where we define $m_t^{(n+1)}(x|s,a) := (1-\gamma)P_t(x|s,a) + \gamma\mathbb{E}_{S'\sim P(\cdot|s,a)}[m_t^{(n)}(x|S',\pi(S'))]$ and $P_t(x|s,a) := \int p_{t|1}(x \mid x_1)P(x_1|s,a)\mathrm{d}x_1$. Moreover $v_t^{(n+1)}$ generates $m_t^{(n+1)}$.*

*Proof.* By Theorem 2 of (Lipman et al., 2023), we have for the first term in $\ell_{\text{TD}^2\text{-CFM}}$

$$\nabla_\theta\vec{\ell}(t,s,a) = \nabla_\theta\mathbb{E}_{X_t\sim P_t(\cdot|s,a)}\left[\left\|v_t(X_t|s,a) - \vec{v}_t(X_t|s,a)\right\|^2\right],$$

where $P_t(x|s,a) := \int p_{t|1}(x \mid x_1)P(x_1|s,a)\mathrm{d}x_1$, $\vec{v}_t(x|s,a) = \frac{\int u_{t|1}(x|x_1)p_{t|1}(x|x_1)P(x_1|s,a)\mathrm{d}x_1}{P_t(x|s,a)}$. Similarly, we have for the second term:

$$\nabla_\theta\widehat{\ell}(t,s,a) = \nabla_\theta\mathbb{E}_{X_t\sim\widehat{p}_t^{(n)}(\cdot|s,a)}\left[\left\|v_t(X_t|s,a) - \widehat{v}_t(X_t|s,a)\right\|^2\right],$$

where $\widehat{p}_t^{(n)} = P^\pi m_t^{(n)}$ and $\widehat{v}_t = \frac{P^\pi(m_t^{(n)}v_t^{(n)})}{P^\pi m_t^{(n)}}$.

Therefore, $\ell_{\text{TD-CFM}}^{(n)}(t,s,a)$ is equivalent, in term of gradient, to a mixture of two marginal flow-matching losses, which implies that $v_t^{(n+1)}$ has the stated expression by Lemma 4. The fact that it generates $m_t^{(n+1)}$ is a consequence of Lemma 3. $\square$

We then define the following operator to characterize the iterates of TD$^2$-CFM.

**Definition 1** (Bellman operator for probability paths)**.** *For any $t \in [0,1]$, we define the operator $\mathcal{B}_t^\pi m := (1-\gamma)P_t + \gamma P^\pi m$, where $P_t(x|s,a) := \int p_{t|1}(x \mid x_1)P(x_1|s,a)\mathrm{d}x_1$.*

The following observation is then immediate from Lemma 5.

**Proposition 3.** *For any $n \geq 0$, the probability path generated by TD$^2$-CFM satisfies $m_t^{(n+1)}(x|s,a) = (\mathcal{B}_t^\pi m_t^{(n)})(x \mid s,a)$, where $\mathcal{B}_t^\pi$ is the operator of Definition 1. Moreover, $m_1^{(n+1)}(x|s,a) = (\mathcal{T}^\pi m_1^{(n)})(x \mid s,a)$.*

**Theorem 4.** *For any $t \in [0,1]$, the operator $\mathcal{B}_t^\pi$ of Definition 1 is a $\gamma^{1/k}$-contraction in Wasserstein k-distance, i.e., for any couple of probability paths $p_t, q_t$ and $k \in [1,\infty)$,*

$$\sup_{s,a} W_k\left((\mathcal{B}_t^\pi p_t)(\cdot \mid s,a), (\mathcal{B}_t^\pi q_t)(\cdot \mid s,a)\right) \leq \gamma^{1/k}\sup_{s,a} W_k\left(p_t(\cdot \mid s,a), q_t(\cdot \mid s,a)\right).$$

*Proof.* Recall that the Wasserstein k-distance between $p_t$ and $q_t$ induced by a metric $d$ is defined as

$$W_k(p_t(\cdot|s,a), q_t(\cdot|s,a)) := \inf_{\Gamma(\cdot|s,a) \in \mathcal{C}(p_t(\cdot|s,a), q_t(\cdot|s,a))} \mathbb{E}_{(X,Y) \sim \Gamma(\cdot|s,a)}[d(X,Y)^k]^{1/k},$$

where $\mathcal{C}(p_t(\cdot|s,a), q_t(\cdot|s,a))$ is the set of all couplings between the two measures. Now take any coupling $\tilde{\Gamma}(\cdot|s,a) \in \mathcal{C}(p_t(\cdot|s,a), q_t(\cdot|s,a))$ for any $s,a$. Then, the following quantity

$$\Theta(x,y|s,a) = (1-\gamma)P(x|s,a)\delta(x-y) + \gamma(P^\pi\tilde{\Gamma})(x,y|s,a)$$

is a valid coupling between $(\mathcal{B}_t^\pi p_t)(\cdot \mid s,a)$ and $(\mathcal{B}_t^\pi q_t)(\cdot \mid s,a)$. In fact,

$$\int \Theta(x,y|s,a)\mathrm{d}x = (1-\gamma)\int P(x|s,a)\delta(x-y)\mathrm{d}x + \gamma\int (P^\pi\tilde{\Gamma})(x,y \mid s,a)\mathrm{d}x$$

$$= (1-\gamma)P(y|s,a) + \gamma\int \mathbb{E}_{s'\sim P(\cdot|s,a)}\Big[\tilde{\Gamma}(x,y|s',\pi(s'))\Big]\mathrm{d}x$$

$$= (1-\gamma)P(y|s,a) + \gamma\,\mathbb{E}_{s'\sim P(\cdot|s,a)}\Big[\int \tilde{\Gamma}(x,y|s',\pi(s'))\mathrm{d}x\Big]$$

$$= (1-\gamma)P(y|s,a) + \gamma\,\mathbb{E}_{s'\sim P(\cdot|s,a)}\big[q_t(y|s',\pi(s'))\big]$$

$$= (\mathcal{T}^\pi q_t)(y|s,a).$$

Analogously, we can prove that $\int \Theta(x,y|s,a)\mathrm{d}y = (\mathcal{B}^\pi p_t)(x|s,a)$. Then,

$$W_k\big((\mathcal{B}_t^\pi p_t)(\cdot \mid s,a), (\mathcal{B}_t^\pi q_t)(\cdot \mid s,a)\big) = \inf_{\Gamma(\cdot|s,a) \in \mathcal{C}([\mathcal{L}_t^\pi p_t](\cdot|s,a), [\mathcal{L}_t^\pi q_t](\cdot|s,a))} \mathbb{E}_{(X,Y)\sim\Gamma(\cdot|s,a)}[d(X,Y)^k]^{1/k}$$

$$\leq \mathbb{E}_{(X,Y)\sim\Theta(\cdot|s,a)}[d(X,Y)^k]^{1/k}$$

$$= \Big((1-\gamma)\mathbb{E}_{(X\sim P(\cdot|s,a), Y\sim\delta_X)}[d(X,Y)^k] + \gamma\mathbb{E}_{(X,Y)\sim[P^\pi\tilde{\Gamma}](\cdot|s,a)}[d(X,Y)^k]\Big)^{1/k}$$

$$= \gamma^{1/k}\mathbb{E}_{s'\sim P(\cdot|s,a), (X,Y)\sim\tilde{\Gamma}(\cdot|s',\pi(s'))}[d(X,Y)^k]^{1/k}.$$

Since this holds for any coupling $\tilde{\Gamma}(\cdot|s,a) \in \mathcal{C}(p_t(\cdot|s,a), q_t(\cdot|s,a))$, we can take the infimum over all such couplings on the right-hand side, so that

$$W_k\big((\mathcal{B}_t^\pi p_t)(\cdot \mid s,a), (\mathcal{B}_t^\pi q_t)(\cdot \mid s,a)\big) \leq \gamma^{1/k}\left(\mathbb{E}_{s'\sim P(\cdot|s,a)}\left[\inf_{\Gamma\in\mathcal{C}(p_t(\cdot|s',\pi(s')), q_t(\cdot|s',\pi(s')))}\mathbb{E}_{(X,Y)\sim\Gamma}[d(X,Y)^k]\right]\right)^{1/k}$$

$$= \gamma^{1/k}\big(\mathbb{E}_{s'\sim P(\cdot|s,a)}\big[W_k(p_t(\cdot|s',\pi(s')), q_t(\cdot|s',\pi(s')))^k\big]\big)^{1/k}$$

$$\leq \gamma^{1/k}\sup_{s,a} W_k(p_t(\cdot \mid s,a), q_t(\cdot \mid s,a)).$$

Taking the supremum over $(s,a)$ of the left-hand side concludes the proof. $\square$

### E.4. Analysis of TD-CFM

We study the learning dynamics of an idealized variant of TD-CFM which minimizes the flow-matching loss exactly. Starting from an arbitrary vector field $v_t^{(0)}$, at each iteration $n \geq 0$ we compute

$$v_t^{(n+1)}(\cdot|s,a) \in \underset{v_t(\cdot):\mathbb{R}^d\to\mathbb{R}^d}{\arg\min}\ \ell_{\text{TD-CFM}}^{(n)}(t,s,a) := \mathbb{E}_{X_1\sim(\mathcal{T}^\pi m_1^{(n)})(s,a), X_t\sim p_{t|1}(\cdot|X_1)}\Big[\|v_t(X_t) - u_{t|1}(X_t|X_1)\|^2\Big], \quad (29)$$

where $m_t^{(n)}(x|s,a)$ is the probability path generated by $v_t^{(n)}(x|s,a)$.

**Lemma 6.** *For any $n \geq 0$, the vector field minimizing (29) is*

$$v_t^{(n+1)}(x \mid s,a) = \int u_{t|1}(x|x_1)\frac{p_{t|1}(x \mid x_1)(\mathcal{T}^\pi m_1^{(n)})(x_1 \mid s,a)}{m_t^{(n+1)}(x|s,a)}\mathrm{d}x_1,$$

*where $m_t^{(n+1)}(x|s,a) := \int p_{t|1}(x \mid x_1)(\mathcal{T}^\pi m_1^{(n)})(x_1 \mid s,a)\mathrm{d}x_1$. Moreover $v_t^{(n+1)}$ generates $m_t^{(n+1)}$.*

*Proof.* Note that (29) is a standard flow matching loss for the target distribution $\mathcal{T}^\pi m_1^{(n)}$. The expression of $v_t^{(n+1)}(x \mid s, a)$ given in the statement is exactly the vector field obtained by marginalization of the conditional vector field $u_{t|1}$, which we know to be the minimizer of the loss from Theorem 2 of (Lipman et al., 2023). The fact that $v_t^{(n+1)}$ generates $m_t^{(n+1)}$ is a consequence of Theorem 1 of (Lipman et al., 2023). $\square$

**Lemma 7.** *For any $n \geq 0$, the probability path generated by (29) satisfies $m_1^{(n+1)}(x|s, a) = \left(\mathcal{T}^\pi m_1^{(n)}\right)(x|s, a)$.*

*Proof.* This is immediate from the definition of conditional probability path, as we set $p_{1|1}(x \mid x_1) = \delta(x - x_1)$ by construction, where $\delta(\cdot)$ is the Dirac's delta function. $\square$

**Theorem 5.** *For any $n \geq 1$, the probability path generated by (29) satisfies*

$$m_t^{(n+1)}(x|s, a) = \left(\mathcal{B}_t^\pi m_t^{(n)}\right)(x|s, a),$$

*where $\mathcal{B}_t^\pi$ is the operator of Definition 1. Moreover, if the initial vector field $v_t^{(0)}$ satisfies*

$$v_t^{(0)}(x \mid s, a) = \int u_{t|1}(x|x_1) \frac{p_{t|1}(x \mid x_1) m_1^{(0)}(x_1 \mid s, a)}{m_t^{(0)}(x|s, a)} \mathrm{d}x_1,$$

*with $m_t^{(0)}$ being its generated proability path, then this result is valid at all $n \geq 0$.*

*Proof.* We know that, for all $n \geq 0$, $v_t^{n+1}$ generates $m_t^{(n+1)}$ (Lemma 6) and that $m_1^{(n+1)} = \mathcal{T}^\pi m_1^{(n)}$ (Lemma 7). Note that $m_t^{(n+1)}$ is written as a function of $m_1^{(n)}$ only, i.e., at each iteration we keep only the distribution generated at time $t = 1$ ($m_1^{(n)}$) and discard the associated probability path ($m_t^{(n)}$ for $t < 1$). We can however express $m_t^{(n+1)}$ as a function of $m_t^{(n)}$ thanks to the linearity of the Bellman operator and the definition of marginal paths. For any $n \geq 1$,

$$
\begin{aligned}
m_t^{(n+1)}(x \mid s, a) &:= \int p_{t|1}(x \mid x_1)\left(\mathcal{T}^\pi m_1^{(n)}\right)(x_1 \mid s, a)\mathrm{d}x_1 \\
&= \int p_{t|1}(x \mid x_1)\left((1-\gamma)P(x_1 \mid s, a) + \gamma\, \mathbb{E}_{s' \sim P(\cdot|s,a)}\left[m_1^{(n)}(x_1 \mid s', \pi(s'))\right]\right)\mathrm{d}x_1 \\
&= (1-\gamma)\int p_{t|1}(x \mid x_1)P(x_1 \mid s, a)\mathrm{d}x_1 + \gamma\, \mathbb{E}_{s' \sim P(\cdot|s,a)}\left[\int p_{t|1}(x \mid x_1)m_1^{(n)}(x_1 \mid s', \pi(s'))\mathrm{d}x_1\right] \\
&= (1-\gamma)\int p_{t|1}(x \mid x_1)P(x_1 \mid s, a)\mathrm{d}x_1 + \gamma\, \mathbb{E}_{s' \sim P(\cdot|s,a)}\left[\int p_{t|1}(x \mid x_1)\left(\mathcal{T}^\pi m_1^{(n-1)}\right)(x_1 \mid s', \pi(s'))\mathrm{d}x_1\right] \\
&= (1-\gamma)\int p_{t|1}(x \mid x_1)P(x_1 \mid s, a)\mathrm{d}x_1 + \gamma\, \mathbb{E}_{s' \sim P(\cdot|s,a)}\left[m_t^{(n)}(x \mid s', \pi(s'))\right] \\
&= (1-\gamma)P_t(x|s, a) + \gamma\, \mathbb{E}_{s' \sim P(\cdot|s,a)}\left[m_t^{(n)}(x \mid s', \pi(s'))\right] = \left(\mathcal{B}_t^\pi m_t^{(n)}\right)(x \mid s, a).
\end{aligned}
$$

This proves the first part of the statement. For the second part, we only need to prove that the result also holds at $n = 0$. Note that the assumption on $v_t^{(0)}$ implies that $m_t^{(0)}(x \mid s, a) := \int p_{t|1}(x \mid x_1)m_1^{(0)}(x_1 \mid s, a)\mathrm{d}x_1$. Thus,

$$
\begin{aligned}
m_t^{(1)}(x \mid s, a) &:= \int p_{t|1}(x \mid x_1)\left(\mathcal{T}^\pi m_1^{(0)}\right)(x_1 \mid s, a)\mathrm{d}x_1 \\
&= \int p_{t|1}(x \mid x_1)\left((1-\gamma)P(x_1 \mid s, a) + \gamma\, \mathbb{E}_{s' \sim P(\cdot|s,a)}\left[m_1^{(0)}(x_1 \mid s', \pi(s'))\right]\right)\mathrm{d}x_1 \\
&= (1-\gamma)\int p_{t|1}(x \mid x_1)P(x_1 \mid s, a)\mathrm{d}x_1 + \gamma\, \mathbb{E}_{s' \sim P(\cdot|s,a)}\left[\int p_{t|1}(x \mid x_1)m_1^{(0)}(x_1 \mid s', \pi(s'))\mathrm{d}x_1\right] \\
&= (1-\gamma)\int p_{t|1}(x \mid x_1)P(x_1 \mid s, a)\mathrm{d}x_1 + \gamma\, \mathbb{E}_{s' \sim P(\cdot|s,a)}\left[m_t^{(0)}(x \mid s', \pi(s'))\right] = \left(\mathcal{B}_t^\pi m_t^{(0)}\right)(x \mid s, a).
\end{aligned}
$$

$\square$

### E.5. Analysis of TD-CFM(C)

The idealized update of TD-CFM(C) is, for any $n \geq 0$,

$$v_t^{(n+1)}(\cdot|s,a) \in \underset{v_t(\cdot):\mathbb{R}^d \to \mathbb{R}^d}{\arg\min} \ell^{(n)}_{\text{TD-CFM(C)}}(t,s,a) \,, \text{where}$$

$$\ell^{(n)}_{\text{TD-CFM(C)}}(t,s,a) := \mathbb{E}_{(X_0,X_1) \sim \Gamma^{(n)}_{0,1}(\cdot|s,a), X_t \sim p_{t|0,1}(\cdot|X_0,X_1)} \Big[ \|v_t(X_t) - u_{t|0,1}(X_t \mid X_0, X_1)\|^2 \Big],$$

(30)

and $\Gamma^{(n)}_{0,1}(\cdot \mid s,a)$ is the coupling between $m_0$ and $\mathcal{T}^\pi m_1^{(n)}$, while $p_{t|0,1}, u_{t|0,1}$ are such that $u_{t|0,1}(x \mid x_0, x_1)$ generates $p_{t|0,1}(x \mid x_0, x_1)$, $p_{1|0,1}(x \mid x_0, x_1) = \delta_{x_1}(x)$, and

$$p_{t|1}(x \mid x_1) = \int p_{t|0,1}(x \mid x_0, x_1) m_0(x_0) \mathrm{d}x_0.$$

(31)

**Lemma 8.** *The coupling $\Gamma^{(n)}_{0,1}(\cdot \mid s,a)$ satisfies*

$$\Gamma^{(n)}_{0,1}(x_0, x_1 \mid s,a) = (1-\gamma)P(x_1 \mid s,a)m_0(x_0) + \gamma \mathbb{E}_{S' \sim P(\cdot|s,a)} \Big[ m^{(n)}_{0,1}(x_0, x_1 \mid S', \pi(S')) \Big],$$

*where $m^{(n)}_{0,1}(x_0, x_1 \mid s,a) = m_0(x_0)\delta_{\psi_1^{(n)}(x_0|s,a)}(x_1)$ is the joint distribution of $(X_0, X_1)$, i.e the endpoints of the ODE.*

*Proof.* For any $x_0, x_1$, we can write $\Gamma^{(n)}_{0,1}(x_0, x_1 \mid s,a) = \Gamma^{(n)}_1(x_1 \mid s,a,x_0)m_0(x_0)$, where $\Gamma^{(n)}_1$ is the corresponding conditional distribution. By definition, we have

$$\Gamma^{(n)}_1(x_1 \mid s,a,x_0) = (1-\gamma)P(x_1 \mid s,a) + \gamma \mathbb{E}_{s' \sim P(\cdot|s,a)} \Big[ \delta_{\psi_1^{(n)}(x_0|s',\pi(s'))}(x_1) \Big]$$

where $\psi_1^{(n)}$ is the flow that generates $m_1^{(n)}$. Multiplying both sides by $m_0(x_0)$ and using that $m^{(n)}_{0,1}(x_0, x_1 \mid s,a) = m_0(x_0)\delta_{\psi_1^{(n)}(x_0|s,a)}(x_1)$ concludes the proof. $\qquad\square$

**Lemma 9.** *For any $n \geq 0$, the vector field minimizing (30) is*

$$v_t^{(n+1)}(x \mid s,a) = \int\int u_{t|0,1}(x \mid x_0, x_1) \frac{p_{t|0,1}(x \mid x_0, x_1)\Gamma^{(n)}_{0,1}(x_0, x_1 \mid s,a)}{m_t^{(n+1)}(x \mid s,a)} \mathrm{d}x_0 \mathrm{d}x_1,$$

*where $m_t^{(n+1)}(x \mid s,a) := \int\int p_{t|0,1}(x \mid x_0, x_1)\Gamma^{(n)}_{0,1}(x_0, x_1 \mid s,a)\mathrm{d}x_0\mathrm{d}x_1$. Moreover $v_t^{(n+1)}$ generates $m_t^{(n+1)}$.*

*Proof.* Note that (30) is a standard conditional flow matching loss since $u_{t|0,1}(x \mid x_0, x_1)$ generates $p_{t|0,1}(x \mid x_0, x_1)$ and $p_{1|0,1}(x \mid x_0, x_1) = \delta_{x_1}(x)$. The expression of $v_t^{(n+1)}(x \mid s,a)$ given in the statement is exactly the vector field obtained by marginalization of the conditional vector field $u_{t|0,1}$, which we know to be the minimizer of the loss from Theorem 2 of (Lipman et al., 2023). The fact that $v_t^{(n+1)}$ generates $m_t^{(n+1)}$ is a consequence of Theorem 1 of (Lipman et al., 2023). $\quad\square$

**Lemma 10.** *For any $n \geq 0$, the probability path generated by (29) satisfies $m_1^{(n+1)}(x \mid s,a) = (\mathcal{T}^\pi m_1^{(n)})(x \mid s,a)$.*

*Proof.* By Lemma 9 and the fact that $p_{1|0,1}(x \mid x_0, x_1) = \delta_{x_1}(x)$,

$$m_1^{(n+1)}(x \mid s,a) := \int\int p_{1|0,1}(x \mid x_0, x_1)\Gamma^{(n)}_{0,1}(x_0, x_1 \mid s,a)\mathrm{d}x_0\mathrm{d}x_1$$

$$= \int \Gamma^{(n)}_{0,1}(x_0, x \mid s,a)\mathrm{d}x_0$$

$$= (\mathcal{T}^\pi m_1^{(n)})(x|s,a).$$

$\square$

**Theorem 6.** *For any $n \geq 1$, the probability path generated by* (29) *satisfies*

$$m_t^{(n+1)}(x \mid s, a) = (\mathcal{B}_t^\pi m_t^{(n)})(x \mid s, a),$$

*where $\mathcal{B}_t^\pi$ is the operator of Definition 1. Moreover, if the initial vector field $v_t^{(0)}$ satisfies*

$$v_t^{(0)}(x \mid s, a) = \int \int u_{t|0,1}(x|x_0, x_1) \frac{p_{t|0,1}(x \mid x_0, x_1) m_{0,1}^{(0)}(x_0, x_1 \mid s, a)}{m_t^{(0)}(x \mid s, a)} dx_0 dx_1,$$

*with $m_t^{(0)}$ being its generated probability path, then this result is valid at all $n \geq 0$.*

*Proof.* We know that, for all $n \geq 0$, $v_t^{n+1}$ generates $m_t^{(n+1)}$ (Lemma 9) and that $m_1^{(n+1)} = \mathcal{T}^\pi m_1^{(n)}$ (Lemma 10). While $m_t^{(n+1)}$ is written as a function of $\Gamma_{0,1}^{(n)}$ only, we can rewrite it as a function of $m_t^{(n)}$ thanks to the linearity of the Bellman operator and the definition of marginal paths. For any $n \geq 1$, By Lemma 8,

$$m_t^{(n+1)}(x \mid s, a) := \int \int p_{t|0,1}(x \mid x_0, x_1) \Gamma_{0,1}^{(n)}(x_0, x_1 \mid s, a) dx_0 dx_1$$

$$= \int \int p_{t|0,1}(x \mid x_0, x_1) \left( (1 - \gamma) P(x_1 \mid s, a) m_0(x_0) + \gamma \mathbb{E}_{S' \sim P(\cdot|s,a)} \left[ m_{0,1}^{(n)}(x_0, x_1 \mid S', \pi(S')) \right] \right) dx_0 dx_1$$

$$= (1 - \gamma) \underbrace{\int \int p_{t|0,1}(x \mid x_0, x_1) P(x_1 \mid s, a) m_0(x_0) dx_0 dx_1}_{(i)}$$

$$+ \gamma \mathbb{E}_{s' \sim P(\cdot|s,a)} \left[ \underbrace{\int \int p_{t|0,1}(x \mid x_0, x_1) m_{0,1}^{(n)}(x_0, x_1 \mid S', \pi(S')) dx_0 dx_1}_{(ii)} \right].$$

By (31),

$$(i) = \int p_{t|1}(x \mid x_1) P(x_1 \mid s, a) dx_1 = P_t(x \mid s, a).$$

For (ii), by Lemma 9, we have $m_t^{(n)}(x \mid s, a) = \int \int p_{t|0,1}(x \mid x_0, x_1) \Gamma_{0,1}^{(n-1)}(x_0, x_1 \mid s, a) dx_0 dx_1, \forall n \geq 0$, which implies

$$m_{0,1}^{(n)}(x_0, x_1 \mid s', \pi(s')) = \Gamma_{0,1}^{(n-1)}(x_0, x_1 \mid s', \pi(s')).$$

Therefore, again by definition of $m_t^{(n)}$ (Lemma 9),

$$(ii) = \mathbb{E}_{s' \sim P(\cdot|s,a)} \left[ \int \int p_{t|0,1}(x \mid x_0, x_1) \Gamma_{0,1}^{(n-1)}(x_0, x_1 \mid s', \pi(s')) dx_0 dx_1 \right]$$

$$= \mathbb{E}_{s' \sim P(\cdot|s,a)} \left[ m_t^{(n)}(x \mid s', \pi(s')) \right].$$

Plugging the expressions of (i) and (ii) into the one of $m_t^{(n+1)}(x \mid s, a)$ yields the first part of the statement.

For the second part, we only need to prove that the result also holds at $n = 0$. Note that the assumption on $v_t^{(0)}$ implies that $m_t^{(0)}(x \mid s, a) = \int \int p_{t|0,1}(x \mid x_0, x_1) m_{0,1}^{(0)}(x_0, x_1 \mid s', \pi(s')) dx_0 dx_1$. Thus, using the same decomposition above, we have

$$m_t^{(1)}(x \mid s, a) = (1 - \gamma) P_t(x \mid s, a) + \gamma \mathbb{E}_{s' \sim P(\cdot|s,a)} \left[ \int \int p_{t|0,1}(x \mid x_0, x_1) m_{0,1}^{(0)}(x_0, x_1 \mid s', \pi(s')) dx_0 dx_1 \right]$$

$$= (1 - \gamma) P_t(x \mid s, a) + \gamma \mathbb{E}_{s' \sim P(\cdot|s,a)} \left[ m_t^{(0)}(x \mid s', \pi(s')) \right],$$

which proves the result. $\qquad \square$

### E.6. Variance Analysis

**Theorem 7.** *Let us define the random variables*

$$g_{\text{TD}^2\text{-CFM}}(t, s, a, s', \vec{X}_t, X_t^{(n)}) := (1 - \gamma)\nabla_\theta v_t(\vec{X}_t|s, a; \theta)^\top \left(v_t(\vec{X}_t|s, a; \theta) - u_{t|1}(\vec{X}_t|s')\right)$$
$$+ \gamma \nabla_\theta v_t(X_t^{(n)}|s, a; \theta)^\top \left(v_t(X_t^{(n)}|s, a; \theta) - v_t^{(n)}(X_t^{(n)}|s', \pi(s'))\right)$$
$$g_{\text{TD-CFM}}(t, s, a, s', \vec{X}_t, X_1, X_t) := (1 - \gamma)\nabla_\theta v_t(\vec{X}_t|s, a; \theta)^\top \left(v_t(\vec{X}_t|s, a; \theta) - u_{t|1}(\vec{X}_t|s')\right)$$
$$+ \gamma \nabla_\theta v_t(X_t|s, a; \theta)^\top \left(v_t(X_t|s, a; \theta) - u_{t|1}(X_t|X_1)\right)$$

*where $t \sim \mathcal{U}([0, 1])$, $(s, a) \sim \rho$, $s' \sim P(\cdot|s, a)$, $\vec{X}_t \sim p_{t|1}(\cdot|s')$, $X_t^{(n)} \sim m_t^{(n)}(\cdot \mid s', \pi(s'))$, $X_1 \sim m_1^{(n)}(\cdot \mid s', \pi(s'))$, and $X_t \sim p_{t|1}(\cdot|X_1)$. Then, $g_{\text{TD}^2\text{-CFM}}$ and $g_{\text{TD-CFM}}$ are respectively unbiased estimates of the gradients $\nabla_\theta \ell_{\text{TD}^2\text{-CFM}}(\theta)$ and $\nabla_\theta \ell_{\text{TD-CFM}}(\theta)$.*

*Moreover, if we consider their respective total variations defined as:*

$$\sigma_{\text{TD}^2\text{-CFM}}^2 = \text{Trace}\left(\text{Cov}_{t,s,a,s',\vec{X}_t,X_t^{(n)}}\left[g_{\text{TD}^2\text{-CFM}}(t, s, a, s', \vec{X}_t, X_t^{(n)})\right]\right)$$
$$\sigma_{\text{TD-CFM}}^2 = \text{Trace}\left(\text{Cov}_{t,s,a,s',\vec{X}_t,X_1,X_t}\left[g_{\text{TD-CFM}}(t, s, a, s', \vec{X}_t, X_1, X_t)\right]\right)$$

*and we assume that $m_t^{(n)}(x \mid s, a) = \int p_{t|1}(x \mid x_1) m_1^{(n)}(x_1 \mid s, a)\mathrm{d}x_1$, then we obtain*

$$\sigma_{\text{TD-CFM}}^2 = \sigma_{\text{TD}^2\text{-CFM}}^2 + \gamma^2 \mathbb{E}_{t,s,a,X_t}\left[\text{Trace}\left(\text{Cov}_{X_1|s,a,X_t}\left[\nabla_\theta v_t(X_t|s, a; \theta)^\top u_{t|1}(X_t \mid X_1)\right]\right)\right].$$

*Proof.* Recall the $\text{TD}^2$-CFM and TD-CFM objectives:

$$\ell_{\text{TD}^2\text{-CFM}}(\theta) = (1 - \gamma)\mathbb{E}_{t,s,a,s',X_t \sim p_{t|1}(\cdot|s')}\left[\left\|v_t(X_t|s, a; \theta) - u_{t|1}(X_t|s')\right\|^2\right]$$
$$+ \gamma \mathbb{E}_{t,s,a,s',X_t \sim m_t^{(n)}(\cdot|s',\pi(s'))}\left[\left\|v_t(X_t|s, a; \theta) - v_t^{(n)}(X_t|s', \pi(s'))\right\|^2\right],$$
$$\ell_{\text{TD-CFM}}(\theta) = (1 - \gamma)\mathbb{E}_{t,s,a,s',X_t \sim p_{t|1}(\cdot|s')}\left[\left\|v_t(X_t|s, a; \theta) - u_{t|1}(X_t|s')\right\|^2\right]$$
$$+ \gamma \mathbb{E}_{t,s,a,s',X_1 \sim m_1^{(n)}(\cdot|s',\pi(s')),X_t \sim p_{t|1}(\cdot|X_1)}\left[\left\|v_t(X_t|s, a; \theta) - u_{t|1}(X_t|X_1)\right\|^2\right].$$

Computing the gradients of these quantities w.r.t. $\theta$, it is easy to check that $g_{\text{TD}^2\text{-CFM}}$ and $g_{\text{TD-CFM}}$ are their unbiased estimates.

Let us now analyze the total variation of these estimators. By assumption, we have $m_t^{(n)}(x \mid s, a) = \int p_{t|1}(x \mid x_1) m_1^{(n)}(x_1 \mid s, a)\mathrm{d}x_1$, which implies that $X_t^{(n)}$ and $X_t$ follow the same law. Moreover, we obtain the following identities:

$$v_t^{(n)}(x \mid s', \pi(s')) = \mathbb{E}_{X_1|x,s'}\left[u_{t|1}(x \mid X_1)\right],$$
$$g_{\text{TD}^2\text{-CFM}}(t, s, a, s', \vec{X}_t, X_t) = \mathbb{E}_{X_1|X_t,s'}\left[g_{\text{TD-CFM}}(t, s, a, s', X_t^o, X_1, X_t)\right],$$
$$\mathbb{E}_{X_t \sim m_t^{(n)}(\cdot|s',\pi(s'))}\left[g_{\text{TD}^2\text{-CFM}}(t, s, a, s', \vec{X}_t, X_t)\right] = \mathbb{E}_{\substack{X_1 \sim m_1^{(n)}(\cdot|s',\pi(s')) \\ X_t \sim p_{t|1}(\cdot|X_1)}}\left[g_{\text{TD-CFM}}(t, s, a, s', \vec{X}_t, X_1, X_t)\right],$$

where $X_1 \mid x, s' \sim \frac{p_{t|1}(x|X_1) m_1^{(n)}(X_1|s',\pi(s'))}{m_t^{(n)}(x|s,a)}$ is the posterior distribution of $X_1$ given $x$ and $s'$.

To simplify notation, we denote by $Y$ the random variable $(t, s, a, s', \vec{X}_t)$. Using the decomposition of variance into

conditional variance, $\mathrm{Var}(X) = \mathbb{E}[\mathrm{Var}(X|Y)]) + \mathrm{Var}(\mathbb{E}[X|Y])$, we conclude that

$$
\begin{aligned}
\sigma_{\text{TD-CFM}} &= \mathrm{Trace}\left(\mathrm{Cov}_{Y,X_1,X_t}\left[g_{\text{TD-CFM}}(Y,X_1,X_t)\right]\right) \\
&= \mathbb{E}_{Y,X_1,X_t}\left[\left\|g_{\text{TD-CFM}}(Y,X_1,X_t) - \mathbb{E}_{Y,X_1,X_t}\left[g_{\text{TD-CFM}}(Y,X_1,X_t)\right]\right\|^2\right] \\
&= \mathbb{E}_{Y,X_t}\left[\left\|\mathbb{E}_{X_1|Y,X_t}\left[g_{\text{TD-CFM}}(Y,X_1,X_t)\right] - \mathbb{E}_{Y,X_1,X_t}\left[g_{\text{TD-CFM}}(Y,X_1,X_t)\right]\right\|^2\right] \\
&\quad + \mathbb{E}_{Y,X_t}\left[\mathbb{E}_{X_1|Y,X_t}\left[\left\|g_{\text{TD-CFM}}(Y,X_1,X_t) - \mathbb{E}_{X_1|Y,X_t}\left[g_{\text{TD-CFM}}(Y,X_1,X_t)\right]\right\|^2\right]\right] \\
&= \mathbb{E}_{Y,X_t}\left[\left\|g_{\text{TD}^2\text{-CFM}}(Y,X_t) - \mathbb{E}_{Y,X_t}\left[g_{\text{TD}^2\text{-CFM}}(Y,X_t)\right]\right\|^2\right] \\
&\quad + \gamma^2 \mathbb{E}_{Y,X_t}\left[\mathbb{E}_{X_1|Y,X_t}\left[\left\|\nabla_\theta v_t(X_t|s,a;\theta)^\top u_{t|1}(X_t\mid X_1) - \mathbb{E}_{X_1|Y,X_t}\left[\nabla_\theta v_t(X_t|s,a;\theta)^\top u_{t|1}(X_t\mid X_1)\right]\right\|^2\right]\right] \\
&= \sigma_{\text{TD}^2\text{-CFM}} + \gamma^2 \mathbb{E}_{Y,X_t}\left[\mathrm{Trace}\left(\mathrm{Cov}_{X_1|Y,X_t}\left[\nabla_\theta v_t(X_t|s,a;\theta)^\top u_{t|1}(X_t\mid X_1)\right]\right)\right] \\
&= \sigma_{\text{TD}^2\text{-CFM}} + \gamma^2 \mathbb{E}_{t,s,a,X_t}\left[\mathrm{Trace}\left(\mathrm{Cov}_{X_1|s,a,X_t}\left[\nabla_\theta v_t(X_t|s,a;\theta)^\top u_{t|1}(X_t\mid X_1)\right]\right)\right].
\end{aligned}
$$

$\square$

**Theorem 8.** *Let us define the random variable*

$$
\begin{aligned}
g_{\text{TD-CFM(C)}}(t,s,a,s',\vec{X}_t,X_0,X_1,X_t) &:= (1-\gamma)\nabla_\theta v_t(\vec{X}_t|s,a;\theta)^\top\left(v_t(\vec{X}_t|s,a;\theta) - u_{t|0,1}(\vec{X}_t|X_0,s')\right) \\
&\quad + \gamma\nabla_\theta v_t(X_t|s,a;\theta)^\top\left(v_t(X_t|s,a;\theta) - u_{t|0,1}(X_t|X_0,X_1)\right)
\end{aligned}
$$

*where* $t \sim \mathcal{U}([0,1]), (s,a) \sim \rho, s' \sim P(\cdot|s,a), \vec{X}_t \sim p_{t|1}(\cdot|s'), (X_0,X_1) \sim m_{0,1}^{(n)}(\cdot \mid s',\pi(s'))$ *and* $X_t \sim p_{t|0,1}(\cdot|X_0,X_1,)$. *Then* $g_{\text{TD-CFM(C)}}$ *is an unbiased estimate of the gradient* $\nabla_\theta \ell_{\text{TD-CFM(C)}}(\theta)$.

*Moreover, if we consider its total variation defined as:*

$$
\sigma_{\text{TD-CFM(C)}} = \mathrm{Trace}\left(\mathrm{Cov}_{t,s,a,s',\vec{X}_t,X_0,X_1,X_t}\left[g_{\text{TD-CFM(C)}}(t,s,a,s',\vec{X}_t,X_0,X_1,X_t)\right]\right)
$$

*and we assume that* $m_t^{(n)}(x\mid s,a) = \int\int p_{t|0,1}(x\mid x_0,x_1)m_{0,1}^{(n)}(x_0,x_1\mid s,a)\mathrm{d}x_0\mathrm{d}x_1$, *then we obtain*

$$
\sigma_{\text{TD-CFM(C)}} = \sigma_{\text{TD}^2\text{-CFM}} + \gamma^2 \mathbb{E}_{t,s,a,X_t}\left[\mathrm{Trace}\left(\mathrm{Cov}_{(X_0,X_1)|s,a,X_t}\left[\nabla_\theta v_t(X_t|s,a;\theta)^\top u_{t|0,1}(X_t\mid X_0,X_1)\right]\right)\right].
$$

*Furthermore, if we use straight conditional paths, i.e.,* $p_{t|0,1}(x|x_0,x_1) = \delta(tx_1 + (1-t)x_0 - x)$, *then*

$$
\begin{aligned}
\sigma_{\text{TD-CFM(C)}} &\leq \sigma_{\text{TD}^2\text{-CFM}} \\
&\quad + \gamma^2 \sup_{t,s,a,x}\left\|\nabla_\theta v_t(x|s,a;\theta)\right\|^2 \mathbb{E}_{t,s,a,s',X_0,X_1,X_t}\left[\|X_1 - X_0 - \mathbb{E}_{(X_1,X_0)|s,a,s',X_t}[X_1 - X_0]\|^2\right].
\end{aligned}
$$

*In particular, when the paths of the linear interpolation* $X_t$ *do not intersect for any* $s,a,s'$, *we have* $\mathbb{E}_{t,s,a,s',X_0,X_1,X_t}\left[\|X_1 - X_0 - \mathbb{E}_{(X_1,X_0)|s,a,s',X_t}[X_1 - X_0]\|^2\right] = 0$ *and* $\sigma_{\text{TD-CFM(C)}} = \sigma_{\text{TD}^2\text{-CFM}}$.

*Proof.* The first two statements can be checked by repeating the proof of Theorem 7 with conditional paths $p_{t|0,1}$ and vector fields $u_{t|0,1}$. Let us thus prove the second part. We know that the flow $\phi_t(x_0,x_1)$ that generates the the conditonal path $p_{t|0,1}(x|x_0,x_1) = \delta_{tx_1+(1-t)x_0}(x)$ is $\phi_t(x_0,x_1) = tx_1 + (1-t)x_0$. Its associated vector field $u_{t|0,1}$ is thus

$$
u_{t|0,1}(\phi_t(x_0,x_1)|x_0,x_1) = \frac{d}{dt}\phi_t(x_0,x_1) = x_1 - x_0.
$$

Theorefore, denoting $Y = (t, s, a)$, we can bound the second term in the decomposition of $\sigma_{\text{TD-CFM(C)}}$ as

$$\mathbb{E}_{Y,X_t}\left[\text{Trace}\left(\text{Cov}_{(X_0,X_1)|Y,X_t}\left[\nabla_\theta v_t(X_t|s,a;\theta)^\top u_{t|1}(X_t \mid X_0,X_1)\right]\right)\right]$$

$$= \mathbb{E}_{Y,X_t}\left[\mathbb{E}_{X_0,X_1|Y,X_t}\left[\left\|\nabla_\theta v_t(X_t|s,a;\theta)^\top u_{t|0,1}(X_t \mid X_0,X_1) - \mathbb{E}_{X_0,X_1|Y,X_t}\left[\nabla_\theta v_t(X_t|s,a;\theta)^\top u_{t|0,1}(X_t \mid X_0,X_1)\right]\right\|^2\right]\right]$$

$$\leq \mathbb{E}_{Y,X_t}\left[\|\nabla_\theta v_t(X_t|s,a;\theta)\|^2 \mathbb{E}_{X_0,X_1|Y,X_t}\left[\left\|u_{t|0,1}(X_t \mid X_0,X_1) - \mathbb{E}_{X_0,X_1|Y,X_t}\left[u_{t|0,1}(X_t \mid X_0,X_1)\right]\right\|^2\right]\right]$$

$$= \mathbb{E}_{Y,X_t}\left[\|\nabla_\theta v_t(X_t|s,a;\theta)\|^2 \mathbb{E}_{X_0,X_1|Y,X_t}\left[\left\|X_0 - X_1 - \mathbb{E}_{X_0,X_1|Y,X_t}\left[X_1 - X_0\right]\right\|^2\right]\right]$$

$$\leq \sup_{t,s,a,x}\|\nabla_\theta v_t(x|s,a;\theta)\|^2 \mathbb{E}_{Y,X_t}\left[\mathbb{E}_{X_0,X_1|Y,X_t}\left[\left\|X_0 - X_1 - \mathbb{E}_{X_0,X_1|Y,X_t}\left[X_1 - X_0\right]\right\|^2\right]\right].$$

This proves the third statement. For the last point, simply note that if the paths generating $X_t$ do not cross, then the distribution of $X_0, X_1|Y, X_t$ is supported over a single couple $(X_0, X_1)$, which means that its variance is zero. □

### E.7. Transport Cost Analysis

**Theorem 9.** *Assume that $m_t^{(n)}(x \mid s, a) = \int p_{t|1}(x \mid x_1) m_1^{(n)}(x_1 \mid s, a) \mathrm{d}x_1$, where $p_{t|1}(\cdot \mid x_1) = \mathcal{N}(tx_1, (1-t)^2 I)$ is a Gaussian path. Then, the conditional paths [6] built by* TD-CFM(C) *and* TD$^2$-CFM *to generate $m_1^{(n+1)} = \mathcal{T}^\pi m_1^{(n)}$ induce a smaller transport cost than those built by* TD-CFM. *Formally, for every $t, s, a$,*

$$\mathbb{E}_{t,s,a,s',X_0 \sim m_0, X_1 \sim (1-\gamma)\delta_{s'} + \gamma\delta_{\psi_1^{(n)}(X_0|s',\pi(s'))}} \left[\|X_1 - X_0\|^2\right] \leq \mathbb{E}_{t,s,a,s',X_0 \sim m_0, X_1 \sim [\mathcal{T}^\pi m_1^{(n)}](\cdot|s,a)} \left[\|X_1 - X_0\|^2\right].$$

*Proof.* The paths generated by TD-CFM(C) and TD$^2$-CFM induce the same transport cost since both algorithms connect the endpoints of the ODE path $m_t^{(n)}$ in the bootstrapped term. Hence,

$$\mathbb{E}_{t,s,a,s',X_0 \sim m_0, X_1 \sim (1-\gamma)\delta_{s'} + \gamma\delta_{\psi_1^{(n)}(X_0|s',\pi(s'))}} \left[\|X_1 - X_0\|^2\right]$$

$$= (1-\gamma)\mathbb{E}_{t,s,a,s',X_0}\left[\|s' - X_0\|^2\right] + \gamma\mathbb{E}_{t,s,a,s',X_0}\left[\|\psi_1^{(n)}(X_0 \mid s', \pi(s')) - X_0\|^2\right]$$

$$\overset{(a)}{=} (1-\gamma)\mathbb{E}_{t,s,a,s',X_0}\left[\|s' - X_0\|^2\right] + \gamma\mathbb{E}_{t,s,a,s',X_0}\left[\left\|\int v_t^{(n)}(\psi_t^{(n)}(X_0 \mid s', \pi(s')))\mathrm{d}t\right\|^2\right]$$

$$\overset{(b)}{\leq} (1-\gamma)\mathbb{E}_{t,s,a,s',X_0}\left[\|s' - X_0\|^2\right] + \gamma\int \mathbb{E}_{t,s,a,s',X_0}\left[\left\|v_t^{(n)}(\psi_t^{(n)}(X_0 \mid s', \pi(s')))\right\|^2\right]\mathrm{d}t$$

$$\overset{(c)}{=} (1-\gamma)\mathbb{E}_{t,s,a,s',X_0}\left[\|s' - X_0\|^2\right] + \gamma\int \mathbb{E}_{t,s,a,s',X_t \sim m_t^{(n)}(\cdot|s',\pi(s'))}\left[\left\|v_t^{(n)}(X_t \mid s', \pi(s'))\right\|^2\right]\mathrm{d}t$$

$$\overset{(d)}{=} (1-\gamma)\mathbb{E}_{t,s,a,s',X_0}\left[\|s' - X_0\|^2\right] + \gamma\int \mathbb{E}_{t,s,a,s',X_t \sim m_t^{(n)}(\cdot|s',\pi(s'))}\left[\left\|\mathbb{E}_{X_1|s',X_t}\left[u_{t|1}(X_t|X_1)\right]\right\|^2\right]\mathrm{d}t$$

$$\overset{(e)}{\leq} (1-\gamma)\mathbb{E}_{t,s,a,s',X_0}\left[\|s' - X_0\|^2\right] + \gamma\int \mathbb{E}_{t,s,a,s',X_t \sim m_t^{(n)}(\cdot|s',\pi(s'))}\left[\mathbb{E}_{X_1|s',X_t}\left[\left\|u_{t|1}(X_t|X_1)\right\|^2\right]\right]\mathrm{d}t$$

$$\overset{(f)}{=} (1-\gamma)\mathbb{E}_{t,s,a,s',X_0}\left[\|s' - X_0\|^2\right] + \gamma\int \mathbb{E}_{t,s,a,s',X_1 \sim m_1^{(n)}(\cdot|s',\pi(s')),X_t \sim p_{t|1}(\cdot|X_1)}\left[\left\|u_{t|1}(X_t|X_1)\right\|^2\right]\mathrm{d}t$$

$$\overset{(g)}{=} (1-\gamma)\mathbb{E}_{t,s,a,s',X_0}\left[\|s' - X_0\|^2\right] + \gamma\int \mathbb{E}_{t,s,a,s',X_1 \sim m_1^{(n)}(\cdot|s',\pi(s')),X_0}\left[\left\|u_{t|1}(tX_1 + (1-t)X_0|X_1)\right\|^2\right]\mathrm{d}t$$

$$\overset{(h)}{=} (1-\gamma)\mathbb{E}_{t,s,a,s',X_0}\left[\|s' - X_0\|^2\right] + \gamma\int \mathbb{E}_{t,s,a,s',X_1 \sim m_1^{(n)}(\cdot|s',\pi(s')),X_0}\left[\left\|X_1 - X_0\right\|^2\right]\mathrm{d}t$$

$$\overset{(i)}{=} (1-\gamma)\mathbb{E}_{t,s,a,s',X_0}\left[\|s' - X_0\|^2\right] + \gamma\mathbb{E}_{t,s,a,s',X_1 \sim m_1^{(n)}(\cdot|s',\pi(s')),X_0}\left[\left\|X_1 - X_0\right\|^2\right]$$

$$\overset{(j)}{=} \mathbb{E}_{t,s,a,s',X_0 \sim m_0, X_1 \sim [\mathcal{T}^\pi m_1^{(n)}](\cdot|s,a)} \left[\|X_1 - X_0\|^2\right],$$

where (a) uses the definition of flow as integration of a vector field, (b) uses Cauchy-Schwarz inequality, (c) uses that $m_0 * \psi_t^{(n)}$ is the pushforward measure generating $m_t^{(n)}$, (d) defines $X_1 \mid x, s' \sim \frac{p_{t|1}(x|X_1) m_1^{(n)}(X_1|s',\pi(s'))}{m_t^{(n)}(x|s,a)}$ as the posterior distribution of $X_1$ given $x, s'$ and uses that $v_t^{(n)}$ is in marginal form by assumption, (e) uses Jensen's inequality, (f) uses the Tower property of expectations, (g) uses the definition of $p_{t|1}$ and the corresponding linear-interpolation flow, (h) uses the definition of $u_{t|1}$, (i) is trivial, and (j) simply combines the two terms using the definition of Bellman operator $\mathcal{T}^\pi$. □

---

[6] Recall that, given a marginal probability path $m_t^{(n)}(x \mid s, a)$, the conditional probability path built by TD-CFM(C) and TD$^2$-CFM to generate $\mathcal{T}^\pi m_1^{(n)}$ is a linear interpolation between noise $X_0 \sim m_0$ and $X_1 \sim (1-\gamma)\delta_{s'} + \gamma\psi_1^{(n)}(X_0|s',\pi(s'))$, while the one built by TD-CFM is a linear interpolation between noise $X_0 \sim m_0$ and a sample $X_1 \sim [\mathcal{T}^\pi m_1^{(n)}](\cdot \mid s, a)$ from the target distribution.

