# OpenReview forum: "Temporal Difference Flows"
_ICML.cc/2025/Conference — ICML 2025 oral_

### Official Review · Reviewer_SPuZ · 2025-02-27

**Overall Recommendation:** 4

**Summary:**

This paper studies the problem of learning generative horizon models (GHMs), which are generative models of the successor measure of a policy. That is, GHMs are models capable of generating samples from the discounted distribution of future states visited by a policy at any given time step $t$. In particular, this paper introduces Temporal Difference Flows (TD-Flow), a class of flow-matching techniques used to learn GHMs. The authors introduce three variants of TD-Flow (TD-CFM, Coupled TD-CFM, TD^2-CFM) and additional extensions to diffusion models (TD-DD, TD^2-DD) that improve the accuracy and stability of the learned GHMs. The methods are evaluated in robotics domains from the DeepMind Control Suite with respect to different error metrics, as well as in terms of performance when the methods are employed in a zero-shot transfer setting in combination with generalized policy improvement (GPI).

---

### Post Rebuttal

I thank the authors for their careful response. I am maintaining my acceptance score since I do not have additional questions.

However, I would like to emphasize the importance of including the confidence intervals for the figures in the revised manuscript.

**Claims And Evidence:**

Overall, the paper provides strong empirical and theoretical support for its claims, with extensive benchmarking and mathematical justification.

Regarding its theoretical results, the paper offers convergence guarantees for the TD-based flow methods, demonstrating that TD2-CFM, TD-CFM, and TD-CFM(C) all converge to a fixed probability path​ (assuming that the flow-matching loss is minimized exactly at each iteration).

Regarding the empirical validation, the results indicate that TD^2-based methods outperform alternative models (GANs, VAEs, and diffusion models) in long-horizon predictions, with significantly lower error metrics such as Earth Mover’s Distance and value-function mean squared error.

The paper also includes ablation experiments to analyze the effect of path design, confirming the robustness of the introduced techniques.

**Essential References Not Discussed:**

To the best of my knowledge, the paper discusses the most relevant current state-of-the-art methods for learning GHMs.

**Experimental Designs Or Analyses:**

The experimental design of the paper is well-conducted and includes an evaluation of the proposed techniques with different relevant metrics (log-likelihood, Earth Mover’s distance, and value function MSE). The authors also investigate how the performance degrades with larger horizons, and how they affect the performance of an agent that employs generalized policy improvement (GPI) for solving tasks.

My only concern with the empirical design is that the authors employ only three random seeds for each domain. The authors should justify the use of such a small value. Moreover, the results in Figure 2 and Figure 3 should also report the confidence level or statistical dispersion of the reported metrics.

**Methods And Evaluation Criteria:**

The proposed methods are evaluated in 4 domains (Maze, Walker, Cheetah, Quadruped) from the DeepMind Control Suite. This benchmark is standard in the RL literature and provides diverse control challenges for evaluating the proposed method. One concern I have is that these domains are almost deterministic. It would be interesting to see how the proposed methods perform in domains with more stochastic transitions, e.g., with multi-modal distributions of next states.

Regarding the baselines employed in the experiments, they represent current state-of-the-art techniques for learning GHMs and make sense for the empirical evaluation.

**Other Comments Or Suggestions:**

- In line 29, Thakoor et al., 2022 refer to GHM as “geometric horizon models”, and not  “generative horizon models”.

- In line 59, why restrict to deterministic policies?

- In Equations 2 and 3, the variable $\rho$ was not defined.

- In Section 5.2, I suggest discussing in more depth the relation with the GPI technique of (Thakoor
et al., 2022), which also extends GPI via GHMs. Currently, in this section, the authors only mention the work that first introduced GPI (Barreto et al., 2017). Did the authors try to perform GPI similarly to Thakoor et al, 2022, that is, using geometric switching policies?

- The introduced method had many variants (TD-CFM, Coupled TD-CFM, TD^2-CFM) and additional extensions to diffusion models (TD-DD, TD^2-DD). This makes it difficult for the reader to track the disadvantages and advantages of each variant, as well as their unique properties. I suggest the authors include a table that summarizes all introduced variants with their properties and pros and cons for improved clarity of the paper.

**Other Strengths And Weaknesses:**

Besides the points raised in the other sections, one of the paper's strengths is its clarity and presentation. In particular, the paper covers many different mathematical formulations (e.g., RL, flow matching, diffusion models) while maintaining mathematical rigor, clarity, and precision.

One point for improvement would be the addition of a dedicated related work section that pinpoints the most relevant contributions of the paper in contrast to the state-of-the-art. Currently, these points are scattered throughout the paper.

**Questions For Authors:**

See Other Comments Or Suggestions.

**Relation To Broader Scientific Literature:**

This paper's main contributions extend previous works based on the successor representation/measure (SR), especially regarding methods for learning generative horizon models (GHMs) of the SR. Previous works have explored learning GHMs using techniques such as GANs, Normalizing Flows, and auto-encoders. This paper introduced new algorithms and theoretical results based on flow match and diffusion models for learning GHMs. Importantly, their methods are proven to converge to the successor measure and are shown to reduce the variance of sample-based gradient estimates.

This paper is also related to policy transfer methods based on generalized policy improvement (GPI). In particular, the authors employ their introduced techniques to perform GPI using GHMs conditioned on policy embeddings.

**Theoretical Claims:**

I checked the proof of Theorems 1, 2, and 3 and did not find any issues. I did not check the proofs of other theoretical results in the paper and in Appendix E.

---

> ### Author Rebuttal · Authors · 2025-04-01
>
> We thank the reviewer for the detailed and thoughtful evaluation, and for recognizing the clarity, rigor, and contributions of the work—both theoretical and empirical. Your careful reading and constructive suggestions are greatly valued.
>
> > **Number of seeds and statistical reporting**
>
>
> Thank you for raising this important point. While we report results using 3 random seeds, this is done per policy and task, across multiple tasks within each domain—resulting in a substantial number of evaluations across the full benchmark suite. For generative modeling alone, the single- and multi-policy experiments involve over 800 individual runs, not including the computational cost of training the TD3 and FB policies. Additionally, in practice, we observed consistently low variance across runs, especially in value estimation and planning metrics, which informed our choice. We’ll be happy to include confidence intervals for figures in the revised manuscript.
>
> > **Related work and comparison to GPI techniques**
>
>
> Thank you for the keen suggestion. While we did not evaluate the geometric switching variant of GPI proposed in [1], we see it as a natural and promising direction for future work—particularly given that TD-Flow significantly improves the underlying GHM used in their approach. We believe combining TD-Flow with geometric switching can further improve performance, but given the technical depth already required to develop and analyze TD-based flow and diffusion models, we felt this extension was best left to future work that can give it proper attention. A more focused related work discussion is already included in Appendix C, and we’d be happy to bring specific references or points into the main text if there are any the reviewer feels would benefit from greater emphasis.
>
> > **Clarifications and presentation improvements**
>
>
> - **Line 29**: Thank you for catching this error, as defined in the appendix this should read geometric not generative.
> - **Line 59**: We restrict to deterministic policies primarily for clarity and simplicity of exposition, especially in the theoretical sections. We’ll clarify this point in the revised manuscript.
> - **Equations 2 and 3**: Thank you for pointing this out. We will add a definition of $\rho$ in the text. Specifically, $\rho$ denotes the empirical distribution over transitions sampled from a replay buffer or dataset.
> - **Many variants and method comparison**: Thank you for the suggestion. To improve clarity, we’ll expand Table 3 in the Appendix to include the TD-diffusion variants and highlight the key differences between all methods. If space allows, we’ll consider moving this table to the main text; at the very least, we’ll ensure it’s clearly referenced from the main body to help guide readers.
>
> ---
>
> We’re grateful for the reviewer’s thoughtful feedback and are excited to incorporate these refinements into the final version. We hope the improved clarity and added discussion further strengthen the case for the paper’s significance.
>
> ### References
> [1] Shantanu Thakoor, Mark Rowland, Diana Borsa, Will Dabney, Rémi Munos, and André Barreto. Generalised Policy Improvement with Geometric Policy Composition. International Conference on Machine Learning (ICML), 2022.

---

> > ### Comment · Reviewer_SPuZ · 2025-04-01
> >
> > I thank the authors for their careful response. I am maintaining my acceptance score since I do not have additional questions.
> >
> > However, I would like to emphasize the importance of including the confidence intervals for the figures in the revised manuscript.

---

### Official Review · Reviewer_J3Z7 · 2025-03-14

**Overall Recommendation:** 3

**Summary:**

This paper introduces a flow-matching and diffusion-based generative modeling framework to learn accurate Geometric Horizon Models by integrating temporal difference structure in the learning objective and sampling process. The proposed TD$^2$-DD and TD$^2$-CFM achieve low modeling errors in both success measure and value function, and remain robust in long effective horizons. Furthermore, the TD-based GHM approaches demonstrate strong planning performance with Generalized Policy Improvement, greatly outperforming the baseline based on Forward-Backward representation.

## Update after rebuttal
I thank the authors for their detailed response. Most of my concerns have been addressed. I will keep my positive score.

**Claims And Evidence:**

Please see the “Other Strengths and Weaknesses” and “Questions” sections below

**Essential References Not Discussed:**

N/A

**Experimental Designs Or Analyses:**

Please see the “Other Strengths and Weaknesses” and “Questions” sections below

**Methods And Evaluation Criteria:**

Please see the “Other Strengths and Weaknesses” and “Questions” sections below

**Other Comments Or Suggestions:**

1. Please re-format the equation (the parentheses in the denominator) on the left column of Line 189.

**Other Strengths And Weaknesses:**

Strengths:
- The proposed method innovatively builds GHMs on flow and diffusion models, while injecting temporal difference structure into the sampling procedure and learning objective.
- Extensive evaluations demonstrate the efficacy of TD$^2$-CFM and TD$^2$-DD through non-trivial improvement over generative metrics and planning performance.
- TD$^2$-CFM remains effective and robust in curved path designs.

Weaknesses:
- It would be helpful to show how the quality and scale of offline datasets might affect the planning performance and modeling accuracy
- FB[1] evaluates tasks of different types, including complex maze navigation, robotic manipulation, and games, whereas the evaluations in Section 5.2 mainly focus on continuous locomotion tasks (3 out of 4 domains), including Walker, Cheetah, and Quadruped. Since TD$^2$-based GHM demonstrates the capabilities of modeling successor measures over a long effective horizon, tasks that require long-horizon reasoning, such as navigation tasks in a large Pointmaze or the four-room setting of MiniGrid, should also be tested.
- So far the evaluation seems mainly performed in low-dimensional but deterministic environments. Given the strong capabilities of flow and diffusion models, it would be a meaningful investigation to understand their performance in stochastic or partially observable environments.

[1] Touati, A. and Ollivier, Y. Learning one representation to optimize all rewards. In Neural Information Processing Systems (NeurIPS), 2021.

**Questions For Authors:**

1. Bootstrapping is one cause of the training instability in RL when combined with off-policy learning and function approximation. How would the use of bootstrapped samples affect the training stability of the proposed GHMs overall?

**Relation To Broader Scientific Literature:**

N/A

**Theoretical Claims:**

No critical issues are observed in my review of theoretical claims.

---

> ### Author Rebuttal · Authors · 2025-04-01
>
> We sincerely thank the reviewer for their detailed and thoughtful evaluation. We appreciate the recognition of our theoretical contributions, the robustness of TD²-CFM and TD²-DD in long-horizon modeling, and the strong planning performance demonstrated through Generalized Policy Improvement (GPI).
>
> > **Impact of dataset quality and scale on planning and modeling performance**
>
>
> We agree that understanding how data quality and scale affects performance is an important question. While we did not vary dataset size in this work, we note that all prior GHM methods similarly rely on offline data and suffer from instability and degraded performance even in much simpler and lower-dimensional environments than those we consider. Our key contribution is to show that by integrating temporal difference structure into both the learning objective and sampling process, we can significantly improve both stability and modeling performance. This is supported by both our theoretical analysis and strong empirical results demonstrating robustness across complex control tasks.
>
> For the planning experiments, we use the same standard offline dataset as the Forward-Backward (FB) representation from [2], which has become a common benchmark in this setting. This choice enables a direct comparison and reveals a compelling result: TD²-based GHMs are able to discover significantly better policies from the same data. This highlights both the strength of our method and the fact that FB fails to identify the best policies within its own class—despite having access to the same data and that its reward embedding is purported to be optimal. We will clarify this in the revised manuscript to better contextualize our design choices.
>
> > **Task diversity and inclusion of discrete/stochastic domains**
>
>
> Thank you for this thoughtful suggestion. In this paper, we focused on continuous control tasks as they are widely used benchmarks for evaluating generative models and the successor measure. Importantly, these settings already pose significant challenges for long-horizon modeling, and prior methods struggle to model even lower-dimensional versions of these tasks.
>
> As a point of reference, the Ms. Pacman maze experiment in [1] only models the successor measure of 2D $(x,y)$ coordinates of the agent (the backward embedding takes as input $(x,y)$ and not full RGB frames), whereas our method learns to model much higher-dimensional state spaces—including full-body continuous dynamics. Additionally, our Pointmass maze tasks are sparse-reward and require long-horizon reasoning, making them meaningfully different in structure and challenge from the locomotion tasks.
>
> We also agree that moving toward partially observable settings would be a valuable next step. At the same time, such environments introduce a distinct set of challenges, including belief-state estimation, memory, and uncertainty modeling, which we view as part of a larger and important research direction. We see our work as establishing a strong foundation for scalable, stable successor state modeling in fully observable settings, and we’re excited about the community building on this in future work.
>
> > **Use of TD and potential training instability due to bootstrapping**
>
>
> Thank you for raising this important point. You're absolutely right that bootstrapping—especially when combined with function approximation and off-policy data—can introduce instability in reinforcement learning. These challenges are not fully avoidable in our setting either, as TD²-based GHMs retain all the characteristics of the so-called "deadly triad."
>
> That said, a central goal of our approach is to mitigate these issues as effectively as possible. By integrating temporal-difference structure into flow and diffusion-based models, we observe significantly improved training stability and sample efficiency compared to prior GHM methods, which often struggle even at relatively short horizons. Our theoretical results further support this by showing that tighter coupling between sampling and regression reduces gradient variance, contributing to more stable learning.
>
> In practice, we also incorporate common stabilization techniques—such as employing an exponential moving average target network. While instability remains a core challenge in TD-based methods, we believe our work represents a meaningful step toward more stable and scalable generative modeling of the successor measure.
>
> > **Formatting**
> - **L189**: Thank you—we will correct the formatting issue in the final version.
>
> ---
>
> We hope our responses help to contextualize the design choices and emphasize the broader relevance of our contributions.
>
> ### References
> [1] Ahmed Touati and Yann Ollivier. Learning One Representation to Optimize All Rewards. Neural Information Processing Systems (NeurIPS), 2021.
>
>
> [2] Ahmed Touati, Jérémy Rapin, and Yann Ollivier. Does Zero-Shot Reinforcement Learning Exist? International Conference on Learning Representations (ICLR), 2023.

---

> > ### Comment · Reviewer_J3Z7 · 2025-04-09
> >
> > I thank the authors for their detailed response. Most of my concerns have been addressed. I will keep my positive score.

---

### Official Review · Reviewer_HuPg · 2025-03-15

**Overall Recommendation:** 3

**Summary:**

This paper introduces TD-flow, a novel approach to using the Bellman equation on probability path with flow(score)-matching techniques for generative models. The approach outperforms the existing ones in terms of more accurate generation over extended horizons. TD-flow is validated across diverse experiments and domains. The paper also contributes theoretical analysis.

**Claims And Evidence:**

Yes, the claims are supported by theoretical analysis and experiment validation.

**Essential References Not Discussed:**

N/A

**Experimental Designs Or Analyses:**

The experimental design makes sense to me. And the results look promising

**Methods And Evaluation Criteria:**

Yes, the evaluation is reasonable.

**Other Comments Or Suggestions:**

1. What do theorem 2 and 3 imply intuitively?
2. Line 230, is the Bellman operator missing?
3. Why does Eq(1) hold?
4. Line 55. The MDP definition is different from the RL community, where $\gamma$ acts on reward and value functions. It would be better to make the difference more clear.

**Other Strengths And Weaknesses:**

Strengths:
1. I would say this paper is well-written. For someone like me with limited knowledge of generative modeling, it is easy to follow and effectively presents the key ideas.
2. It studies a very important yet hard problem, accurate generative modeling for long horizons. And the experiment results look quite promising. The RL community needs an accurate world model for sure, which could facilitate the development of model-based RL by resolving the modeling errors and biases.

Weaknesses:
1. It is not clear to me what benefits TD could bring to the flow/score matching and how these benefits help with accurate generative modeling for extended horizons.

**Questions For Authors:**

See above

**Relation To Broader Scientific Literature:**

This paper is very relevant to the generative modeling domain, which is also obvious by its name TD-flow, using TD learning (concept from RL) to aid the score/flowing matching in generative models. I feel this paper can/will receive a lot of attention from the domain.

**Theoretical Claims:**

I roughly went through Section 4 theoretical analysis.

---

> ### Author Rebuttal · Authors · 2025-04-01
>
> We thank the reviewer for the thoughtful and encouraging feedback. We're especially glad to hear that the paper was clear and accessible, even for readers less familiar with generative modeling.
>
> > **What benefits does TD bring to flow/score matching?**
>
>
> This is a great question, and we appreciate you highlighting it. We’ll be sure to clarify it in the revised manuscript.
>
> Temporal Difference (TD) learning, when applied to flow models, brings the classic benefits of TD methods into learning generative models of the successor measure of a given policy. By using current predictions to estimate future outcomes, TD allows us to train on shorter segments of data while still capturing long-horizon structure. More concretely, this provides several key advantages:
>
>
> * **Reduced variance and greater sample efficiency**: Rather than relying on full long-horizon rollouts (which can be noisy, high variance, or unavailable), TD updates allow us to learn from short segments (e.g., 1-step transitions) and still reason about long-term behavior.
> * **“Stitching” across time**: TD enables combining information from different parts of the environment/dataset to construct consistent long-horizon predictions, even when full trajectories are missing or limited.
> * **Off-policy learning**: TD naturally supports learning from off-policy data—that is, trajectories collected from different policies or behaviours. This is critical in the offline setting, where the model must learn from a fixed dataset. Our planning results in particular are only possible because of this capability: TD-Flow can plan effectively using policies not seen during data collection.
>
> > **What do Theorems 2 and 3 imply intuitively?**
>
>
> The theorems compare the variance in estimating the gradient of the loss functions. The smaller the variance, the less the number of data samples and generative samples are needed to have an accurate estimate of the gradient.
> Intuitively, we can think of the gap in variance as being caused by the misalignment of the conditional vector field $u_t$ with the true vector field $v_t$. For conditional paths, under certain conditions (i.e., coupling between $(X_0, X_1)$ and straight non-crossing paths) there is alignment and this gap is zero.
>
> > **Why does Eq(1) hold?**
>
>
> By definition we have,
> $$
> \begin{align*}
> Q(s, a) &= \mathbb{E}_{\pi} \left[ \sum\_{k=1}^\infty \gamma^{k-1} r(S\_{t+k}) \mid S\_t = s, A_t = a \right] \\\\
> &= \sum\_{k=1}^\infty \gamma^{k-1} \int\_{\mathcal{S}} r(x)\\, \mathrm{Pr}(S\_{t+k} = x \mid S\_t = s, A\_t = a; \pi) \mathrm{d}x \\\\
> &= \int\_{\mathcal{S}} r(x) \underbrace{\sum\_{k=1}^\infty\\, \gamma^{k-1} \mathrm{Pr}(S\_{t+k} = x \mid S\_t = s, A\_t = a; \pi) \mathrm{d}x}\_{(1-\gamma)^{-1}m^\pi(x\mid s,a)} \\\\
> &= (1-\gamma)^{-1} \mathbb{E}\_{X \sim m^\pi(\cdot\mid s, a)} \left[ r(X) \right]
> \end{align*}
> $$
>
>
>
> > **The MDP definition is different from the RL community, where γ acts on reward and value functions.**
>
>
> Thank you for pointing this out. You're absolutely right that in standard RL formulations, the discount factor $\gamma$ typically appears in the definition of return, i.e., as a weight on future rewards.
>
> In our setting, since we model the successor measure directly, $\gamma$ instead appears in the distribution over time steps—specifically, it defines a geometric distribution over how far into the future we sample. This reparameterization is equivalent but shifts the emphasis from discounting rewards to discounting the probability of future state occurrences. Importantly, this does not imply that the MDP itself terminates with probability $1 – \gamma$ at each step—rather, it’s a modeling choice that reflects the probabilistic weighting of future events, not actual environment termination.
>
> Since this interpretation doesn’t appear frequently in standard RL formulations, we’ll expand on it in the revised version to help clarify the connection.
>
> ---
>
> We hope this response strengthens your overall impression of our work.

---

### Official Review · Reviewer_uS9Q · 2025-03-22

**Overall Recommendation:** 4

**Summary:**

This paper introduces a new family of algorithms–TD-CFM, Coupled TD-CFM, and TD^2-CFM which propose a Flow-matching-based generative modeling framework to learn to sample from the discounted successor measure. Learning the successor measure rather than predicting long term rollouts prevents the usual accumulations of prediction errors. The paper proposes several extensions of TD-CFM to coupled flows, analogous denoising diffusion models along with an interesting variance reduction technique using boot-strapped updates (TD^2-CFM). Experiments on several simulated control tasks show that these methods yield more accurate predictions and better value estimates improving planning performance.

**Claims And Evidence:**

This paper claims that
-The proposed TD-flow framework can learn a successor measure that more effectively models long-horizon state distributions.
-TD^2-CFM can reduce gradient variance compared to standard TD-CFM by using direct velocity matching under certain conditions.
-TD-flow translates into better downstream value estimation and planning performance in simulated tasks.
Theorem 1 shows that when the TD-CFM is solved exactly, it forms a contraction (in the 1-Wasserstein distance) and Theorem 2-3 shows that variance is reduced through a tighter coupling of bootstrapped targets. Experimental results support improved value function estimation and planning and generally support the claim the TD^2 can improve performance via variance reduction.

**Essential References Not Discussed:**

I'm not familiar enough with offline modeling of successor measure literature to suggest additional references.

**Experimental Designs Or Analyses:**

The experiments are quite thorough. I would like to see some run-time analysis for training the proposed algorithms and baselines. It may also be interesting to better how this planning with the successor measure compares to other finite horizon shooting methods /  model-based planning methods if applicable.

It could be useful to see an ablation as dimension increases or in the presence of noisy data.

**Methods And Evaluation Criteria:**

I think the proposed method is a very interesting solution to efficiently learning successor measures. I'm not exactly sure if successor measures are a necessity for the task of planning compared to other offline methods using direct Q learning or simply finite-horizon MPC. However, their benchmarks directly quantify Q function estimation accuracy and improved planning with generalized policy improvement over finite families of pre-trained policy and show significant improvements.

**Other Comments Or Suggestions:**

A typo on line 190, left column in the equation denominator.

**Other Strengths And Weaknesses:**

Strengths:

-The paper is well-written and I found the ideas generally interesting. It is my impression that the integration of TD learning with flow matching to directly model the successor measure is quite novel with several improvements and variations already proposed here.

-Theoretical analysis provides insights into contraction properties and variance reduction.

-Comprehensive experimental evaluation over several downstream application of successor measures.

Weaknesses:

-Even after the extensive evaluations, I don’t have a good feeling for how this framework compares to more direct methods for learning Q function from offline data or model-based finite-horizon planners (e.g. shooting methods, cross entropy method). Even if there isn't a directly comparable setting, its worth discussing their context more and why the added complexity of learning to sample successor measures is useful.

**Questions For Authors:**

1. How does this framework compare to more direct methods for learning Q function from offline data or model-based finite-horizon planners (e.g. shooting methods, cross entropy method) ? It could be worth discussing the context of these other methods more and why the added complexity of learning to sample successor measures is useful in practice (besides controlling long term errors better).

2. How does the run-time in training and inference compare between the proposed methods and baselines? It would also be nice to get an idea for what dimension the variance and iterative training scheme becomes an issue through a simple ablation study.

After understanding these points, I'm open to increasing my score.

**Relation To Broader Scientific Literature:**

The work builds on several recent advances in generative modeling for reinforcement learning, including generative horizon models. I believe it is a particularly original and interesting application of flow-based generative modeling.

**Theoretical Claims:**

I briefly reviewed the proofs, but did not verify them carefully. Many results follow directly from CFM. The assumption of exactly solving CFM at each iteration for Theorem 1 is reasonable since we are training in the offline setting and CFM objective can be efficiently optimized if the dataset is sufficiently large enough.

---

> ### Author Rebuttal · Authors · 2025-04-01
>
> We thank the reviewer for their thoughtful and constructive feedback, and we’re glad that the novelty and potential of TD-Flow and its variants were appreciated. We respond below to the key points raised and will revise the paper accordingly to better highlight these aspects.
>
> > **Comparison to direct Q-function learning and finite-horizon planning methods**
>
> We agree that placing TD-Flow in the context of existing value-based and model-based planning methods is important. Our approach offers several distinct advantages:
>
>
> * **Generalization across reward functions**: Unlike traditional value-based methods, which are tied to a specific task, TD-Flow learns the successor measure which is reward agnostic. This enables zero-shot transfer to new reward functions, which we demonstrate in our planning experiments—an especially valuable property in offline or multi-task environments.
> * **Long-horizon reasoning**: Because TD-Flow directly learns to sample from long-horizon state distributions, it avoids limitations of standard MPC methods (e.g., shooting or cross-entropy methods). This is particularly beneficial in sparse reward or goal-conditioned tasks, where short-horizon planners fail to make meaningful progress due to limited planning depth.
> * **Planning-time efficiency**: Although training iterative generative models may be more expensive, inference can be significantly more efficient. For example, with a discount factor of 0.99, the expected time horizon is ~100 steps. Whereas a model-based planner would need to simulate 100 one-step transitions to reach that horizon, TD-Flow can sample directly from that distribution in far fewer steps using an ODE solver—often with just 10 integration steps—yielding substantial computational savings at inference time.
>
> To directly assess this comparison, we include results for a Model Predictive Path Integral (MPPI) controller with a learned dynamics model. We train a similar capacity dynamics model to that of TD-Flow before evaluating MPPI with a finite horizon of 32 for locomotion tasks and 128 for maze, where at each step we sample 256 action candidates and perform 10 optimization rounds with 64 elites (top-k actions) per round. The results below show that TD-Flow significantly outperforms MPPI in 3/4  domains with comparable results in Walker. MPPI notably displayed instability related to compounding errors in environments with difficult to model dynamics. We hope this helps clarify the benefits of directly modeling the successor measure.
>
> |  | FB | TD²-CFM-GPI | MPPI |
> |---|:---:|:---:|:---:|
> | Cheetah | 479.35 (14.56) | **693.63 (5.50)** | 541.22 (5.28) |
> | Pointmass | 472.45 (14.40) | **800.99 (8.56)** | 286.43 (54.95) |
> | Quadruped | 627.28 (1.98) | **695.73 (2.07)** | 156.80 (122.89) |
> | Walker | 526.66 (5.94) | 627.63 (7.97) | **658.15 (21.46)** |
>
> > **Run-time analysis and computational cost**
>
>
> This is a valuable point. Both the runtime at training and inference is dominated by the number of iterative steps to solve the ODE or SDE during sampling. A key insight—shown theoretically in Appendix E.7—is that our method naturally minimizes the transport cost which enables the use of a relatively small number of solver steps without significantly compromising accuracy.
>
> To support this claim, we ran an empirical analysis showing how prediction quality degrades as we reduce the number of integration steps on the Loop task in Pointmass Maze. The results below show that TD²-CFM remains robust even at coarse discretizations of the ODE with as little as 5 integration steps, while we observewith a predictable degradation whenas the number of steps is too smalldecreases. Additionally, orthogonal work on consistency models [1] and self-distillation [2] can be applied to TD-Flow to further increase accuracy for few-step generation.
>
> |  | NLL↓ | EMD↓ | VF Error↓ |
> |---|:---:|:---:|:---:|
> | 2 Steps | -0.48 (0.21) | 0.076 (0.003) | 379.52 (81.75) |
> | 5 Steps | -2.75 (0.15) | 0.036 (0.000) | 23.82 (2.05) |
> | 10 Steps | -2.85 (0.17) | 0.025 (0.001) | 7.71 (2.75) |
> | 20 Steps | -2.99 (0.04) | 0.0218 (0.001) | 4.40 (0.82) |
>
> ---
>
>
> We thank the reviewer again for their insightful suggestions and willingness to reconsider their evaluation. We hope our response clarifies the unique strengths of TD-Flow and its potential to the community.
>
> ### References
>
>
> [1] Yang Song, Prafulla Dhariwal, Mark Chen, and Ilya Sutskever. Consistency Models. International Conference on Machine Learning (ICML), 2023.
>
>
> [2] Kevin Frans, Danijar Hafner, Sergey Levine, and Pieter Abbeel. One Step Diffusion via Shortcut Models. International Conference on Learning Representations (ICLR), 2025.
>
>
> [3] Manan Tomar, Philippe Hansen-Estruch, Philip Bachman, Alex Lamb, John Langford, Matthew E Taylor, and Sergey Levine. Video occupancy models. CoRR, abs/2407.09533, 2024.

---

> > ### Comment · Reviewer_uS9Q · 2025-04-07
> >
> > I want to thank the authors for their responses. All of my concerns have been addressed. I have raised my score (3->4).

---

### Decision · Program_Chairs · 2025-05-01

**Decision:**

Accept (oral)

**Comment:**

This paper leverages a novel Bellman equation structure along the probability paths to improve the learning of geometric horizon models, significantly enlarging their prediction horizon. Theoretical analysis is provided accompanied with convincing empirical results. All the reviewers appreciate the solid contribution and the clear presentation. I, therefore, recommend accept.